# Learning Dynamical Systems via Koopman Operator Regression in Reproducing Kernel Hilbert Spaces

**Vladimir R. Kostic**[*]
Istituto Italiano di Tecnologia
University of Novi Sad
`vladimir.kostic@iit.it`

**Pietro Novelli** [*]
Istituto Italiano di Tecnologia
`pietro.novelli@iit.it`

**Andreas Maurer**
Istituto Italiano di Tecnologia
`am@andreas-maurer.eu`

**Carlo Ciliberto**
University College London
`c.ciliberto@ucl.ac.uk`

**Lorenzo Rosasco**
University of Genova
Massachusetts Institute of Technology
Istituto Italiano di Tecnologia
`lrosasco@mit.edu`

**Massimiliano Pontil**
Istituto Italiano di Tecnologia
University College London
`massimiliano.pontil@iit.it`

## Abstract

We study a class of dynamical systems modelled as Markov chains that admit an invariant distribution via the corresponding transfer, or Koopman, operator. While data-driven algorithms to reconstruct such operators are well known, their relationship with statistical learning is largely unexplored. We formalize a framework to learn the Koopman operator from finite data trajectories of the dynamical system. We consider the restriction of this operator to a reproducing kernel Hilbert space and introduce a notion of risk, from which different estimators naturally arise. We link the risk with the estimation of the spectral decomposition of the Koopman operator. These observations motivate a reduced-rank operator regression (RRR) estimator. We derive learning bounds for the proposed estimator, holding both in i.i.d. and non i.i.d. settings, the latter in terms of mixing coefficients. Our results suggest RRR might be beneficial over other widely used estimators as confirmed in numerical experiments both for forecasting and mode decomposition.

## 1 Introduction

Dynamical systems [26, 38] provide a framework to study a variety of complex phenomena in science and engineering. For instance, they find wide applications in diverse fields such as finance [46], robotics [6, 16], atomistic simulations [32, 36, 53], open quantum system dynamics [20, 28], and many more. Because of their practical importance, research around dynamical systems is and has been abundant, see e.g. [17, 58] and references therein.

In light of recent machine learning progress, it is appealing to ask if the properties of dynamical systems can be estimated (*learned*) from empirical data. Beyond machine learning this question has a long history in dynamical systems [7]. The go-to reference for data-driven algorithms to reconstruct dynamical systems is [25], where numerous methods based on so-called dynamic mode

---

[*]Equal contribution, corresponding authors.

36th Conference on Neural Information Processing Systems (NeurIPS 2022).

decomposition (DMD) are discussed along with interesting applications. The literature on various theoretical aspects of dynamical systems is also rich [35, 38]. Our starting observation is that although data-driven algorithms to reconstruct dynamical systems are well known, their relationship with statistical learning [61] is largely unexplored. Our broad goal is to build a tie between these two important areas of research and to establish firm theoretical grounds for data driven approaches, to derive statistical guarantees and a foundation in which learning dynamical systems can be tackled in great generality.

In this paper, we present a framework for *learning* dynamical systems from data obtained from one or multiple trajectories. The focus is both *predicting* the future states of the system and *interpreting* the underlying dynamic. The initial observation is the fact that, under suitable assumptions, a dynamical system can be completely characterized by a *linear operator*, known as Koopman (or transfer) operator [8, 35]. More precisely, the Koopman operator describes how functions (observables) of the state of the system evolve over time along its trajectories. Further, the spectral decomposition of the Koopman operator, along with the mode decomposition, allows us to interpret the dynamical and spatial properties of the system [11, 39, 52]. In view of these results, learning a dynamical system can be cast as the problem of learning the corresponding Koopman operator and associated mode decomposition.

A key insight in our approach is to consider the restriction of Koopman operators to reproducing kernel Hilbert spaces. With this choice, Hilbert-Schmidt operators become the natural hypothesis space and kernel methods can be exploited [24]. We further link the proposed framework to conditional mean embeddings [21, 24, 43]. This allows us to formalize the estimation of the Koopman operator as a risk minimization problem and derive a number of estimators as instances of classical empirical risk minimization under different constraints. We dub the problem Koopman operator regression. In our framework DMD and some of its variants [25] are recovered as special cases. Moreover, our analysis highlights the importance of rank constrained estimators, and, following this observation, we introduce and analyze an estimator akin to reduced rank regression (RRR) [22]. Within our statistical learning framework the learning properties of the studied estimators can be characterized in terms of non asymptotic error bounds derived from concentration of measure results for mixing processes. Theoretical results are complemented by numerical experiments where we investigate the properties the estimators, and show they can be smoothly interfaced with deep learning techniques. We note that, both kernel methods [1, 5, 13, 23, 24, 67] and deep learning approaches [4, 15, 30], have been recently considered to learn Koopman operators. Compared to these works (notably [24], whose setting is closely related to ours), we provide a statistical learning framework connecting to the classical notions of risk, which we further link to the estimation of the spectrum of the Koopman operator. Moreover, we derive non asymptotic and non-i.i.d. learning bounds and introduce and study a novel constrained rank estimator RRR.

**Contributions.** In summary our main contributions are: **1)** We present a statistical learning framework for Koopman operator regression; **2)** We bound the error in estimating the Koopman mode decomposition and its eigenvalues by the risk of an estimator (Theorem 1); **3)** We present a novel reduced-rank estimator and show that it can be computed and used efficiently (Theorem 2); **4)** We provide a statistical risk bound supporting the proposed estimator (Theorem 3) and introduce a new tool (Lemma 1) which is key in extending the bound to the non-i.i.d. setting.

**Notation.** For any non-negative integers $n, m$ with $n > m$ we use the notation $[m{:}n] = \{m, \ldots, n\}$ and $[n] = [1{:}n]$. $L^2_\pi(\mathcal{X}) := L^2(\mathcal{X}, \pi)$ is the space of real valued functions on $\mathcal{X}$, that are square-integrable with respect to $\pi$. Given two separable Hilbert spaces $\mathcal{H}$ and $\mathcal{G}$, we let $\mathrm{HS}\,(\mathcal{H}, \mathcal{G})$ be the Hilbert space of Hilbert-Schmidt (HS) operators from $\mathcal{H}$ to $\mathcal{G}$ endowed with the norm $\|A\|^2_{\mathrm{HS}} \equiv \sum_{i \in \mathbb{N}} \|Ae_i\|^2_{\mathcal{G}}$, for $A \in \mathrm{HS}\,(\mathcal{H}, \mathcal{G})$, where $(e_i)_{i \in \mathbb{N}}$ is an orthonormal basis of $\mathcal{H}$. We use the convention $\mathrm{HS}\,(\mathcal{H}) = \mathrm{HS}\,(\mathcal{H}, \mathcal{H})$. The standard norms in Hilbert spaces and operator norms are denoted by $\|\cdot\|$, where the space is clear from the context. Given an operator $A \in \mathrm{HS}\,(\mathcal{H})$, we denote with $[\![A]\!]_r$ its $r$-truncated singular value decomposition and its $i$-th singualr value by $\sigma_i(A)$.

## 2 Background on Koopman operator theory

We briefly recall the basic notions related to Markov chains and Koopman operators and refer to App. A and [26, 35, 38] for further details.

Let $\boldsymbol{X} := \{X_t \colon t \in \mathbb{N}\}$ be a family of random variables with values in a measurable space $(\mathcal{X}, \Sigma_{\mathcal{X}})$, called state space. We call $\boldsymbol{X}$ a *Markov chain* if $\mathbb{P}\{X_{t+1} \in B \mid X_{[t]}\} = \mathbb{P}\{X_{t+1} \in B \mid X_t\}$. Further, we call $\boldsymbol{X}$ *time-homogeneous* if there exists $p \colon \mathcal{X} \times \Sigma_{\mathcal{X}} \to [0,1]$, called *transition kernel*, such that, for every $(x, B) \in \mathcal{X} \times \Sigma_{\mathcal{X}}$ and every $t \in \mathbb{N}$,

$$\mathbb{P}\{X_{t+1} \in B \mid X_t = x\} = p(x, B).$$

In this work we consider only discrete Markov chains with $t \in \mathbb{N}$, but we note that any continuous Markov process with $t \in \mathbb{R}$ can be reduced to a discrete chain by sampling it at times $t_n = n\Delta t$ with $n \in \mathbb{N}$ and $\Delta t$ fixed. For an alternative approach to approximate continuous dynamics see e.g. [51].

For a set $\mathcal{F}$ of real valued and measurable functions on $\mathcal{X}$, the *Markov transfer operator* $A_{\mathcal{F}} \colon \mathcal{F} \to \mathcal{F}$ is defined as

$$A_{\mathcal{F}} f(x) := \int_{\mathcal{X}} p(x, dy) f(y) = \mathbb{E}\left[ f(X_{t+1}) \mid X_t = x \right], \quad f \in \mathcal{F},\ x \in \mathcal{X}. \tag{1}$$

A possible choice is $\mathcal{F} = L^{\infty}(\mathcal{X})$, the space of bounded functions on $\mathcal{X}$ [26]. We are interested in another common choice related to the existence of an *invariant measure* $\pi$ satisfying $\pi(B) = \int_{\mathcal{X}} \pi(dx) p(x, B)$, $B \in \Sigma_{\mathcal{X}}$. In this case, it is possible to take $\mathcal{F} = L^2_{\pi}(\mathcal{X})$, and easy to see that $\|A_{\mathcal{F}}\| \leq 1$, that is the Markov transfer operator is a bounded linear operator. In the following, we denote by $A_{\pi}$ the Markov transfer operator on $L^2_{\pi}(\mathcal{X})$, and always assume the existence of an invariant measure. We note that, its existence can be proven for large classes of Markov chains, see e.g. [12]. Also, to derive the statistical bounds in Sec. 5 we assume that the Markov chain is mixing [26].

**Example 1.** *An important example of the above construction is given by discrete dynamical systems with additive noise. That is, given a state space $\mathcal{X} \subseteq \mathbb{R}^d$, a mapping $F \colon \mathcal{X} \to \mathcal{X}$ and a probability distribution $\Omega$ on $\mathcal{X}$ we let $X_{t+1} = F(X_t) + \omega_t$, $t \in \mathbb{N}$, where $\omega_t$ are i.i.d. zero mean random variables with law $\Omega$. The corresponding transition kernel is $p(x, B) = \Omega(B - F(x))$, for which the existence of an invariant measure is ensured e.g. when $\Omega$ is absolutely continuous with respect to the Lebesgue measure and its density is strictly positive (see Remark 10.5.4 in [26]).*

**Remark 1.** *Whenever $\pi(dx) p(x, dy) = \pi(dy) p(y, dx)$ the Markov chain is said to be* reversible. *In this case it readily follows that the Koopman operator is self-adjoint $A_{\pi} = A_{\pi}^{*}$. In statistical physics the reversibility condition is also called* detailed balance *and is linked to the symmetry with respect to time reversal. Since a large amount of microscopical equations of motion in both classical and quantum physics are time-reversal invariant, learning self-adjoint Koopman operators is of paramount importance in the field of machine learning for physical sciences.*

**Koopman Operator and Mode Decomposition.** In dynamical systems, $A_{\mathcal{F}}$ is known as the (stochastic) *Koopman operator* on the space of observables $\mathcal{F}$. An important fact is that its linearity can be exploited to compute a spectral decomposition. Indeed, in many situations, and notably for compact Koopman operators, there exist scalars $\lambda_i \in \mathbb{C}$, and observables $\psi_i \in L^2_{\pi}(\mathcal{X})$ satisfying the eigenvalue equation $A_{\pi} \psi_i = \lambda_i \psi_i$. Leveraging the eigenvalue decomposition, the dynamical system can be decomposed into superposition of simpler signals that can be used in different tasks such as system identification and control, see e.g. [7]. More precisely, given an observable $f \in \mathrm{span}\{\psi_i \mid i \in \mathbb{N}\}$ there exist corresponding scalars $\gamma_i^f \in \mathbb{C}$ known as Koopman modes of $f$, such that

$$A_{\pi}^t f(x) = \mathbb{E}[f(X_t) \mid X_0 = x] = \sum_{i \in \mathbb{N}} \lambda_i^t \gamma_i^f \psi_i(x), \quad x \in \mathcal{X},\ t \in \mathbb{N}. \tag{2}$$

This formula is known as *Koopman Mode Decomposition* (KMD) [2, 8]. It decomposes the expected dynamics observed by $f$ into *stationary* modes $\gamma_i^f$ that are combined with *temporal changes* governed by eigenvalues $\lambda_i$ and *spatial changes* governed by the eigenfunctions $\psi_i$. We notice however that the Koopman operator, in general, is not a normal compact operator, hence its eigenfunctions may not form a complete orthonormal basis of the space which makes learning KMD challenging.

In many practical scenarios the transition kernel $p$, hence $A_{\pi}$, is unknown, but data from one or multiple system trajectories are available. We are then interested into learning the Koopman operator, and corresponding mode decomposition, from the data. Next we discuss how to accomplish this task with the aid of kernel methods.

# 3 Statistical Learning Framework

In this section we choose the space of observables $\mathcal{F}$ to be a reproducing kernel Hilbert space (RKHS) and present a framework for learning the Koopman operator on $L^2_\pi(\mathcal{X})$ restricted to this space and the associated Koopman mode decomposition.

**Learning Koopman Operators.** Let $\mathcal{H}$ be an RKHS with kernel $k : \mathcal{X} \times \mathcal{X} \to \mathbb{R}$ [3] and let $\phi : \mathcal{X} \to \mathcal{H}$ be an associated feature map, such that $k(x,y) = \langle \phi(x), \phi(y) \rangle_\mathcal{H}$ for all $x, y \in \mathcal{X}$. We assume that that $k(x,x) < \infty$, $\pi$-almost surely. This ensures that $\mathcal{H} \subseteq L^2_\pi(\mathcal{X})$ and the injection operator $S_\pi : \mathcal{H} \to L^2_\pi(\mathcal{X})$ given by $(S_\pi f)(x) = f(x)$, $x \in \mathcal{X}$ is a well defined Hilbert-Schmidt operator [9, 55]. Then, the Koopman operator restricted to $\mathcal{H}$ is given by

$$Z_\pi := A_\pi S_\pi : \mathcal{H} \to L^2_\pi(\mathcal{X}).$$

Note that unlike $A_\pi$, $Z_\pi$ is Hilbert-Schmidt since $S_\pi$ is so. It is then natural to approximate $Z_\pi$ by means of Hilbert-Schmidt operators. More precisely, for $G \in \mathrm{HS}(\mathcal{H})$ we approximate $Z_\pi$ by $S_\pi G$, and measure the corresponding error as $\|Z_\pi - S_\pi G\|^2_{\mathrm{HS}}$. To that end, given an an orthonormal basis $(h_i)_{i \in \mathbb{N}}$ of $\mathcal{H}$, we introduce the risk

$$\mathcal{R}(G) := \sum_{i \in \mathbb{N}} \mathbb{E}_{x \sim \pi} \mathbb{E}\big[ \left[ h_i(X_{t+1}) - (Gh_i)(X_t) \right]^2 \,|\, X_t = x \big] \tag{3}$$

as the cumulative expected one-step-ahead prediction error over *all* observables in $\mathcal{H}$. One can show (see Prop. (4) in App. B) that such risk can be decomposed as $\mathcal{R}(G) = \mathcal{R}_0 + \mathcal{E}(G)$, where

$$\mathcal{R}_0 := \|S_\pi\|^2_{\mathrm{HS}} - \|Z_\pi\|^2_{\mathrm{HS}} \geq 0 \ \text{ and } \ \mathcal{E}(G) = \|Z_\pi - S_\pi G\|^2_{\mathrm{HS}}, \tag{4}$$

are the *irreducible risk* and the *excess risk*, respectively. As clear from the above discussion, the Koopman operator and corresponding risk are typically not available in practice and what is available is a dataset of observations $\mathcal{D} := (x_i, y_i)^n_{i=1} \in (\mathcal{X} \times \mathcal{X})^n$. Here, $x_i$ and $y_i$ are two consecutive observations of the state of the system. In classical statistical learning, the data is assumed sampled i.i.d. from the joint probability measure $\rho(dx, dy) := \pi(dx) p(x, dy)$. In the case of dynamical systems, it is natural to assume the the data are obtained by sampling a trajectory $y_i = x_{i+1}$, for $i \in [n-1]$. Then, the problem of learning $A_\pi$ on a RKHS, named here Koopman operator regression, reduces to:

$$\text{Given the data } \mathcal{D}, \ \text{ solve } \ \min_{G \in \mathrm{HS}(\mathcal{H})} \mathcal{R}(G). \tag{5}$$

As discussed in Sec. 2, a central idea associated to Koopman operators is the corresponding mode decomposition. It is then natural to ask whether an approximate mode decomposition can be derived from a Koopman estimator. The following proposition provides a useful step in this direction. Here and in the rest of the paper by $\mathrm{cl}(\cdot)$ we denote the closure of a subspace of the Hilbert space, and we say that a finite rank operator $G \in \mathrm{HS}(\mathcal{H})$ is *non-defective* if and only if its matrix representation is non-defective, i.e. (not necessarily unitarily) diagonalizable.

**Proposition 1.** *If* $\mathrm{Im}(Z_\pi) \subseteq \mathrm{cl}(\mathrm{Im}(S_\pi))$, *then for every* $\delta > 0$ *there exists a finite rank non-defective operator* $G \in \mathrm{HS}(\mathcal{H})$ *such that* $\mathcal{E}(G) < \delta$.

The above proposition shows that if the RKHS $\mathcal{H}$ is, up to its closure in $L^2_\pi(\mathcal{X})$, an invariant subspace of the Koopman operator $A_\pi$, then finite rank non-defective HS operators on $\mathcal{H}$ approximate arbitrarily well the restriction of $A_\pi$ onto $\mathcal{H}$. In particular, this is always true for a wide class of kernels, called universal kernels [55], for which $\mathcal{H}$ is dense in $L^2_\pi(\mathcal{X})$, i.e. $\mathrm{cl}(\mathrm{Im}(S_\pi)) = L^2_\pi(\mathcal{X})$.

**Remark 2.** *Since in the above proposition* $\inf_{G \in \mathrm{HS}(\mathcal{H})} \mathcal{E}(G) = 0$, *we can distinguish between two cases depending on whether the infimum is attained or not. In the former case, known in the literature as well-specified case, there exists* $G_\mathcal{H} \in \mathrm{HS}(\mathcal{H})$ *such that* $Z_\pi = S_\pi G_\mathcal{H}$, *which implies that* $G_\mathcal{H} : \mathcal{H} \to \mathcal{H}$ *defines* $\pi$*-a.e. the Koopman operator on the observable space* $\mathcal{H}$, *i.e.* $G_\mathcal{H} f = \mathbb{E}[f(X_{t+1}) \,|\, X_t = \cdot]$ $\pi$*-a.e. for every* $f \in \mathcal{H}$. *In the latter case, known as misspecified case,* $\mathcal{H}$ *does not admit a Hilbert-Schmidt Koopman operator* $\mathcal{H} \to \mathcal{H}$.

**Remark 3.** *In the misspecified case, a Koopman operator on* $\mathcal{H}$ *might still exist as a bounded albeit not Hilbert-Schmidt operator. In App. B.2 we show that for reversible Markov processes (see Rem. 1), if* $\mathrm{Im}(Z_\pi) \subseteq \mathrm{cl}(\mathrm{Im}(S_\pi))$, *there exists an estimator of the Koopman operator* $G$ *such that* $\|Z_\pi - S_\pi G\|$ *is arbitrarily small and* $\|G\| \leq 1$.

**Learning the Koopman mode decomposition.** Techniques to estimate the Koopman mode decomposition (2) from data, are broadly referred to as *Dynamic Mode Decomposition (DMD)* [25]. We next introduce a DMD approach following the discussion above. Let $r \in \mathbb{N}$ and a non-defective $G \in \mathrm{HS}_r(\mathcal{H}) := \{G \in \mathrm{HS}(\mathcal{H}) \mid \mathrm{rank}(G) \leq r\}$. Then, there exists a spectral decomposition of $G$ given by $(\lambda_i, \xi_i, \psi_i)_{i=1}^r$ where $\lambda_i \in \mathbb{C}$ and $\xi_i$ and $\psi_i$ are complex-valued function with components in $\mathcal{H}$, such that $G = \sum_{i=1}^r \lambda_i \psi_i \otimes \overline{\xi}_i$, where $G\psi_i = \lambda_i \psi_i$, $G^* \xi_i = \overline{\lambda}_i \xi_i$ and $\langle \psi_i, \overline{\xi}_j \rangle_{\mathcal{H}} = \delta_{ij}$, where $\delta_{ij}$ is Kronecker delta symbol, $i, j \in [r]$. This implies, for any $f \in \mathcal{H}$, that

$$G^t f = \sum_{i \in [r]} \lambda_i^t \gamma_i^f \psi_i, \quad t \geq 1. \tag{6}$$

The coefficients $\gamma_i^f := \langle f, \overline{\xi}_i \rangle_{\mathcal{H}}$, $i \in [r]$, are called *dynamic modes* of the observable $f$ and expression (6) is known as the DMD corresponding to $G$. Next, we upper bound the error in estimating the mode decomposition of $A_\pi$ (see Eq. (2)) by the DMD of a non-defective operator $G \in \mathrm{HS}_r(\mathcal{H})$.

**Theorem 1.** *Let $G \in \mathrm{HS}_r(\mathcal{H})$ and $(\lambda_i, \xi_i, \psi_i)_{i=1}^r$ its spectral decomposition. Then for every $f \in \mathcal{H}$*

$$\mathbb{E}[f(X_t) \mid X_0 = x] = \sum_{i \in [r]} \lambda_i^t \gamma_i^f \psi_i(x) + \|Z_\pi - S_\pi G\| \, \mathrm{err}^f(x), \quad x \in \mathcal{X}, \tag{7}$$

*where $\mathrm{err}^f \in L_\pi^2(\mathcal{X})$, and $\|\mathrm{err}^f\| \leq (t-1)\|Gf\| + \|f\|$, $t \geq 1$. Moreover, for any $i \in [r]$,*

$$\|A_\pi S_\pi \psi_i - \lambda_i S_\pi \psi_i\| \leq \frac{\|Z_\pi - S_\pi G\| \, \|G\|}{\sigma_r(S_\pi G)} \|S_\pi \psi_i\|. \tag{8}$$

This theorem provides KMD approximation results for the DMD obtained from an estimator $G$ in $\mathrm{HS}_r(\mathcal{H})$. First, since $\|Z_\pi - S_\pi G\| \leq \sqrt{\mathcal{E}(G)}$, equation (7) shows that DMD for an estimate $G \in \mathrm{HS}_r(\mathcal{H})$ incurs an error which is at most proportional to the (square root) of the corresponding excess risk $\mathcal{E}(G)$. Further, the error degrades as $t$ increases. This implies that the prediction error ($t = 1$) is only controlled by the risk but forecasting ($t \geq 1$) will get in general increasingly harder for larger $t$. Second, inequality (8) shows that $(\lambda_i, S_\pi \psi_i)$ is approximately an eigenpair of the Koopman operator $A_\pi$. Indeed, it guarantees that all the eigenfunctions of the estimator $G$, considered as an equivalence class in $L_\pi^2(\mathcal{X})$, approximately satisfy the Koopman eigenvalue equation. Inequality (8) provides a relative error bound controlled by the excess risk $\mathcal{E}(G)$, where the approximation quality worsen as higher ranks are considered. This provides additional motivation to study low rank estimators of $A_\pi$, see Sec. 4. Moreover, we point out that the bounds in Thm. 1 are tight, since on any RKHS $\mathcal{H}$ spanned by a finite-number of Koopman eigenfunctions it exists a finite rank $G_{\mathcal{H}}$ yielding $\|Z_\pi - S_\pi G_{\mathcal{H}}\|_{\mathrm{HS}} = 0$. For additional discussions see Exm. 3 of App. B.1.

We conclude this section with two remarks regarding the estimation of the eigenvalues of $A_\pi$ and a connection to conditional mean embeddings [54].

**Remark 4** (Spectral Estimation). *Eq. (8) alone is not sufficient to derive strong guarantees on how well the spectra of $G$ estimates the spectra of $A_\pi$. While we address this in detail in App. B.1, here we comment that in general it may happen that $S_\pi G \approx A_\pi S_\pi$, while $\mathrm{Sp}(A_\pi)$ is far from $\mathrm{Sp}(G)$. If $A_\pi$ is a normal compact operator, however, for $G \in \mathrm{HS}_r(\mathcal{H})$ and every $i \in [r]$, there exists $\lambda_{\pi,i} \in \mathrm{Sp}(A_\pi)$ such that $|\lambda_i - \lambda_{\pi,i}| \leq \|Z_\pi - S_\pi G\| \, \|G\|/\sigma_r(S_\pi G)$. If additionally $G$ is also normal, then $|\lambda_i - \lambda_{\pi,i}| \leq \sqrt{\mathcal{E}(G)} \, \|S_\pi\|_{\mathrm{HS}}$.*

**Remark 5** (Link to Conditional Mean Embeddings). *The Koopman operator is a specific form of conditional expectation operator and can be studied within the framework of conditional mean embeddings [54]. Here, the goal is to learn the function $g_p \colon \mathcal{X} \to \mathcal{H}$ defined as*

$$g_p(x) := \mathbb{E}[\phi(X_{t+1}) \mid X_t = x] = \int_{\mathcal{X}} p(x, dy)\phi(y), \quad x \in \mathcal{X}, \tag{9}$$

*called the conditional mean embedding (CME) of the conditional probability $p$ into $\mathcal{H}$. In App. B.2 we show a "duality" between Koopman operator regression and CME expressed by the reproducing property $(Z_\pi f)(x) = \langle f, g_p(x) \rangle_{\mathcal{H}}$. In particular, recalling that $\rho$ is the joint probability measure on $\mathcal{X} \times \mathcal{X}$ defined by $\rho(dx, dy) = p(x, dy)\pi(dx)$, the risk we proposed in (3) can be written as*

$$\underbrace{\mathbb{E}_{(x,y)\sim\rho}\|\phi(y) - G^*\phi(x)\|^2}_{\mathcal{R}(G)} = \underbrace{\mathbb{E}_{(x,y)\sim\rho}\|g_p(x) - \phi(y)\|^2}_{\mathcal{R}_0} + \underbrace{\mathbb{E}_{x\sim\pi}\|g_p(x) - G^*\phi(x)\|^2}_{\mathcal{E}(G)} \tag{10}$$

*In this sense the Koopman operator regression problem (5) is equivalent to learning CME of the Markov transition kernel $p$.*

## 4 Empirical Risk Minimization

We next describe different estimators for the Koopman operator. Let $\widehat{S}, \widehat{Z} \in \mathrm{HS}(\mathcal{H}, \mathbb{R}^n)$ be the *sampling* operators of the inputs and outputs defined, for every $f \in \mathcal{H}$, as $\widehat{S}f = \left(n^{-\frac{1}{2}} f(x_i)\right)_{i=1}^n$ and $\widehat{Z}f = \left(n^{-\frac{1}{2}} f(y_i)\right)_{i=1}^n$, respectively. The Koopman operator is estimated by minimizing, under different constraints, the *empirical risk*

$$\widehat{\mathcal{R}}(G) := \left\| \widehat{Z} - \widehat{S}G \right\|_{\mathrm{HS}}^2 = \frac{1}{n} \sum_{i \in [n]} \|\phi(y_i) - G^*\phi(x_i)\|^2, \qquad G \in \mathrm{HS}(\mathcal{H}). \tag{11}$$

The first expression is the empirical version of the risk in (3), while the second expression is the empirical version of the risk as in (10), with a remark that sampling operator $\widehat{Z}$ is not an estimator of the regression operator $Z_\pi$ and that, since $\pi$ is an invariant measure, we have $\mathbb{E}[\widehat{Z}^*\widehat{Z}] = \mathbb{E}[\widehat{S}^*\widehat{S}] = S_\pi^* S_\pi$. Further, notice that for the linear kernel $\phi(x) = x$ Eq. (11) is essentially the problem minimized by the classical DMD, whereas if $\phi$ is built from a dictionary of functions it is minimized by the *extended* DMD [25].

Before discussing further, we introduce the empirical input, output and cross covariances, given by $\widehat{C} := \widehat{S}^*\widehat{S}$, $\widehat{D} := \widehat{Z}^*\widehat{Z}$ and $\widehat{T} := \widehat{S}^*\widehat{Z}$, respectively, and the corresponding kernel Gram matrices given by $K := \widehat{S}\widehat{S}^*$, $L := \widehat{Z}\widehat{Z}^*$ and $M := \widehat{Z}\widehat{S}^*$. We also let $\widehat{C}_\gamma := \widehat{C} + \gamma I_{\mathcal{H}}$ be the regularized empirical covariance and $K_\gamma := K + \gamma I_n$ the regularized kernel Gram matrix.

**Kernel Ridge Regression (KRR).** A natural approach is to add a Tikhonov regularization term to (11) obtaining a *Kernel Ridge Rregression* (KRR) estimator

$$\widehat{G}_\gamma := \arg\min \left\{ \widehat{\mathcal{R}}(G) + \gamma \|G\|_{\mathrm{HS}}^2 : G \in \mathrm{HS}(\mathcal{H}) \right\}. \tag{12}$$

It is easy to see that $\widehat{G}_\gamma = \widehat{C}_\gamma^{-1}\widehat{T} = \widehat{S}^* K_\gamma^{-1}\widehat{Z}$. One issue with the above estimator is that the computation of its spectral decomposition becomes unstable with large datasets, see below. Consequently, low rank estimators have been advocated [25] as a way to overcome these limitations.

**Principal Component Regression (PCR).** A standard strategy to obtain a low-rank estimator is *Principal Component Regression* (PCR). Here, the input data are projected to the principal subspace of the covariance matrix $\widehat{C}$, and ordinary least squares on the projected data is performed, yielding the estimator $\widehat{G}_r^{\mathrm{PCR}} = [\![\widehat{C}]\!]_r^\dagger \widehat{T} = \widehat{S}^* [\![K]\!]_r^\dagger \widehat{Z}$. In the context of dynamical systems, this estimator is known as *kernel Dynamic Mode Decomposition*, and is of utter importance in a variety of applications [7, 25]. Note, however, that PCR does *not* minimize the empirical risk under the low-rank constraint.

**Reduced Rank Regression (RRR).** The optimal rank $r$ empirical risk minimizer is

$$\widehat{G}_{r,\gamma} := \arg\min \left\{ \widehat{\mathcal{R}}(G) + \gamma \|G\|_{\mathrm{HS}}^2 : G \in \mathrm{HS}_r(\mathcal{H}) \right\}. \tag{13}$$

In classical linear regression this problem is known as *reduced rank regression* (RRR) [22]. While extensions to infinite dimensions have been considered [44, 65], we are not aware of any work considering the HS operator setting presented here. The minimizer of (13) is given by $\widehat{G}_{r,\gamma} = \widehat{C}_\gamma^{-\frac{1}{2}} [\![\widehat{C}_\gamma^{-\frac{1}{2}}\widehat{T}]\!]_r = \widehat{S}^* U_r V_r^\top \widehat{Z}$. Here $V_r = K U_r$ and $U_r = [u_1 | \ldots | u_r] \in \mathbb{R}^{n \times r}$ is the matrix whose columns are the $r$ leading eigenvectors of the generalized eigenvalue problem $LK u_i = \sigma_i^2 K_\gamma u_i$, normalized as $u_i^\top K K_\gamma u_i = 1$, $i \in [r]$.

**Forecasting and Modal Decomposition.** The above estimators are all of the form $\widehat{G} = \widehat{S}^* W \widehat{Z}$, for some $n \times n$ matrix $W$. Given $f \in \mathcal{H}$, the one-step-ahead expected value $\mathbb{E}[f(X_{t+1}) \,|\, X_t = x]$ is estimated by

$$[\widehat{G}f](x) = [\widehat{S}^* W \widehat{Z} f](x) = \frac{1}{n} \sum_{i=1}^n (W f_n)_i k(x, x_i),$$

where $f_n = (f(y_i))_{i=1}^n$ and we used the definition of $\widehat{S}$ and $\widehat{Z}$. Computing the above estimator is demanding in large scale settings, mostly because a large kernel matrix needs be stored and manipulated. Predicting with KRR, therefore, is tantamount to solve a linear system of dimension $n$, whereas for rank $r$ estimators we only need matrix multiplications. Notice that the computational complexity of both PCR and RRR estimators is of order $\mathcal{O}(r^2 n^2)$, which is better than the $\mathcal{O}(n^3)$

complexity of KRR. However, a number of recent ideas for scaling kernel methods can be applied for KRR, see e.g. [18, 37] and references therein. Perhaps more importantly, specific to the context of dynamical systems is the fact that an approximate mode decomposition needs to be further computed, requiring the spectral decomposition of $\widehat{G}$. As showed in App. C.2, this reduces to computing the spectral decomposition of an $n \times n$ matrix $WM$, where recall $M = \widehat{Z}\widehat{S}^*$. For KRR computing the spectral decomposition, in general, has complexity $\mathcal{O}(n^3)$ and may become numerically ill-conditioned for theoretically optimal regularization parameters. In contrast, the low rank structure of PCR and RRR estimators allows efficient and numerically stable spectral computation of complexity $\mathcal{O}(r^2 n^2)$ as we show next.

**Theorem 2.** *Let $\widehat{G} = \widehat{S}^* U_r V_r^\top \widehat{Z}$, with $U_r, V_r \in \mathbb{R}^{n \times r}$. If $V_r^\top M U_r \in \mathbb{R}^{r \times r}$ is full rank and non-defective, the spectral decomposition $(\lambda_i, \xi_i, \psi_i)_{i \in [r]}$ of $\widehat{G}$ can be expressed in terms of the spectral decomposition $(\lambda_i, \widetilde{u}_i, \widetilde{v}_i)_{i \in [r]}$ of $V_r^\top M U_r$. Namely, $\xi_i = \widehat{Z}^* V_r \widetilde{u}_i / \overline{\lambda_i}$ and $\psi_i = \widehat{S}^* U_r \widetilde{v}_i$, for all $i \in [r]$. In addition, for every $f \in \mathcal{H}$, its dynamic modes are given by $\gamma_i^f = \widetilde{u}_i^* V_r^\top f_n / (\lambda_i \sqrt{n}) \in \mathbb{C}$.*

In addition to the estimators presented above, several other popular DMD methods are captured by our Koopman operator regression framework. In App. C we review some of them providing also the proof of Thm. 2. We now turn to the study of risk bounds for the proposed low rank estimators.

# 5 Learning Bounds

In this section, we bound the deviation of the risk from the empirical risk, uniformly over a prescribed set of HS operators on $\mathcal{H}$. The analysis here is presented for Ivanov regularization for simplicity, but our results can be linked to Tikhonov regularization (see [29] and reference therein for a discussion). To state the result, we denote (true) input, output and cross covariances by $C := \mathbb{E}_{x \sim \pi}[\phi(x) \otimes \phi(x)]$, $D := \mathbb{E}_{y \sim \pi}[\phi(y) \otimes \phi(y)]$ and $T := \mathbb{E}_{(x,y) \sim \rho}[\phi(x) \otimes \phi(y)]$, respectively. Note that since $\pi$ is invariant measure, the input covariance and output covariance are the same $C = D = S_\pi^* S_\pi$, while for the cross covaraince we have $T = S_\pi^* Z_\pi$. Without loss of generality we present the results in the case that $\|\phi(x)\| \leq 1$, for all $x \in \mathcal{X}$ (the bounds below need otherwise to be rescaled by a constant).

We start by presenting a theorem holding in the setting where the data is sampled i.i.d. from the joint probability measure $\rho(dx, dy) = \pi(dx)p(x, dy)$.

**Theorem 3.** *Let $\mathcal{G}_{r,\gamma} = \{G \in \mathrm{HS}_r(\mathcal{H}) : \|G\|_{\mathrm{HS}} \leq \gamma\}$ and define $\sigma^2 = \mathbb{E}(\|\phi(y)\|^2 - \mathbb{E}\|\phi(y)\|^2)^2$. With probability at least $1 - \delta$ in the i.i.d. draw of $(x_i, y_i)_{i=1}^n$ from $\rho$, we have for every $G \in \mathcal{G}_{\gamma,r}$*

$$|\mathcal{R}(G) - \widehat{\mathcal{R}}(G)| \leq \sqrt{\frac{2\sigma^2 \ln \frac{6}{\delta}}{n}} + 3(4\sqrt{2r}\gamma + \gamma^2)\sqrt{\frac{\|C\| \ln \frac{24n^2}{\delta}}{n}} + \frac{(1 + 24\gamma\sqrt{r}) \ln \frac{6}{\delta} + 6\gamma^2 \ln \frac{24n^2}{\delta}}{n}.$$

A key tool in the proof is the following proposition, which is a natural extension of [34, Theorem 7] who provided concentration inequalities for classes of positive operators.

**Proposition 2.** *With probability at least $1 - \delta$ in the i.i.d. draw of $(x_i, y_i)_{i=1}^n$ from $\rho$,*

$$\left\| \widehat{T} - T \right\| \leq 12 \frac{\ln \frac{8n^2}{\delta}}{n} + 6 \sqrt{\frac{\|C\| \ln \frac{8n^2}{\delta}}{n}}.$$

*Sketch of the proof of Thm. 3.* Recalling the definition of the true and empirical risk, a direct computation gives

$$\mathcal{R}(G) - \widehat{\mathcal{R}}(G) = \mathrm{tr}\left[D - \widehat{D}\right] + \mathrm{tr}\left[GG^*(C - \widehat{C})\right] - 2\,\mathrm{tr}\left[G^*(T - \widehat{T})\right].$$

We use Hölder inequality to bound the last two terms in the r.h.s., obtaining

$$\mathcal{R}(G) - \widehat{\mathcal{R}}(G) \leq \mathrm{tr}\left[D - \widehat{D}\right] + \gamma^2 \|C - \widehat{C}\| + 2\sqrt{r}\gamma \|T - \widehat{T}\|. \tag{14}$$

We then bound the first term as $\mathrm{tr}[D - \widehat{D}] \leq \frac{\ln \frac{2}{\delta}}{3n} + \sqrt{\frac{2\sigma^2 \ln \frac{2}{\delta}}{n}}$, see App. D.1, use [34, Theorem 7-(i)] to bound the second term, and Prop. 2 for the last term. The result then follows by a union bound. $\square$

We state several remarks on this theorem and its implications.

1. It is interesting to compare the bound for RRR and PCR estimators. Assuming that both estimators have the same HS norm, then they will satisfy the same uniform bound. However, the empirical risk may be (possibly much) smaller for the RRR estimator and hence preferable, see Fig. 1.

2. Using the reasoning in [33, 34] and [41] we can replace the variance term and the term $\|C\|$ in the bound with their empirical estimates, obtaining a fully data dependent bound. Notice also that the bound readily applies to the more general CME case, which could be subject of future work.

3. A related bound can be derived using Cor. 3.1 in [40] in place of Prop. 2. This essentially replaces the term $\|C\|$ with $\|\mathbb{E}AA^*\|$ where $A := (\phi(x) \otimes \phi(y) - T)$. This bound is more difficult to turn into a data dependent bound, but it allows for a more direct comparison to bounds without the rank constraint, which may be potentially much larger; see the discussion in App. D.1.

4. One can use the uniform bound to obtain an excess risk bound. In the setting of Thm. 3 and well specified case in which $Z_\pi = S_\pi G_{\mathcal{H}}$ this requires studying the approximation error $\min_{G \in \mathcal{G}_{\gamma, r}} \|S_\pi (G_{\mathcal{H}} - G)\|_{\mathrm{HS}}^2$.

**Dealing with Sampled Trajectories.** We now study ERM with time dependent data. We consider that a trajectory $x_1, \ldots, x_{n+1}$ has been sampled from the process as $x_1 \sim \pi, y_{k-1} = x_k \sim p(x_{k-1}, \cdot)$, $k \in [2{:}n]$. For a strictly stationary Markov process the $\beta$-mixing coefficients are the numbers $\beta_{\mathbf{X}}(\tau)$ defined for $\tau \in \mathbb{N}$ by

$$\beta_{\mathbf{X}}(\tau) = \sup_{B \in \Sigma \otimes \Sigma} |\rho_\tau(B) - (\pi \times \pi)(B)|,$$

where $\rho_\tau$ is the joint distribution of $X_1$ and $X_{1+\tau}$. The basic strategy, going back to at least [68], to transfer a concentration result for i.i.d. variables to the non-i.i.d. case represents the process $\mathbf{X}$ by interlaced block-processes $\mathbf{Y}$ and $\mathbf{Y}'$, which are constructed in a way that $Y_j$ and $Y_{j+1}$ are sufficiently separated to be regarded as independent. Specifically, they are defined as

$$Y_j = \sum_{i=2(j-1)\tau+1}^{(2j-1)\tau} X_i \quad \text{and} \quad Y_j' = \sum_{i=(2j-1)\tau+1}^{2j\tau} X_i \quad \text{for } j \in \mathbb{N}.$$

This construction naturally yields the following key lemma, which allows us to extend several results from the i.i.d. case to time dependent stationary Markov chains. The proof is presented in App. D.2.

**Lemma 1.** *Let $\mathbf{X}$ be strictly stationary with values in a normed space $(\mathcal{X}, \|\cdot\|)$, and assume $n = 2m\tau$ for $\tau, m \in \mathbb{N}$. Moreover, let $Z_1, \ldots, Z_m$ be $m$ independent copies of $Z_1 = \sum_{i=1}^\tau X_i$. Then for $s > 0$*

$$\mathbb{P}\Big\{\Big\|\sum_{i=1}^n X_i\Big\| > s\Big\} \leq 2\,\mathbb{P}\Big\{\Big\|\sum_{j=1}^m Z_j\Big\| > \frac{s}{2}\Big\} + 2\,(m-1)\,\beta_{\mathbf{X}}(\tau).$$

As an application of this result we transfer Prop. 2, which was key in the proof of Thm. 3 to give an estimation bound for $\|T - \widehat{T}\|$, to the non-i.i.d. setting. Fix $\tau \in \mathbb{N}$ and let $Z_1, \ldots, Z_m$ be independent copies of $Z_1 = \frac{1}{\tau} \sum_{i=1}^\tau \phi(x_i) \otimes \phi(x_{i+1}) - T$. Applying Lem. 1 with $\phi(x_i) \otimes \phi(x_{i+1}) - T$ in place of $X_i$ we obtain the following.

**Proposition 3.** *Let $\delta > (m-1)\beta_{\mathbf{X}}(\tau - 1)$. With probability at least $1 - \delta$ in the draw $x_1 \sim \pi, x_i \sim p(x_{i-1}, \cdot)$, $i \in [2{:}n]$,*

$$\|\widehat{T} - T\| \leq \frac{48}{m} \ln \frac{4m\tau}{\delta - (m-1)\beta_{\mathbf{X}}(\tau - 1)} + 12 \sqrt{\frac{2\,\|C\|}{m} \ln \frac{4m\tau}{\delta - (m-1)\beta_{\mathbf{X}}(\tau - 1)}}.$$

We notice that, apart from slightly larger numerical constants and a logarithmic term, Prop. 3 is conceptually identical to Prop. 2 provided the sample size $n$ is replaced by the effective sample size $m \approx n/2\tau$. Similar conclusions can be made to bound the other random terms appearing in (14), see App. D.2 for a discussion.

## 6 Experiments

In this section we show that the proposed framework can be applied to dissect and forecast dynamical systems. While we keep the presentation concise, all the technical aspects, as well as additional

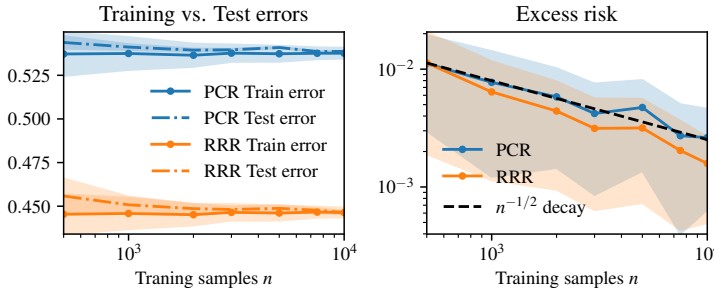

Figure 1: Numerical verification of the uniform bound presented in Thm. 3 for the noisy Logistic map. Left panel: the training and test risk for RRR are consistently than PCR. Right panel: the deviation between training and test risk as a function of the number of training samples have $\approx n^{-1/2}$ decay.

experiments, are deferred to App. E. Along with the code to reproduce the experiments, at the url https://github.com/CSML-IIT-UCL/kooplearn, we release a Python module implementing `sklearn`-compliant [47] estimators to learn the Koopman operator.

**Noisy Logistic Map.** We study the noisy logistic map, a non-linear dynamical system defined by the recursive relation $x_{t+1} = (4x_t(1-x_t)+\xi_t) \mod 1$ over the state space $\mathcal{X} = [0,1]$. Here, $\xi_t$ is i.i.d. additive *trigonometric* noise as defined in [45]. The probability distribution of trigonometric noise is supported in $[-0.5, 0.5]$ and is proportional to $\cos^N(\pi\xi)$, $N$ being an *even* integer. In this setting, the true invariant distribution, transition kernel and Koopman eigenvalues are easily computed. In Tab. 1 we compare the performance of KRR, PCR and RRR (see Sec. 4) trained with a Gaussian kernel. We average over 100 different training datasets each containing $10^4$ data points and evaluate the test error on 500 unseen points. In Tab. 1 we show the approximation error for the three largest eigenvalues of the Koopman operator, $\lambda_1 = 1$ and $\lambda_{2,3} = -0.193 \pm 0.191i$ as well as training and test errors. The following eigenvalues $|\lambda_{4,5}| \approx 0.027$ are an order of magnitude smaller than $|\lambda_{2,3}|$. Both PCR and RRR have been trained with the rank constraint $r = 3$. The regularization parameter $\gamma$ for KRR and RRR is the value $\gamma \in [10^{-7}, 1]$ minimizing the validation error. The RRR estimator always outperforms PCR, and in the estimation of the non-trivial eigenvalues $\lambda_{2,3}$ ($\lambda_1$ corresponding to the equilibrium mode is well approximated by every estimator) attains the best results. In Fig. 1 we report the results of a comparison between PCR and RRR performed under Ivanov regularization. This experiment was designed to empirically test the uniform bounds presented in Sec. 5. Again, RRR consistently outperforms the PCR estimator.

Table 1: Comparison of the estimators proposed in Section 4 on the noisy logistic map.

| Estimator | Training error | Test error | $|\lambda_1 - \hat{\lambda}_1|/|\lambda_1|$ | $|\lambda_{2,3} - \hat{\lambda}_{2,3}|/|\lambda_{2,3}|$ |
|---|---|---|---|---|
| PCR | $0.2 \pm 0.003$ | $0.18 \pm 0.00051$ | $9.6 \cdot 10^{-5} \pm 7.2 \cdot 10^{-5}$ | $0.85 \pm 0.03$ |
| RRR | $0.13 \pm 0.002$ | $\mathbf{0.13 \pm 0.00032}$ | $5.1 \cdot 10^{-6} \pm 3.8 \cdot 10^{-6}$ | $\mathbf{0.16 \pm 0.1}$ |
| KRR | $\mathbf{0.032 \pm 0.00057}$ | $\mathbf{0.13 \pm 0.00068}$ | $\mathbf{7.9 \cdot 10^{-7} \pm 5.7 \cdot 10^{-7}}$ | $0.48 \pm 0.17$ |

**The molecule Alanine Dipeptide.** We analyse a simulation of the small molecule Alanine dipeptide reported in Ref. [66]. The dataset, here, is a time series of the Alanine dipeptide atomic positions. The trajectory spans an interval of 250 ns and the number of features for each data point is 45. The dynamics is Markovian and governed by the Langevin equation [27]. The system supports an invariant distribution, known as the Boltzmann distribution, and the equations are time-reversal-invariant. The latter implies that the true Koopman operator is self-adjoint and has real eigenvalues. For Alanine dipeptide it is well known that the dihedral angles play a special role, and characterize the *state* of the molecule. Broadly speaking, we can associate specific regions of the dihedral angles space to metastable states, i.e. configurations of the molecule which are "stable" over an appreciable span of time. To substantiate this claim we point to the left panel of Figure 2. From this plot it is evident that the molecule spend a large amount of time around specific values of the angle $\psi$, and transitions from one region to another are quite rare. We now try to recover the same informations from the spectral decomposition of the Koopman operator. We train the RRR estimator with rank

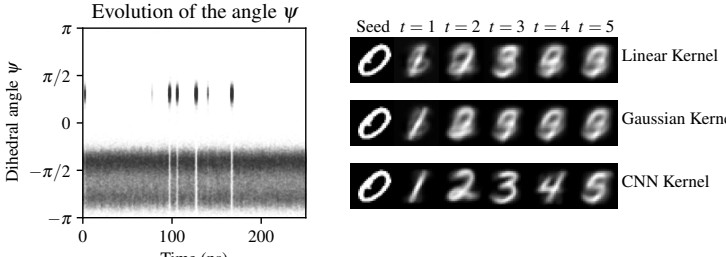

Figure 2: Left: dihedral angle $\psi$ of the Alanine dipeptide as a function of time. Right: comparison of different kernels in the generation of a series of digits. Starting from a seed image, the next ones are obtained by iteratively using the rank-10 RRR Koopman operator estimator.

5 and a standard Gaussian kernel (length scale $\ell = 0.2$). We remark that the dataset is comprised of atomic positions, and not dihedral angles. We show that the computed eigenfunctions are highly correlated with the dihedral angles, meaning that our estimator was able to *learn the correct physical quantities* starting only from the raw atomic positions. The estimated eigenvalues are $\lambda_1 = 0.99920$, $\lambda_2 = 0.9177$, $\lambda_3 = 0.4731$, $\lambda_4 = -0.0042$ and $\lambda_5 = -0.0252$. Notice that they are all real (as they should be, since the system is time-reversal-invariant). In Fig. 6 of App. E.4 we report the plots of the (non-trivial) eigenfunctions corresponding to $\lambda_2$ and $\lambda_3$ in the dihedral angle space. From these plots it is clear that the eigenfunctions are to a good approximation piecewise constant and identify different metastability regions, as expected.

**Koopman Operator Regression with Deep Learning Embeddings.** To highlight the importance of choosing the kernel, here we consider a computer vision setting, where standard kernels (e.g. Matérn or Gaussian), are less suitable than features given by pre-trained deep learning models. We take a sequence $(x_t)_{t \in \mathbb{N}}$ of images from the MNIST dataset [14] starting from $x_0$ corresponding to an image depicting a digit 0 and such that for every $x_t$ depicting a digit $c_t \in \{0, \ldots, 9\}$ we sample $x_{t+1}$ from the set of images depicting the digit $c_t + 1 \pmod{10}$. We compare the rank-10 RRR estimators using Linear and Gaussian kernels, with a *Convolutional Neural Network (CNN) kernel* $k_{\boldsymbol{\theta}}(x, x') := \langle \phi_{\boldsymbol{\theta}}(x), \phi_{\boldsymbol{\theta}}(x') \rangle$, where $\phi_{\boldsymbol{\theta}}$ is a feature map obtained from the last layer of a convolutional neural network classifier trained on the same images in $(x_t)_{t \in \mathbb{N}}$. We trained the three Koopman estimators on 1000 samples. The right panel of Fig. 2 shows the first 4 forecasting steps starting from a digit 0. Only the forecasts by the CNN kernel maintain a sharp (and correct) shape for the predicted digits. In contrast, the other two kernels are less suited to capture visual structures and their predictions quickly lose any resemblance to a digit. This effect can be appreciated starting from other digits, too (see App. E).

## 7   Conclusions

We proposed a statistical framework to learn Koopman operators in RKHS. In addition, we investigated how the spectral and modal decompositions of the Koopman estimators approximate the true ones, providing novel perturbation bounds. In particular, we studied three Koopman operator estimators, KRR, PCR and the newly proposed RRR. We observed that KRR and PCR correspond to well-established estimators from the dynamical systems literature. Then, by leveraging recent results from kernel operator learning we observed that such estimators enjoy strong statistical guarantees. Focusing on the RRR estimator, we provided generalization bounds, both in i.i.d. and non i.i.d. settings, one of the key novel contributions of this work.

In this work we consider only time-homogeneous dynamical systems admitting an invariant distribution. Weakening these assumptions would allow the study of more general systems. Moreover, the extension of our results to continuous dynamical systems and in general to non-uniformly sampled datapoints deserves further investigations. Finally, the choice of the kernel is fundamental for efficient learning. Designing a kernel function incorporating prior knowledge of the dynamical system (such as structure of the data points, symmetries, smoothness assumptions etc.) is a topic of paramount interest.

**Acknowledgements:** This work was supported in part by the EU Projects ELISE and ELSA. We thank all anonymous reviewers for their useful insights and suggestions.

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
