# Supplementary Material

Below we give an overview of the structure of the supplementary material.

- App. A contains additional technical background on Markov processes and Koopman operators, notably on Koopman Mode Decomposition.

- In App. B we provide detailed proofs of the results presented in Sec. 3. In particular, we provide bounds on the distance between spectra of the Koopman operator and its estimation in App. B.1, and discuss a duality between Koopman operator regression (KOR) and conditional mean embeddings (CME) in App. B.2.

- In App. C we expand the content of Sec. 4. In App. C.1 we discuss the computation of three estimators considered in this work, while in App. C.2 we show how to compute their modal decompositions.

- In App. D we prove the statistical learning bounds presented in Sec. 5 and briefly discuss their implications and future research directions.

- Finally, in App. E we provide more details on the experimental section, as well as present additional experiments.

## A    Background on the Koopman Operator Theory

We now recall basic results concerning the theory of Koopman (i.e. transfer) operators. As mentioned in the main text, the natural function space $\mathcal{F}$ in which the Koopman operator can be defined is $\mathcal{F} = L^{\infty}(\mathcal{X})$. In this case, given a transition kernel $p$, by integrating, we can define the *transfer operator* acting either on $L^{\infty}(\mathcal{X})$-measurable functions (from the right) or $\sigma$-finite measures on $\Sigma_{\mathcal{X}}$ (from the left).

**Definition 1** (Transfer operator)**.** *We define the linear transfer operator $P$ acting on the right on functions $f \in L^{\infty}(\mathcal{X})$*

$$(Pf)(x) := \int_{\mathcal{X}} p(x, dy) f(y) = \mathbb{E}\left[f(X_{i+1}) | X_i = x\right] \tag{15a}$$

*and on the left on $\sigma$-finite measures on $\Sigma_{\mathcal{X}}$*

$$(\mu P)(B) := \int_{\mathcal{X}} \mu(dx) p(x, B) \qquad B \in \Sigma_{\mathcal{X}}. \tag{15b}$$

We notice that (15a) acts exactly as the Koopman operator defined in the main text, although on a different function space. Equation (15b), on the other hand, can be interpreted as evolving distributions. Indeed, given an initial distribution of states $\mu$, evolving each state for one step forward, will yield the distribution $\mu P$. If the transition kernel is *non-singular*, that is for all $B \in \Sigma_{\mathcal{X}}$ such that $\mu(B) = 0$ one has $(\mu P)(B) = 0$, in view of the Radon–Nikodym theorem we also have that (15b) can be interpreted as the adjoint of (15a) with respect to the Banach duality pairing between $L^{\infty}(\mathcal{X}, \mu)$ and $L^1(\mathcal{X}, \mu)$, see e.g. [26]. From (15b) it also follows that $\pi$ being an invariant distribution means that $\pi P = \pi$, i.e. it is a fixed point of the transfer operator acting on the left. The following lemma proves that if $\pi$ is an invariant distribution, (15a) is a non-expansive operator in every Lebesgue space $L^q(\mathcal{X}, \pi)$ with $1 \leq q < \infty$.

**Lemma 2.** *If $\pi$ is an invariant probability measure, the operator $P$ is a weak contraction on $L_q(\mathcal{X}, \pi)$ for all $1 \leq q < \infty$. Additionally, it holds that $\|P\| = 1$.*

*Proof.* For the first part, Jensen's inequality and the invariance of $\pi$ directly give

$$\int_{\mathcal{X}} |(Pf)(dx)|^q \pi(dx) = \int_{\mathcal{X}} \pi(dx) \left(\int_{\mathcal{X}} p(x, dy)|f(y)|\right)^q \leq \int_{\mathcal{X}} \pi(dx) \int_{\mathcal{X}} p(x, dy)|f(y)|^q$$

$$= \int_{\mathcal{X}} (\pi P)(dy)|f(y)|^q = \int_{\mathcal{X}} P(dy)|f(y)|^q.$$

For the second part we notice that for any constant function $c$, one has $\|c\| = \|Pc\| \leq \|P\|\|c\|$, that is $\|P\| \geq 1$. This fact coupled with the first part of the lemma yields $\|P\| = 1$. $\qquad\square$

**Corollary 1.** *If $\pi$ is an invariant probability measure, the operator $P$ is well defined in $L_q(\mathcal{X}, \pi)$ for all $1 \leq q < \infty$, and in particular for $q = 2$, the case explored in the main text.*

We just proved that whenever an invariant probability measure $\pi$ exists, (15a) can be defined directly in $L_\pi^2(\mathcal{X})$. An interesting question is therefore what is the equivalent of (15b), seen as the adjoint of (15a) with respect to the Banach duality pairing. To characterize the adjoint operator $P^*$, we define the *time reversal* of $p$ as the Markov transition kernel $p^*(x, B) := \mathbb{P}\{X_{t-1} \in B | X_t = x\}$, and a simple calculation shows that $P^* \colon L_\pi^2(\mathcal{X}) \to L_\pi^2(\mathcal{X})$ is given by:

$$(P^* f)(x) := \int_{\mathcal{X}} p^*(x, dy) f(y), \tag{16}$$

which can be seen as the *backward* transfer operator $[P^* f](x) = \mathbb{E}[f(X_{t-1}) | X_t = x]$. Notice that when the transfer operator on $L_\pi^2(\mathcal{X})$ is self-adjoint, i.e. $P = P^*$, the Markov chain is called *time-reversal invariant* which is a relevant case in various fields such as physics and chemistry [53].

The following example shows that the basic tools developed in the classical theory of (deterministic) dynamical systems [26] can be easily recovered in terms of transfer operators.

**Example 2** (Deterministic Dynamical System). *Let $X_{i+1} = F(X_i)$ for all $i$, with $F : \mathcal{X} \to \mathcal{X}$. Clearly, the transition kernel for this Markov chain is*

$$p(x, B) = \begin{cases} 1 & \text{if } F(x) \in B \\ 0 & \text{otherwise} \end{cases}. \tag{17}$$

*This corresponds to $p(\cdot, A) = \mathbb{1}_B \circ F = \mathbb{1}_{F^{-1}(B)}$, which in turn implies that*

$$(\mu P)(A) = \int_{F^{-1}(A)} \mu(dx). \tag{18}$$

*This is the Perron-Frobenius operator [26] as defined in the classical theory of dynamical systems. Analogously, $p(x, \cdot) = \delta_{F(x)}$ (the Dirac measure centered at $F(x)$) and*

$$(Pf)(x) = f(F(x)) \tag{19}$$

*is the deterministic Koopman operator [26]. When an invariant measure $\pi$ exists, the Koopman operator defined in $L_\pi^2(\mathcal{X})$ is known to be unitary [8] and hence normal. In this respect, see [19], where general misconceptions on the Koopman operator (such as the one of always being a unitary) are discussed in detail.*

We conclude this section recalling the notion of spectra of linear operators. Let $T$ be a bounded linear operator on some Hilbert space $\mathcal{H}$. The *resolvent set* of the operator $T$ is defined as

$$\text{Res}(T) := \{\lambda \in \mathbb{C} \colon T - \lambda I \text{ is bijective}\}.$$

If $\lambda \notin \text{Res}(T)$, then $\lambda$ is said to be in the *spectrum* $\text{Sp}(T)$ of $T$. Recalling that $T - \lambda I$ bijective implies that it has a *bounded* inverse $(T - \lambda I)^{-1}$, in infinite-dimensional spaces we can distinguish three subsets of the spectrum:

1. Any $x \in \mathcal{H}$ such that $x \neq 0$ and $Tx = \lambda x$ for some $\lambda \in \mathbb{C}$ is called an *eigenvector* of $T$ with corresponding *eigenvalue* $\lambda$. If $\lambda$ is an eigenvalue, the operator $T - \lambda I$ is not injective and $\lambda \in \text{Sp}(T)$. The set of all eigenvalues is called the *point spectrum* of $T$.

2. The set of all $\lambda \in \text{Sp}(T)$ for which $T - \lambda I$ is not surjective and the range of $T - \lambda I$ is dense in $\mathcal{H}$ is called the *continuous spectrum*.

3. The set of all $\lambda \in \text{Sp}(T)$ for which $T - \lambda I$ is not surjective and the range of $T - \lambda I$ is not dense in $\mathcal{H}$ is called the *residual spectrum*.

Finally if $T$ is a *compact* operator, the Riesz-Schauder theorem [49], assures that $\text{Sp}(T)$ is a discrete set having no limit points except possibly $\lambda = 0$. Moreover, for any nonzero $\lambda \in \text{Sp}(T)$, then $\lambda$ is an *eigenvalue* (i.e. it belongs to the point spectrum) of finite multiplicity.

# B Learning Theory in RKHS

We begin by proving the identity (3), which we restate in the following proposition.

**Proposition 4.** *Let $G \in \mathrm{HS}\,(\mathcal{H})$ and let $(h_i)_{i \in \mathbb{N}}$ be complete a orthonormal system of $\mathcal{H}$, then*

$$\sum_{i \in \mathbb{N}} \mathbb{E}_{(x,y) \sim \rho} \left[ (S_\pi h_i)(y) - (S_\pi G h_i)(x) \right]^2 = \|S_\pi\|_{\mathrm{HS}}^2 - \|Z_\pi\|_{\mathrm{HS}}^2 + \|Z_\pi - S_\pi G\|_{\mathrm{HS}}^2. \qquad (20)$$

*Proof.* Given $A \colon \mathcal{H} \to L_\pi^2(\mathcal{X})$ for an arbitrary $h \in \mathcal{H}$, denoting $f = S_\pi h$, we have that

$$\mathbb{E}_{(x,y) \sim \rho} \left[ f(y) - (Ah)(x) \right]^2 = \int_{\mathcal{X} \times \mathcal{X}} \pi(dx) p(x, dy) \Big( (f(y)^2 - 2f(y)(Ah)(x) + (Ah)(x)^2 \Big).$$

Using that $\pi(dy) = \int_{\mathcal{X}} \pi(dx) p(x, dy)$, we have both

$$\int_{\mathcal{X} \times \mathcal{X}} \pi(dx) p(x, dy) f(y) = \int_{\mathcal{X}} \pi(dy) f(y) \text{ and } \int_{\mathcal{X} \times \mathcal{X}} \pi(dx) p(x, dy) f(y)^2 = \int_{\mathcal{X}} \pi(dy) f(y)^2.$$

A direct computation then gives that

$$\mathbb{E}_{(x,y) \sim \rho} \left[ f(y) - (Ah)(x) \right]^2 = \|f\|^2 - \|A_\pi f\|^2 + \|A_\pi f - Ah\|^2.$$

Replacing $A$ and in the above expression by $S_\pi G$ and summing over $i \in \mathbb{N}$, we obtain

$$\sum_{i \in \mathbb{N}} \mathbb{E}_{(x,y) \sim \rho} \left[ (S_\pi h_i)(y) - (S_\pi G h_i)(x) \right]^2 = \|S_\pi\|_{\mathrm{HS}}^2 - \|Z_\pi\|_{\mathrm{HS}}^2 + \|Z_\pi - S_\pi G\|_{\mathrm{HS}}^2.$$

Now, since $\|Z_\pi - S_\pi G\|_{\mathrm{HS}} < \infty$, by Tonelli's theorem we can exchange summation and expectation $\mathbb{E}_{x \sim \pi}$, and the proof is completed. We remark that in the risk definition (3) in the main text, we slightly abused notation as $h_i \in \mathcal{H}$, but the expectation value is defined in $L_\pi^2(\mathcal{X})$. The formally correct version of (3) is obtained with the substitution $h_i \mapsto S_\pi h_i$. $\qquad \square$

Next, we prove our main result on the approximation of the Koopman operator via RKHS. We show that if an RKHS $\mathcal{H}$ is, up to its closure in $L_\pi^2(\mathcal{X})$, invariant subspace of the Koopman operator $A_\pi$, then finite rank non-defective operators on $\mathcal{H}$ approximate arbitrarily well the restriction of $A_\pi$ onto $\mathcal{H}$.

**Proposition 1.** *If $\mathrm{Im}(Z_\pi) \subseteq \mathrm{cl}(\mathrm{Im}(S_\pi))$, then for every $\delta > 0$ there exists a finite rank non-defective operator $G \in \mathrm{HS}\,(\mathcal{H})$ such that $\mathcal{E}(G) < \delta$.*

*Proof.* Let us start by observing that $Z_\pi \in \mathrm{HS}\left( \mathcal{H}, L_\pi^2(\mathcal{X}) \right)$, according to the spectral theorem for positive self-adjoint operators, has an SVD, i.e. there exists at most countable positive sequence $(\sigma_j)_{j \in J}$, where $J := \{1, 2, \dots, \} \subseteq \mathbb{N}$, and ortho-normal systems $(\ell_j)_{j \in J}$ and $(h_j)_{j \in J}$ of $\mathrm{cl}(\mathrm{Im}(Z_\pi))$ and $\mathrm{Ker}(Z_\pi)^\perp$, respectively, such that $Z_\pi h_j = \sigma_j \ell_j$ and $Z_\pi^* \ell_j = \sigma_j h_j$, $j \in J$.

Now, recalling that $[\![ \cdot ]\!]_r$ denotes the $r$-truncated SVD, i.e. $[\![ Z_\pi ]\!]_r = \sum_{j \in [r]} \sigma_j \ell_j \otimes h_j$, since $\|Z_\pi - [\![ Z_\pi ]\!]_r\|_{\mathrm{HS}}^2 = \sum_{j > r} \sigma_j^2$, for every $\delta > 0$ there exists $r \in \mathbb{N}$ such that $\|Z_\pi - [\![ Z_\pi ]\!]_r\|_{\mathrm{HS}} < \delta/3$.

Next, since $\mathrm{Im}(Z_\pi) \subseteq \mathrm{cl}(\mathrm{Im}(S_\pi))$, for every $j \in [r]$, we have that $\ell_j \in \mathrm{cl}(\mathrm{Im}(Z_\pi)) \subseteq \mathrm{cl}(\mathrm{Im}(S_\pi))$, which implies that there exists $g_j \in \mathcal{H}$ s.t. $\|\ell_j - S_\pi g_j\| \leq \frac{\delta}{3r}$, and, denoting $B_r := \sum_{j \in [r]} \sigma_j g_j \otimes h_j$ we conclude $\|[\![ Z_\pi ]\!]_r - S_\pi B_r\|_{\mathrm{HS}} \leq \delta/3$.

Finally we recall that the set of non-defective matrices is dense in the space of matrices [59], implying that the set of non-defective rank-$r$ linear operators is dense in the space of rank-$r$ linear operators on a Hilbert space. Therefore, there exists a non-defective $G \in \mathrm{HS}_r(\mathcal{H})$ such that $\|G - B_r\|_{\mathrm{HS}} < \delta/(3\sigma_1(S_\pi))$. So, we conclude

$$\|Z_\pi - S_\pi G\|_{\mathrm{HS}} \leq \|Z_\pi - [\![ Z_\pi ]\!]_r\|_{\mathrm{HS}} + \|[\![ Z_\pi ]\!]_r - S_\pi B_r\|_{\mathrm{HS}} + \|S_\pi(G - B_r)\|_{\mathrm{HS}} = \delta.$$

$\square$

As a consequence of the previous result, we see that if $\mathcal{H}$ (as a subspace of $L^2_\pi(\mathcal{X})$) is spanned by finitely many Koopman eigenfunctions, we have that $Z_\pi$ can be approximated arbitrarily well. In practice such an assumption is not easy to check. On the other hand, for universal kernels we have that $\mathrm{Im}(Z_\pi) \subseteq L^2_\pi(\mathcal{X}) = \mathrm{cl}(\mathrm{Im}(S_\pi))$, and hence we can learn $Z_\pi$ arbitrarily well.

We end this section with a brief discussion of the well-specified and misspecified cases mentioned in Rem. 2 and prove the claim of Rem. 3 in the proposition that follows. To discuss this, we first introduce the following Tikhonov regularized version of problem (5),

$$\min_{G \in \mathrm{HS}(\mathcal{H})} \mathcal{R}(G) + \gamma \|G\|^2_{\mathrm{HS}}, \ \ \gamma > 0 \tag{21}$$

and note, by strong convexity, that its unique solution is given by $G_\gamma = (S^*_\pi S_\pi + \gamma I_\mathcal{H})^{-1} S^*_\pi Z_\pi$.

*The well-specified case:* There exists $G_\mathcal{H} \in \mathrm{HS}(\mathcal{H})$ such that $Z_\pi = S_\pi G_\mathcal{H}$. In this case, $\mathcal{H}$ as a subspace of $L^2_\pi(\mathcal{X})$ is an invariant subspace of $A_\pi$, and, hence, $G_\mathcal{H} \colon \mathcal{H} \to \mathcal{H}$ defines $\pi$-a.e. the Koopman operator on the observable space $\mathcal{H}$, i.e. $G_\mathcal{H} f = \mathbb{E}[f(X_{t+1}) \,|\, X_t = \cdot] \ \pi$-a.e. for every $f \in \mathcal{H}$. Moreover, in this case one has that $G_\mathcal{H} = (S^*_\pi S_\pi)^\dagger S^*_\pi Z_\pi = \lim_{\gamma \to 0} G_\gamma$, where $(\cdot)^\dagger$ denotes the densely defined Moore–Penrose pseudoinverse operator [54].

*The misspecified case:* is when RKHS $\mathcal{H}$ as a space of observables doesn't admit Hilbert-Schmidt $\pi$-a.e. Koopman operator. This can clearly bring difficulties in learning $A_\pi$ since, while one reduces $\mathcal{E}(G)$, the HS norm $\|G\|_{\mathrm{HS}}$ may become progressively large. Note, however, that in this case it still might happen that operator norm $\|G\|$ stays bounded, and, even more, that $\mathcal{H}$ as a subspace of $L^2_\pi(\mathcal{X})$ is an invariant set of $A_\pi$, i.e. $\mathrm{Im}(Z_\pi) \subseteq \mathrm{Im}(S_\pi)$. As the following result shows, this always happens when one learns self-adjoint Koopman operator via a universal kernel.

**Proposition 5.** *If the Markov process is reversible and $\mathrm{Im}(Z_\pi) \subseteq \mathrm{cl}(\mathrm{Im}(S_\pi))$, then $\|G_\gamma\| \le 1$ and for every $\varepsilon > 0$, there exists $\gamma > 0$ such that $\|Z_\pi - S_\pi G_\gamma\| \le \varepsilon$.*

*Proof.* Let us start by observing that $S_\pi \in \mathrm{HS}\left(\mathcal{H}, L^2_\pi(\mathcal{X})\right)$, according to the spectral theorem for positive self-adjoint operators, has an SVD, i.e. there exists at most countable positive sequence $(\sigma_j)_{j \in J}$, where $J := \{1, 2, \dots, \} \subseteq \mathbb{N}$, and ortho-normal systems $(\ell_j)_{j \in J}$ and $(h_j)_{j \in J}$ of $\mathrm{cl}(\mathrm{Im}(S_\pi))$ and $\mathrm{Ker}(S_\pi)^\perp$, respectively, such that $S_\pi h_j = \sigma_j \ell_j$ and $S^*_\pi \ell_j = \sigma_j h_j$, $j \in J$.

Using the above, we first prove that there exists a positive real non-increasing sequence $(\gamma_n)_{n \in \mathbb{N}}$ such that $\lim_{n \to \infty} \gamma_n = 0$ and $\lim_{n \to \infty} \|Z_\pi - S_\pi G_\gamma\| = 0$. To that end, let $P$ and $Q$ denote orthogonal projectors in $L^2_\pi(\mathcal{X})$ onto $\mathrm{cl}(\mathrm{Im}(S_\pi))$ and in $\mathcal{H}$ onto $\mathrm{Ker}(S_\pi)^\perp$, respectively, i.e. $P = \sum_{j \in J} \ell_j \otimes \ell_j$ and $Q = \sum_{j \in J} h_j \otimes h_j$. So, for every $\gamma > 0$ we have

$$\|Z_\pi - S_\pi G_\gamma\| = \|P Z_\pi - S_\pi G_\gamma\| = \|(P - S_\pi (S^*_\pi S_\pi + \gamma I_\mathcal{H})^{-1} S^*_\pi) Z_\pi\|,$$

where the first equality is due to the fact that $\mathrm{Im}(Z_\pi) \subseteq \mathrm{cl}(\mathrm{Im}(S_\pi))$. Moreover, we have that $S_\pi = \sum_{j \in J} \sigma_j \ell_j \otimes h_j$, and, hence,

$$P - S_\pi (S^*_\pi S_\pi + \gamma I_\mathcal{H})^{-1} S^*_\pi = \sum_{j \in J} \frac{\gamma}{\gamma + \sigma_j^2} \ell_j \otimes \ell_j \preceq I_{L^2_\pi(\mathcal{X})}, \text{ and}$$

$$S^*_\pi (P - S_\pi (S^*_\pi S_\pi + \gamma I_\mathcal{H})^{-1} S^*_\pi)^2 S_\pi = \sum_{j \in J} \frac{\gamma^2 \sigma_j^2}{(\gamma + \sigma_j^2)^2} h_j \otimes h_j \preceq \gamma Q.$$

imply $\|P - S_\pi (S^*_\pi S_\pi + \gamma I_\mathcal{H})^{-1} S^*_\pi\| \le 1$ and $\|(P - S_\pi (S^*_\pi S_\pi + \gamma I_\mathcal{H})^{-1} S^*_\pi) S_\pi\| \le \sqrt{\gamma}$, respectively.

Now, since $\mathrm{Im}(Z_\pi) \subseteq \mathrm{cl}(\mathrm{Im}(S_\pi))$, according to Prop. 1, for every $n \in \mathbb{N}$ there exists a finite rank operator $B_n \colon \mathcal{H} \to \mathcal{H}$ such that $\|Z_\pi - S_\pi B_n\| \le 1/n$. Thus, denoting $Q_j := \sum_{i \in [j]} h_i \otimes h_i$, $j \in J$, we have that for every $n \in \mathbb{N}$ there exists $j_n \in J$ such that

$$\|S_\pi (Q - Q_{j_n}) B_n\| \le \|S_\pi (Q - Q_{j_n}\|) \|B_n\| = \|S_\pi - [\![S_\pi]\!]_{j_n}\| \|B_n\| \le 1/n,$$

and, hence, $\|Z_\pi - S_\pi Q_{j_n} B_n\| \le 2/n$.

Therefore, for every $n \in \mathbb{N}$, there exists $j_n \in J$ such that for every $\gamma > 0$ it holds that

$$\|Z_\pi - S_\pi G_\gamma\| = \|(P - S_\pi (S^*_\pi S_\pi + \gamma I_\mathcal{H})^{-1} S^*_\pi)(Z_\pi \pm S_\pi Q_{j_n} B_n)\| \le 2/n + \sqrt{\gamma} \|Q_{j_n} B_n\|.$$

On the other hand, for the bounded operator $Q_{j_n} B_n$, let $h \in \mathcal{H}$ be such that $\|h\| = 1$ and $\|Q_{j_n} B_n h\| = \|Q_{j_n} B_n\|$. So, since $Q_{j_n} B_n h \in \mathrm{Ker}(S_\pi)^\perp$,

$$\frac{\|S_\pi Q_{j_n} B_n\|}{\|Q_{j_n} B_n\|} = \frac{\|S_\pi Q_{j_n} B_n\| \|h\|}{\|Q_{j_n} B_n h\|} \geq \frac{\|S_\pi Q_{j_n} B_n h\|}{\|Q_{j_n} B_n h\|} = \frac{\|S_\pi Q_{j_n} Q_{j_n} B_n h\|}{\|Q_{j_n} B_n h\|} \geq \sigma_{\min}^+(S_\pi Q_{j_n}) = \sigma_{j_n},$$

and, thus, $\|Q_{j_n} B_n\| \leq \|S_\pi Q_{j_n} B_n\|/\sigma_{j_n} \leq (\|Z_\pi\| + 2/n)/\sigma_{j_n}$. So, defining a sequence $\gamma_n := \frac{1}{n^2} \sigma_{j_n}^2$, we obtain

$$\|Z_\pi - S_\pi G_{\gamma_n}\| \leq \frac{1}{n} \left( \|S_\pi\| + \frac{2}{n} \right) + \frac{2}{n},$$

which converges to zero as $n \to \infty$.

Finally, since one has that $S_\pi^* Z_\pi$ is self-adjoint and that $S_\pi^* Z_\pi \preceq S_\pi^* S_\pi$,

$$\|G_\gamma\|^2 = \|(S_\pi^* S_\pi + \gamma I_\mathcal{H})^{-1}(S_\pi^* Z_\pi)^2(S_\pi^* S_\pi + \gamma I_\mathcal{H})^{-1}\| \leq 1.$$

$\square$

## B.1 Approximating Koopman Mode Decomposition by DMD

In this section we prove results stated in Thm. 1 and Rem. 4.

**Theorem 1.** *Let $G \in \mathrm{HS}_r(\mathcal{H})$ and $(\lambda_i, \xi_i, \psi_i)_{i=1}^r$ its spectral decomposition. Then for every $f \in \mathcal{H}$*

$$\mathbb{E}[f(X_t) \,|\, X_0 = x] = \sum_{i \in [r]} \lambda_i^t \gamma_i^f \psi_i(x) + \|Z_\pi - S_\pi G\| \,\mathrm{err}^f(x), \quad x \in \mathcal{X}, \tag{7}$$

*where $\mathrm{err}^f \in L_\pi^2(\mathcal{X})$, and $\|\mathrm{err}^f\| \leq (t-1)\|Gf\| + \|f\|$, $t \geq 1$. Moreover, for any $i \in [r]$,*

$$\|A_\pi S_\pi \psi_i - \lambda_i S_\pi \psi_i\| \leq \frac{\|Z_\pi - S_\pi G\| \|G\|}{\sigma_r(S_\pi G)} \|S_\pi \psi_i\|. \tag{8}$$

*Proof.* Given $f \in \mathcal{H}$, denote $g := (Z_\pi - S_\pi G)f$, and $g_i := (Z_\pi - S_\pi G)\psi_i$, $i \in [r]$. Then, for every $t \geq 1$ we have $A_\pi^t S_\pi f = A_\pi^{t-1} Z_\pi f = A_\pi^{t-1} S_\pi G f + A_\pi^{t-1} g$. Hence, using $S_\pi G f = \sum_{i=1}^r \lambda_i \gamma_i^f S_\pi \psi_i$ and $Z_\pi \psi_i = \lambda_i S_\pi \psi_i + g_i$, $i \in [r]$, we obtain

$$A_\pi^t S_\pi f = A_\pi^{t-1} \Big( \sum_{i=1}^r \lambda_i \gamma_i^f S_\pi \psi_i \Big) + A_\pi^{t-1} g = A_\pi^{t-2} \Big( \sum_{i=1}^r \lambda_i \gamma_i^f Z_\pi \psi_i \Big) + A_\pi^{t-1} g$$

$$= A_\pi^{t-2} \Big( \sum_{i=1}^r \lambda_i^2 \gamma_i^f S_\pi \psi_i \Big) + A_\pi^{t-2} \Big( \sum_{i=1}^r \lambda_i \gamma_i^f g_i \Big) + A_\pi^{t-1} g$$

$$= \cdots$$

$$= \sum_{i=1}^r \lambda_i^t \gamma_i^f S_\pi \psi_i + \Big( \sum_{k=0}^{t-2} A_\pi^k \Big) \Big( \sum_{i=1}^r \lambda_i \gamma_i^f g_i \Big) + A_\pi^{t-1} g.$$

However, having that $A_\pi^{t-1} g = A_\pi^{t-1}(Z_\pi - S_\pi G)f$ and

$$\sum_{i=1}^r \lambda_i \gamma_i^f g_i = \sum_{i=1}^r \lambda_i g_i \langle f, \overline{\xi}_i \rangle_\mathcal{H} = \sum_{i=1}^r \lambda_i (Z_\pi - S_\pi G)\psi_i \langle f, \overline{\xi}_i \rangle_\mathcal{H} = (Z_\pi - S_\pi G)Gf$$

we obtain

$$A_\pi^t S_\pi f - \sum_{i=1}^r \lambda_i^t \gamma_i^f S_\pi \psi_i = \Big( \sum_{k=0}^{t-2} A_\pi^k \Big)(Z_\pi - S_\pi G)Gf + A_\pi^{t-1}(Z_\pi - S_\pi G)f.$$

So, to conclude (7), it suffices to recall that $\|A_\pi\| = 1$ and apply norm in $L_\pi^2(\mathcal{X})$

$$\|A_\pi^t S_\pi f - \sum_{i=1}^r \lambda_i^t \gamma_i^f S_\pi \psi_i\| \leq \|Z_\pi - S_\pi G\|((t-1)\|Gf\| + \|f\|)$$

We now prove (8). Since $g_i = Z_\pi \psi_i - S_\pi(\lambda_i \psi_i) = A_\pi(S_\pi \psi_i) - \lambda_i(S_\pi \psi_i)$, $i \in [r]$, we obtain that

$$\|(A_\pi(S_\pi \psi_i) - \lambda_i(S_\pi \psi_i)\| = \|g_i\| \leq \|Z_\pi - S_\pi G\| \|\psi_i\|.$$

However, since $\psi_i \in \mathrm{Im}(G) \setminus \{0\}$, there exists $h_i \in \mathrm{Ker}(G)^\perp$ so that $\psi_i = G h_i$. Recalling that $C_\gamma = C + \gamma I_\mathcal{H}$ is positive definite for $\gamma > 0$, we have that $\mathrm{Im}(G^*) = \mathrm{Im}(G^* C_\gamma^{1/2})$, and, consequently, $\mathrm{Ker}(G)^\perp = \mathrm{Ker}(C_\gamma^{1/2} G)^\perp$. Thus, since

$$\inf_{h \in \mathrm{Ker}(C_\gamma^{1/2} G)^\perp} \frac{\|C_\gamma^{1/2} G h\|}{\|h\|} = \sigma_{\min}^+(C_\gamma^{1/2} G) = \sigma_r(C_\gamma^{1/2} G),$$

we obtain that $\|C_\gamma^{1/2} \psi_i\| \geq \sigma_r(C_\gamma^{1/2} G) \|h_i\|$, which letting $\gamma \to 0$ implies that $\|S_\pi \psi_i\| \geq \sigma_r(S_\pi G) \|h_i\|$. Hence, we derive $\|\psi_i\| \leq \|G\| \|h_i\| \leq \|G\| \|S_\pi \psi_i\| / \sigma_r(S_\pi G)$, which proves (8). $\qquad \square$

In the following example we show that the bound (8) w.r.t. arbitrary estimator is tight.

**Example 3.** *As a specific instance of Exm. 1 is the equidistant sampling of Ornstein–Uhlenbeck process $X_{t+1} = F X_t + \omega_t$, where $F \in \mathbb{R}^{d \times d}$ and the noise $\omega_t$ is Gaussian. For simplicity, let $F = F^*$ with eigenvalues $\lambda_i$ in $]0, 1[$, and let the noise be i.i.d. from $\mathcal{N}(0, I_d)$. It is well-known [38, Chapter 10.5], that the invariant distribution is $\mathcal{N}(0, C)$, where $C = (I_d - F^2)^{-1}$. If the linear kernel is used, it is readily checked that the corresponding RKHS $\mathcal{H}$ is a closed invariant subspace of $A_\pi$ and, moreover, $Z_\pi = S_\pi G_\mathcal{H}$, where $G_\mathcal{H}$ is given by $F$. Now, consider the rank-$r$ estimator $G = 2[\![G_\mathcal{H}]\!]_r$. Denoting $\beta_i = \lambda_i / \sqrt{1 - \lambda_i^2} > 0$, $i \in [n]$, we have that $\mathcal{E}(G) = \|Z_\pi - S_\pi G\| = \|(I_d - F^2)^{-1/2}(F - 2[\![F]\!]_r)\| = \beta_1$, $\|G\| = 2\|F\| = 2\lambda_1$ and $\sigma_r(S_\pi G) = 2\|(I_d - F^2)^{-1/2}[\![F]\!]_r\| = 2\beta_r$. Therefore, for every eigenpair $(\lambda_i, v_i)$ of $F$, $i \in [r]$, we have $\psi_i = \langle v_i, \cdot \rangle_\mathcal{H} \in \mathcal{H}$ and $G \psi_i = 2\lambda_i \psi_i$, so, consequently, $\|(A_\pi(S_\pi \psi_i) - 2\lambda_i(S_\pi \psi_i)\| = \lambda_i \|S_\pi \psi_i\|$. Therefore, assuming that $\lambda_1 = \ldots = \lambda_r$, for this estimator, we attain equality in (8) for all $i \in [r]$.*

We conclude this section with the result that links the introduced risk to two key concepts of eigenvalue perturbation analysis. First is Stewart's definition of *spectral separation* between two bounded operators on (possibly different) Hilbert spaces, see [56],

$$\mathrm{sep}(A, B) := \min_{\|C\|_{\mathrm{HS}}=1} \|AC - CB\|_{\mathrm{HS}}, \tag{22}$$

and the second is pseudospectrum of bounded linear operators, see [59],

$$\mathrm{Sp}_\varepsilon(A) := \bigcup_{\|B\| \leq \varepsilon} \mathrm{Sp}(A + B) = \{z \in \mathbb{C} \mid \|(A - zI)^{-1}\|^{-1} \leq \varepsilon\}, \tag{23}$$

with the convention $\|(A - zI)^{-1}\|^{-1} = 0$ whenever $z$ is not in the resolvent set of $A$, i.e. $z \in \mathrm{Sp}(A)$.

**Corollary 2.** *If the eigenfunctions of $G \in \mathrm{HS}_r(\mathcal{H})$ are not $\pi$-a.e. zero, then*

$$\mathrm{sep}(A_\pi, G) \leq \sqrt{\mathcal{E}(G)} \|S_\pi\|_{\mathrm{HS}}, \tag{24}$$

*and*

$$\mathrm{Sp}(G) \subseteq \mathrm{Sp}_\varepsilon(A_\pi), \quad for \ \varepsilon = \|Z_\pi - S_\pi G\| \|G\| / \sigma_r(S_\pi G). \tag{25}$$

*Consequently, if $A_\pi$ is normal, then for every $\lambda \in \mathrm{Sp}(G)$ there exists $\lambda_\pi \in \mathrm{Sp}(A_\pi)$ such that $|\lambda_\pi - \lambda| \leq \|Z_\pi - S_\pi G\| \|G\| / \sigma_r(S_\pi G)$. If additionally $G$ is normal, then $|\lambda_\pi - \lambda| \leq \sqrt{\mathcal{E}(G)} \|S_\pi\|_{\mathrm{HS}}$.*

*Proof.* Inequality (24) is a direct consequence of the definition of the separation. On the other hand, (25) follows immediately from (8) and the fact that

$$\|(z I_{L_\pi^2(\mathcal{X})} - A_\pi)^{-1}\|_{L_\pi^2(\mathcal{X})}^{-1} = \min_{f \in L_\pi^2(\mathcal{X})} \frac{\|A_\pi f - z f\|}{\|f\|}, \quad z \in \mathrm{Res}(A_\pi), \tag{26}$$

by taking $S_\pi \psi_i \neq 0$ in place of $f$ and $\lambda_i$ in place of $z$.

Now, using that, see [59], for any normal operator $A$

$$\min_{z' \in \mathrm{Sp}(A)} |z - z'| \le \varepsilon, \quad z \in \mathrm{Sp}_\varepsilon(A), \tag{27}$$

and that for any two normal operators $A$ and $B$

$$\mathrm{sep}(A, B) = \min\{|z - z'| \mid z \in \mathrm{Sp}(A), z' \in \mathrm{Sp}(B)\}, \tag{28}$$

the last two statements follow. $\qquad\square$

**Remark 6.** *In App. A we discussed two important cases in which Koopman operator on $L_\pi^2(\mathcal{X})$ is normal, namely the case of deterministic dynamical systems when $A_\pi$ is unitary (cf. Ex. 2) and the case of time reversible Markov chains, i.e. when $A_\pi$ is self-adjoint (cf. Rem. 1). In such cases, the previous result motivates one to consider normal estimators of the Koopman operator.*

## B.2 Duality Between KOR and CME

In this section we clarify Rem. 5 on the relationship between conditional mean embeddings (CME) and Koopman operator regression (KOR). Recalling the definition of CME in (9) and the restriction of the Koopman operator on $\mathcal{H}$, $Z_\pi := A_\pi S_\pi$, it is easy to see that for every $f \in \mathcal{H}$ it holds

$$(Z_\pi f)(x) = \mathbb{E}[f(X_{t+1}) \mid X_t = x] = \mathbb{E}[\langle f, \phi(X_{t+1})\rangle_{\mathcal{H}}, \mid X_t = x] = \langle f, g_p(x)\rangle_{\mathcal{H}}. \tag{29}$$

The CME of the Markov transition kernel $p$ is therefore just the Riesz representation of the functional evaluating the Koopman operator restricted to $\mathcal{H}$.

We now prove the identity (10) in the main text.

**Proposition 6.** *For every $G \in \mathrm{HS}(\mathcal{H})$ the risk (3) can be equivalently written as*

$$\underbrace{\mathbb{E}_{(x,y)\sim\rho}\|\phi(y) - G^*\phi(x)\|^2}_{\mathcal{R}(G)} = \underbrace{\mathbb{E}_{(x,y)\sim\rho}\|g_p(x) - \phi(y)\|^2}_{\mathcal{R}_0} + \underbrace{\mathbb{E}_{x\sim\pi}\|g_p(x) - G^*\phi(x)\|^2}_{\mathcal{E}(G)}. \tag{30}$$

*Proof.* Starting from (3) and using the reproducing property we obtain

$$\mathcal{R}(G) = \sum_{i\in\mathbb{N}} \mathbb{E}_{(x,y)\sim\rho}[h_i(y) - (Gh_i)(x)]^2 = \sum_{i\in\mathbb{N}} \mathbb{E}_{(x,y)\sim\rho}[\langle h_i, \phi(y)\rangle_{\mathcal{H}} - \langle Gh_i, \phi(x)\rangle_{\mathcal{H}}]^2$$

$$= \sum_{i\in\mathbb{N}} \mathbb{E}_{(x,y)\sim\rho}[\langle h_i, \phi(y)\rangle_{\mathcal{H}} - \langle h_i, G^*\phi(x)\rangle_{\mathcal{H}}]^2 = \sum_{i\in\mathbb{N}} \mathbb{E}_{(x,y)\sim\rho}\langle h_i, \phi(y) - G^*\phi(x)\rangle_{\mathcal{H}}^2$$

$$= \mathbb{E}_{(x,y)\sim\rho} \sum_{i\in\mathbb{N}} \langle h_i, \phi(y) - G^*\phi(x)\rangle_{\mathcal{H}}^2 = \mathbb{E}_{(x,y)\sim\rho}\|\phi(y) - G^*\phi(x)\|_{\mathcal{H}}^2.$$

By the reproducing properties of $(Z_\pi h_i)(x) = \langle h_i, g_p(x)\rangle_{\mathcal{H}}$ and $(S_\pi h_i)(x) = \langle h_i, \phi(x)\rangle_{\mathcal{H}}$ and Proposition 4 we have that

$$\mathcal{E}(G) = \|Z_\pi - S_\pi G\|_{\mathrm{HS}}^2 = \sum_{i\in\mathbb{N}} \|Z_\pi h_i - S_\pi Gh_i\|^2 = \sum_{i\in\mathbb{N}} \mathbb{E}_{x\sim\pi}[|(Z_\pi h_i)(x) - (S_\pi Gh_i)(x)|^2]$$

$$= \mathbb{E}_{x\sim\pi}\Big[\sum_{i\in\mathbb{N}} \langle h_i, g_p(x) - G^*\phi(x)\rangle_{\mathcal{H}}^2\Big] = \mathbb{E}_{x\sim\pi}\Big[\|g_p(x) - G^*\phi(x)\|^2\Big].$$

Moreover, since $S_\pi^* S_\pi = \mathbb{E}_{y\sim\pi}[\phi(y) \otimes \phi(y)]$ and $Z_\pi^* Z_\pi = \mathbb{E}_{x\sim\pi}[g_p(x) \otimes g_p(x)]$, we have

$$\mathbb{E}_{(x,y)\sim\rho}\|\phi(y) - g_p(x)\|^2 = \mathrm{tr}(S_\pi^* S_\pi) - 2\mathbb{E}_{(x,y)\sim\rho}\langle\phi(y), g_p(x)\rangle_{\mathcal{H}} + \mathrm{tr}(Z_\pi^* Z_\pi),$$

which, along with Prop. 4 and the identity $\mathbb{E}_{(x,y)\sim\rho}\langle\phi(y), g_p(x)\rangle_{\mathcal{H}} = \mathbb{E}_{x\sim\pi}\mathbb{E}[\langle\phi(Y), g_p(x)\rangle_{\mathcal{H}} \mid X = x] = \mathbb{E}_{x\sim\pi}\langle g_p(x), g_p(x)\rangle_{\mathcal{H}} = \mathrm{tr}(Z_\pi^* Z_\pi)$ completes the proof. $\qquad\square$

Prop. 6 implies that $G_\star$ is a solution of the KOR problem (5) if and only if $G_\star^*$ is a solution of the CME regression problem $\min_G \mathbb{E}_{(x,y)\sim\rho}\|\phi(y) - G\phi(x)\|^2$. In this sense the Koopman regression problem is *dual* to learning CME of the Markov transition kernel $p$.

Moreover, from the perspective of CME, well-specified case is identified by $g_p(\cdot) = G_{\mathcal{H}}^*\phi(\cdot)$, i.e. it is the case when regression operator $g_p$ belongs to the vector-valued RKHS $\mathcal{G}$ defined by the operator-valued kernel $g(x, x') := k(x, x')I_{\mathcal{H}}$. This vector-valued RKHS is isometrically isomorphic to $\mathrm{HS}(\mathcal{H})$, where the isomorphism is given by $\mathrm{HS}(\mathcal{H}) \ni A \longleftrightarrow A\phi(\cdot) \in \mathcal{G}$, see [10, Ex.3.6(i)]. On the other hand, the misspecified case is simply when $g_p \notin \mathcal{G}$.

## C    Empirical Risk Minimization

In this section we provide details on computing the estimators of the Koopman operator. For convenience, we denote the regularized risk by

$$\widehat{\mathcal{R}}^\gamma(G) := \|\widehat{Z} - \widehat{S}G\|_{\mathrm{HS}}^2 + \gamma\|G\|_{\mathrm{HS}}^2, \qquad G \in \mathrm{HS}\,(\mathcal{H})\,. \tag{31}$$

### C.1    Computation of the Estimators

In Theorem 4 we derive the closed form solution of (13) and in Theorem 5 we formulate it in a numerically computable representation. In Theorem 6 we show the same for the PCR estimator, highlighting its equivalence to the kernel DMD algorithm [25].

**Theorem 4.** *The optimal solution of problem* (13) *is given by* $\widehat{G}_{r,\gamma} = \widehat{C}_\gamma^{-\frac{1}{2}}[\![\widehat{C}_\gamma^{-\frac{1}{2}}\widehat{T}]\!]_r$. *Moreover,* $\widehat{\mathcal{R}}^\gamma(\widehat{G}_{r,\gamma}) = \mathrm{tr}(\widehat{D}) - \sum_{i=1}^r \sigma_i^2$, *where* $\sigma_1 \geq \cdots \geq \sigma_r$ *are leading singular values of* $\widehat{C}_\gamma^{-\frac{1}{2}}\widehat{T}$.

*Proof.* Start by observing that, according to (11),

$$\widehat{\mathcal{R}}^\gamma(G) = \frac{1}{n}\sum_{i=1}^n \|\phi(y_i) - G^*\phi(x_i)\|^2 + \gamma\|G\|_{\mathrm{HS}}^2$$

$$= \frac{1}{n}\sum_{i=1}^n \mathrm{tr}(\phi(y_i) \otimes \phi(y_i)) - 2\langle\phi(y_i), G^*\phi(x_i)\rangle_\mathcal{H} + \mathrm{tr}(GG^*\phi(x_i) \otimes \phi(x_i)) + \gamma\,\mathrm{tr}(GG^*)$$

$$= \mathrm{tr}(\widehat{D}) + \mathrm{tr}(GG^*\widehat{C}_\gamma) - 2\,\mathrm{tr}(G^*\widehat{T}) = \mathrm{tr}(\widehat{D}) - \|\widehat{C}_\gamma^{-\frac{1}{2}}\widehat{T}\|_{\mathrm{HS}}^2 + \|\widehat{C}_\gamma^{\frac{1}{2}}G - \widehat{C}_\gamma^{-\frac{1}{2}}\widehat{T}\|_{\mathrm{HS}}^2.$$

The last equality follows from simple algebra after adding and subtracting the term $\|\widehat{C}_\gamma^{-\frac{1}{2}}\widehat{T}\|_{\mathrm{HS}}^2$.

We now focus on the last term of the previous equation, the only one entering the minimization. We have

$$\|[\![\widehat{C}_\gamma^{-\frac{1}{2}}\widehat{T}]\!]_r - \widehat{C}_\gamma^{-\frac{1}{2}}\widehat{T}\|_{\mathrm{HS}}^2 = \min_{B\in\mathrm{HS}_r(\mathcal{H})}\|B - \widehat{C}_\gamma^{-\frac{1}{2}}\widehat{T}\|_{\mathrm{HS}}^2 \leq \min_{G\in\mathrm{HS}_r(\mathcal{H})}\|\widehat{C}_\gamma^{\frac{1}{2}}G - \widehat{C}_\gamma^{-\frac{1}{2}}\widehat{T}\|_{\mathrm{HS}}^2. \tag{32}$$

The equality above comes from the Eckart–Young–Mirsky theorem, while the inequality from the fact that $G \in \mathrm{HS}_r(\mathcal{H}) \implies B := \widehat{C}_\gamma^{\frac{1}{2}}G \in \mathrm{HS}_r(\mathcal{H})$. From (32) we conclude that $\widehat{G}_{r,\gamma} = \widehat{C}_\gamma^{-\frac{1}{2}}[\![\widehat{C}_\gamma^{-\frac{1}{2}}\widehat{T}]\!]_r$ minimizes $\widehat{\mathcal{R}}^\gamma$. The same theorem also guarantees that $\|\widehat{C}_\gamma^{\frac{1}{2}}\widehat{G}_{r,\gamma} - \widehat{C}_\gamma^{-\frac{1}{2}}\widehat{T}\| = \sum_{i=r+1}^\infty \sigma_i^2$, hence

$$\widehat{\mathcal{R}}^\gamma(\widehat{G}_{r,\gamma}) = \mathrm{tr}(\widehat{D}) - \sum_{i=1}^\infty \sigma_i^2 + \sum_{i=r+1}^\infty \sigma_i^2 = \mathrm{tr}(\widehat{D}) - \sum_{i=1}^r \sigma_i^2.$$

$\square$

While the previous theorem provides a method to compute the RRR estimator when the RKHS is finite-dimensional (in fact, an efficient one when the number of features is smaller than the number of samples), the following result shows how one can compute RRR for infinite-dimensional RKHSs.

**Theorem 5.** *If* $U_r = [u_1\,|\dots|\,u_r] \in \mathbb{R}^{n\times r}$ *is such that* $(\sigma_i^2, u_i)$ *are the solutions of the generalized eigenvalue problem*

$$LKu_i = \sigma_i^2 K_\gamma u_i \quad \text{normalized such that} \quad u_i^\top KK_\gamma u_i = 1, \quad i \in [r] \tag{33}$$

*and* $V_r = KU_r$, *then the optimal solution of* (13) *is given by* $\widehat{G}_{r,\gamma} = \widehat{S}^*U_rV_r^\top\widehat{Z}$. *Moreover, we have*

$$\widehat{\mathcal{R}}^\gamma(\widehat{G}_{r,\gamma}) = \mathrm{tr}(L) - \sum_{i=1}^r \sigma_i^2 \quad \text{and} \quad \widehat{\mathcal{R}}(\widehat{G}_{r,\gamma}) = \mathrm{tr}\left(\left(I - KU_rV_r^\top - \gamma K(U_rV_r^\top)^2\right)L\right). \tag{34}$$

*Proof.* Start by observing that, according to Thm 4, $\widehat{G}_{r,\gamma}$ is obtained from the truncated SVD of the operator $(\widehat{C} + \gamma I_\mathcal{H})^{-\frac{1}{2}}\widehat{T} = (\widehat{S}^*\widehat{S} + \gamma I_\mathcal{H})^{-\frac{1}{2}}\widehat{S}^*\widehat{Z} = \widehat{S}^*(\widehat{S}\widehat{S}^* + \gamma I_\mathcal{H})^{-\frac{1}{2}}\widehat{Z} = \widehat{S}^*K_\gamma^{-\frac{1}{2}}\widehat{Z}$. Its

leading singular values $\sigma_1 \geq \ldots \geq \sigma_r$ and the corresponding *left* singular vectors $g_1, \ldots, g_r \in \mathcal{H}$ are obtained by solving the eigenvalue problem

$$\left(\widehat{S}^* K_\gamma^{-\frac{1}{2}} \widehat{Z}\right) \left(\widehat{S}^* K_\gamma^{-\frac{1}{2}} \widehat{Z}\right)^* g_i = \sigma_i^2 g_i, \ i \in [r]. \tag{35}$$

From the above equation, clearly $g_i \in \text{Im}(\widehat{S}^* K_\gamma^{-\frac{1}{2}}) = \text{Im}(\widehat{S}^* K_\gamma^{\frac{1}{2}})$, and we can represent the singular vectors as $g_i = \widehat{S}^* K_\gamma^{\frac{1}{2}} u_i$ for some $u_i \in \mathbb{R}^n$, $i \in [r]$. Therefore, substituting $g_i = \widehat{S}^* K_\gamma^{\frac{1}{2}} u_i$ in (35) and simplifying one has the finite-dimensional eigenvalue equation

$$L K u_i = \sigma_i^2 K_\gamma u_i, \ i \in [r]. \tag{36}$$

Solving (36) and using that $g_i = \widehat{S}^* K_\gamma^{\frac{1}{2}} u_i$, one obtains $(\sigma_i^2, g_i)$, $i \in [r]$, the solutions of the eigenvalue problem (35). In order to have properly normalized $g_i$, it must hold for all $i \in [r]$ that

$$1 = g_i^* g_i = u_i^\top K_\gamma^{\frac{1}{2}} \widehat{S} \widehat{S}^* K_\gamma^{\frac{1}{2}} u_i = u_i^\top K K_\gamma u_i. \tag{37}$$

Now, the subspace of the leading left singular vectors is $\text{Im}(\widehat{S}^* K_\gamma^{\frac{1}{2}} U_r)$. As the columns of $U_r$ are properly normalized according to (37), the orthogonal projector onto the range of $\widehat{S}^* K_\gamma^{\frac{1}{2}} U_r$ is given by $\Pi_r := \widehat{S}^* K_\gamma^{\frac{1}{2}} U_r U_r^\top K_\gamma^{\frac{1}{2}} \widehat{S}$. We therefore have that $[\![\widehat{C}_\gamma^{-\frac{1}{2}} \widehat{T}]\!]_r = \Pi_r \widehat{C}_\gamma^{-\frac{1}{2}} \widehat{T} = \widehat{S}^* K_\gamma^{\frac{1}{2}} U_r U_r^\top K \widehat{Z}$. Thus, defining $V_r := K U_r$, we conclude that

$$\widehat{G}_{r,\gamma} = \widehat{C}_\gamma^{-\frac{1}{2}} \widehat{S}^* K_\gamma^{\frac{1}{2}} U_r V_r^\top \widehat{Z} = \widehat{S}^* U_r V_r^\top \widehat{Z}.$$

To conclude the proof we have to evaluate the error $\widehat{\mathcal{R}}(\widehat{G}_{r,\gamma})$. We notice that $\text{tr}(\widehat{D}) = \text{tr}(L)$ and that

$$\widehat{\mathcal{R}}(\widehat{G}_{r,\gamma}) = \text{tr}(L) - 2\text{tr}(V_r V_r^\top L) + \text{tr}(V_r V_r^\top V_r V_r^\top L) = \text{tr}\left(\left(I - K U_r V_r^\top - \gamma K (U_r V_r^\top)^2\right) L\right).$$

Here, along with some simple algebric manipulations, we have used $U_r^\top K (K + \gamma I) U_r = I$, i.e. $V_r^\top V_r + \gamma V_r^\top U_r = I$. $\qquad\square$

**Remark 7.** *Since for $r = n$ the estimators RRR and KRR coincide, the previous result implies that the empirical risk for the KRR estimator can be written as*

$$\widehat{\mathcal{R}}(\widehat{G}_\gamma) = \text{tr}\left(\left(I - K K_\gamma^{-1} + \gamma K K_\gamma^{-2}\right) L\right) = \gamma^2 \text{tr}\left(K_\gamma^{-2} L\right). \tag{38}$$

As discussed above, see also [25], for the choice of linear kernel PCR estimator $\widehat{G}_r^{\text{PCR}} = [\![\widehat{C}]\!]_r^\dagger \widehat{T}$ is known as DMD, while for the finite-dimensional (nonlinear) kernels it is known as extended EDMD. In these cases previous formula gives also a practical way to compute it. On the other hand, for infinite-dimensional kernels, PCR is known as kernel DMD, and in this case its practical computation can be done using the following result.

**Theorem 6.** *The PCR estimator $\widehat{G}_r^{\text{PCR}} = [\![\widehat{C}]\!]_r^\dagger \widehat{T}$ can be equivalently written as $\widehat{G}_r^{\text{PCR}} = \widehat{S}^* U_r V_r^\top \widehat{Z}$, where $[\![K]\!]_r^\dagger = V_r \Sigma_r V_r^\top$ is $r$-trunacted SVD and $U_r = V_r \Sigma_r^\dagger$. Moreover, it holds that*

$$\widehat{\mathcal{R}}(\widehat{G}_r^{\text{PCR}}) = \text{tr}((I_n - K U_r V_r^\top) L). \tag{39}$$

*Proof.* Without loss of generality assume that $\text{rank}(\widehat{C}) \geq r$. As for the proof of Theorem 5, the leading $r$ singular vectors $(g_i)_{i \in [r]}$ of $\widehat{C}$ can be written in the form $g_i = \widehat{S}^* v_i / \sqrt{\sigma_i}$, where $v_i$ are the eigenvectors corresponding to the $r$ leading eigenvalues $\sigma_i$ of $K$. Then, it readily follows that $[\![\widehat{C}]\!]_r^\dagger \widehat{T} = \widehat{S}^* V_r \Sigma_r^{-2} V_r^\top \widehat{S} \widehat{S}^* \widehat{Z}$. We therefore conclude that

$$\widehat{G}_r^{\text{PCR}} = \widehat{S}^* V_r \Sigma_r^{-2} (K V_r)^\top \widehat{Z} = \widehat{S}^* V_r \Sigma_r^{-1} V_r^\top \widehat{Z} = \widehat{S}^* U_r V_r^\top \widehat{Z}$$

Finally, since $\widehat{\mathcal{R}}(\widehat{G}_r^{\text{PCR}}) = \text{tr}(L) - 2\text{tr}(V_r V_r^\top L) + \text{tr}(V_r V_r^\top V_r V_r^\top L)$, using that $V_r V_r^\top$ is orthogonal projector and $K U_r = V_r$ we obtain (39). $\qquad\square$

## C.2 Mode Decomposition and Prediction

In this section we show how an estimator of the Koopman operator can be used to predict future states of the system and how its mode decomposition can be evaluated. We will address a slightly more general setting than the one presented in Sec. 4, that is we allow for *vectorial* observables $f = (f_\ell)_{\ell=1}^m \in \mathcal{H}^m$ for which the action of the Koopman operator is naturally extended as $A_\pi f = (A_\pi f_\ell)_{\ell \in [m]}$. To that end, given the data $\mathcal{D}$ and a vector valued observable $f = (f_\ell)_{\ell=1}^m \in \mathcal{H}^m$ we denote the observable evaluated along the data points as $\Gamma^f = \begin{bmatrix} f(y_1) \mid \ldots \mid f(y_n) \end{bmatrix} \in \mathbb{R}^{m \times n}$.

**Remark 8.** *If $\mathcal{X} \subseteq \mathbb{R}^d$, we will argue that an important observable of the system is given by the identity function $\mathrm{Id} : \mathcal{X} \to \mathcal{X}$. If the projection onto the $i$-th component is a function belonging to $\mathcal{H}$ for all $i \in [d]$, then $\mathrm{Id} \in \mathcal{H}^d$ and Thm. 1 holds for this specific observable. While this may not hold in general, note that for every kernel we can take the sum with a linear kernel to obtain RKHS that contains $\mathrm{Id}$.*

**Prediction.** Each empirical estimator $\widehat{G} = \widehat{S}^* W \widehat{Z}$ allows one to estimate a future state given a starting point. According to the bound (7) in Thm. 1 we obtain that given $f = (f_\ell)_{\ell=1}^m \in \mathcal{H}^m$, if $x$ is the current state of the Markov process, the expected value of $f$ at the next iteration is approximated as $[A_\pi f](x) = [\widehat{G} f](x) + \mathrm{err}^f(x)$, i.e.

$$\mathbb{E}[f(X_{t+1}) \mid X_t = x] = \sum_{j=1}^n \beta_j^f k(x_j, x) + \sqrt{\mathcal{E}(\widehat{G})} \, \mathrm{err}^f(x), \tag{40}$$

where $\beta^f = \frac{1}{n} \Gamma^f W^\top \in \mathbb{R}^{m \times n}$ and $\mathrm{err}^f \in (L_\pi^2(\mathcal{X}))^m$ such that $\|(\mathrm{err}^f)_\ell\| \leq \|f_\ell\|$, $\ell \in [m]$. If $f = \mathrm{Id}$, we obtain a prediction of the future state $\mathbb{E}[X_{t+1} \mid X_t = x]$.

**Modal Decomposition & Forecasting.** A more general instance of prediction is given by forecasting through modal decomposition as showed (7). The main ingredient needed to forecast via mode decomposition is the spectral decomposition of the estimator $\widehat{G}$. We now prove a slightly more general version of Theorem 2, allowing us to compute the eigenvalue decomposition of $\widehat{G}$ numerically.

**Theorem 7.** *Let $\widehat{G} = \widehat{S}^* U_r V_r^\top \widehat{Z}$, with $U_r, V_r \in \mathbb{R}^{n \times r}$. If $V_r^\top M U_r \in \mathbb{R}^{r \times r}$ is full rank and non-defective, the spectral decomposition $(\lambda_i, \xi_i, \psi_i)_{i \in [r]}$ of $\widehat{G}$ can be expressed in terms of the spectral decomposition $(\lambda_i, \widetilde{u}_i, \widetilde{v}_i)_{i \in [r]}$ of $V_r^\top M U_r$. Indeed, for all $i \in [r]$, one has $\xi_i = \widehat{Z}^* V_r \widetilde{u}_i / \overline{\lambda}_i$ and $\psi_i = \widehat{S}^* U_r \widetilde{v}_i$. In addition, for every $f \in \mathcal{H}^m$ dynamic modes are $\gamma_i^f = \Gamma^f (\widetilde{u}_i^* V_r^\top)^\top / (\lambda_i \sqrt{n}) \in \mathbb{C}^m$.*

*Proof.* First note that since in general $\widehat{G}$ is not self-adjoint, its eigenvalues may come in complex conjugate pairs. Hence, for $\xi_i \in \mathcal{H}$ and $\psi_i \in \mathcal{H}$ left and right eigenfunctions of $\widehat{G}$ corresponding to its eigenvalue $\lambda_i$, we have $\widehat{G}^* \xi_i = \overline{\lambda}_i \xi_i$ and $\widehat{G} \psi_i = \lambda_i \psi_i$, $i \in [r]$. To avoid cluttering, in the following we will only show the explicit calculation of the right eigenfunctions $\psi_i$. We stress, however, that the calculation of the left eigenfucntions $\xi_i$ follows exactly the same arguments.

From $\mathrm{Im}(\widehat{G}) \subseteq \mathrm{Im}(\widehat{S}^*)$ it follows that for all $i \in [r]$, $\psi_i \in \mathcal{S} := \{ \sum_{j=1}^n w_j \phi(x_j) \mid w \in \mathbb{C}^n \}$. Using [42, Prop. 3.8], we have that $\psi_i = \widehat{S}^* \widehat{v}_i$, where $\widehat{v}_i \in \mathbb{C}^n \setminus \{0\}$ are eigenvectors of $U_r V_r^\top M$. Since all the eigenvalues $\lambda_i$ we are considering are nonzero, the spectral decomposition of $U_r V_r^\top M$ is equivalent [57] to

$$V_r^\top M U_r \widetilde{v}_i = \lambda_i \widetilde{v}_i \quad \text{and} \quad \widehat{v}_i = U_r \widetilde{v}_i, \quad i \in [r]. \tag{41}$$

Therefore $\psi_i = \widehat{S}^* U_r \widetilde{v}_i$ are the right eigenfunctions of $\widehat{G} = \widehat{S}^* U_r V_r^\top \widehat{Z}$. With the same arguments we can show that $\xi_i = \widehat{Z}^* V_r \widetilde{u}_i$ are the left eigenfunctions, $\widetilde{u}_i$ being the leading eigenvectors of $U_r^\top M^\top V_r$. Re-normalizing $\xi_j = \widehat{Z}^* V_r \widetilde{u}_j / \overline{\lambda}_j$ we obtain that for every $i, j \in [r]$

$$\langle \psi_i, \overline{\xi}_j \rangle_{\mathcal{H}} = \widetilde{u}_j^* V_r^\top \widehat{Z} \widehat{S}^* U_r \widetilde{v}_i / \lambda_j = \widetilde{u}_j^* V_r^\top M U_r \widetilde{v}_i / \lambda_j = (U_r^\top M^\top V_r \widetilde{u}_j)^* \widetilde{v}_i / \lambda_j = \widetilde{u}_j^* \widetilde{v}_i = \delta_{ij},$$

which assures that $(\lambda_i, \xi_i, \psi_i)_{i \in [r]}$ is the spectral decomposition of $\widehat{G}$. Above we have assumed (without loss of generality) that $\widetilde{u}_j^* \widetilde{v}_i = \delta_{ij}$, $i, j \in [r]$, i.e. that the left and right eigenvectors of $V_r^\top M U_r$ are mutually orthonormal.

Finally, since $\widehat{G} = \sum_{i \in [r]} \lambda_i \psi_i \otimes \overline{\xi}_i$ we have that $\widehat{G} f_\ell = \sum_{i \in [r]} \lambda_i \psi_i \langle f_\ell, \overline{\xi}_i \rangle_{\mathcal{H}}$ and, consequently

$$\gamma_i^f = (\langle f_\ell, \overline{\xi}_i \rangle_{\mathcal{H}})_{\ell \in [m]} = (\langle \widetilde{u}_i^* V_r^\top Z f_\ell \rangle_{\mathcal{H}} / \lambda_i)_{\ell \in [m]} = \Gamma^f (\widetilde{u}_i^* V_r^\top)^\top / (\lambda_i \sqrt{n}) \in \mathbb{C}^m.$$

$\square$

**Remark 9.** *The previous result can also be applied to KRR estimator $\widehat{G}_\gamma$ since we can always take $r = n$, and take $U_r = I_n$ and $V_r = K_\gamma^{-1}$ to represent $K_\gamma^{-1} = U_r V_r^\top$. As a consequence, we can compute spectral decomposition of $\widehat{G}_\gamma$ by solving a generalized eigenvalue problem*

$$M^\top \widetilde{v}_i = \lambda_i K_\gamma \widetilde{v}_i, \quad i \in [n], \tag{42}$$

*and setting $\widetilde{U} := \widetilde{V}^{-*}$. This is possible when $K_\gamma^{-1} M^\top$ is non-defective matrix, which is typically the case for kernel Gram matrices from real data.*

## D   Learning Bounds

### D.1   Uniform Bounds for i.i.d. Data

We first present a concentration inequality for bounded finite rank self-adjoint operators, which is a natural extension of [34, Theorem 4], that dealt with positive operators.

**Proposition 7.** *Let $A_1, \ldots, A_n$ be independent random operators of finite rank $\tau$ and $\|A_i\| \leq 1$, $i \in [n]$. Then*

$$\mathbb{P}\left\{ \left\| \sum_{i=1}^n A_i - \mathbb{E} A_i \right\| > s \right\} \leq 8(n\tau)^2 \exp\left\{ \frac{-s^2}{36\| \sum_i \mathbb{E}(A_i^* A_i)^{\frac{1}{2}} \| + 12s} \right\}. \tag{43}$$

*Proof.* Let $B_i = \begin{bmatrix} 0 & A_i \\ A_i^* & 0 \end{bmatrix} = P_i - N_i$, where

$$P_i = \frac{1}{2} \begin{bmatrix} (A_i A_i^*)^{\frac{1}{2}} & A_i \\ A_i^* & (A_i^* A_i)^{\frac{1}{2}} \end{bmatrix} \quad \text{and} \quad N_i = \frac{1}{2} \begin{bmatrix} (A_i A_i^*)^{\frac{1}{2}} & -A_i \\ -A_i^* & (A_i^* A_i)^{\frac{1}{2}} \end{bmatrix}.$$

One verifies that the operators $P_i$ and $N_i$ are positive semi-definite, have the same rank as $A_i$, and $\|P_i\| = \|N_i\| = \|A_i\|$. Then

$$\mathbb{P}\left\{ \left\| \sum_{i=1}^n A_i - \mathbb{E} A_i \right\| > s \right\} = \mathbb{P}\left\{ \left\| \sum_{i=1}^n B_i - \mathbb{E} B_i \right\| > s \right\}$$

$$\leq \mathbb{P}\left\{ \left\| \sum_{i=1}^n P_i - \mathbb{E} P_i \right\| + \left\| \sum_{i=1}^n N_i - \mathbb{E} N_i \right\| > s \right\}$$

$$\leq \mathbb{P}\left\{ \left\| \sum_{i=1}^n P_i - \mathbb{E} P_i \right\| > \frac{s}{2} \right\} + \mathbb{P}\left\{ \left\| \sum_{i=1}^n N_i - \mathbb{E} N_i \right\| > \frac{s}{2} \right\}$$

$$\leq 8(n\tau)^2 \exp\left\{ \frac{-s^2}{36 \max\{\| \sum_i \mathbb{E} P_i \|, \| \sum_i \mathbb{E} N_i \|\} + 12s} \right\}$$

where the first inequality follows by triangle inequality, the second by the union bound and the last from [34, Thm. 7-(i)]. The result follows by noting that $\| \sum_i \mathbb{E} P_i \| = \| \sum_i \mathbb{E} N_i \| \leq \| \sum_i \mathbb{E}(A_i^* A_i)^{\frac{1}{2}} \|$. $\square$

A special case of the above proposition is Prop. 2 which we restate here for the reader's convenience.

**Proposition 2.** *With probability at least $1 - \delta$ in the i.i.d. draw of $(x_i, y_i)_{i=1}^n$ from $\rho$,*

$$\left\| \widehat{T} - T \right\| \leq 12 \frac{\ln \frac{8n^2}{\delta}}{n} + 6 \sqrt{\frac{\|C\| \ln \frac{8n^2}{\delta}}{n}}.$$

*Proof.* We apply Prop. 7 with $A_i = \phi(x_i) \otimes \phi(y_i)$ and $\tau = 1$. We have

$$P_i = \frac{1}{2} \left[ \begin{array}{cc} \phi(x_i) \otimes \phi(x_i) & \phi(x_i) \otimes \phi(y_i) \\ \phi(y_i) \otimes \phi(x_i) & \phi(y_i) \otimes \phi(y_i) \end{array} \right] \text{ and } N_i = \frac{1}{2} \left[ \begin{array}{cc} \phi(x_i) \otimes \phi(x_i) & -\phi(x_i) \otimes \phi(y_i) \\ -\phi(y_i) \otimes \phi(x_i) & \phi(y_i) \otimes \phi(y_i) \end{array} \right].$$

Since $(x_1, y_1), \ldots, (x_n, y_n)$ are i.i.d. from $\rho$, for every $i \in [n]$

$$\mathbb{E}P_i = \frac{1}{2} \left[ \begin{array}{cc} C & T \\ T^* & D \end{array} \right] \text{ and } \mathbb{E}N_i = \frac{1}{2} \left[ \begin{array}{cc} C & -T \\ -T^* & D \end{array} \right] \tag{44}$$

Thus $\|\mathbb{E}P_i\| = \|\mathbb{E}N_i\| \leq \max\{\|C\|, \|D\|\} = \|C\|$, where the last equality is due to $\pi$ being invariant measure and, hence, $D = C$. Then setting the r.h.s. of (43) equal to $\delta$ and solving for $s$ gives Prop. 2. $\square$

**Theorem 3.** *Let $\mathcal{G}_{r,\gamma} = \{G \in \mathrm{HS}_r(\mathcal{H}) : \|G\|_{\mathrm{HS}} \leq \gamma\}$ and define $\sigma^2 = \mathbb{E}(\|\phi(y)\|^2 - \mathbb{E}\|\phi(y)\|^2)^2$. With probability at least $1 - \delta$ in the i.i.d. draw of $(x_i, y_i)_{i=1}^n$ from $\rho$, we have for every $G \in \mathcal{G}_{\gamma,r}$*

$$|\mathcal{R}(G) - \widehat{\mathcal{R}}(G)| \leq \sqrt{\frac{2\sigma^2 \ln\frac{6}{\delta}}{n}} + 3(4\sqrt{2r}\gamma + \gamma^2)\sqrt{\frac{\|C\| \ln\frac{24n^2}{\delta}}{n}} + \frac{(1 + 24\gamma\sqrt{r})\ln\frac{6}{\delta} + 6\gamma^2 \ln\frac{24n^2}{\delta}}{n}.$$

*Proof.* Recalling the definition of the risk $\mathcal{R}(G) = \mathrm{tr}\left[D\right] + \mathrm{tr}\left[GG^*C\right] - 2\,\mathrm{tr}\left[G^*T\right]$, and, analogously, empirical risk, a direct computation gives that

$$\mathcal{R}(G) - \widehat{\mathcal{R}}(G) = \mathrm{tr}\left(D - \widehat{D}\right) + \mathrm{tr}\left(GG^*(C - \hat{C})\right) - 2\,\mathrm{tr}\left(G^*(T - \hat{T})\right)$$
$$\leq \mathrm{tr}\left(D - \widehat{D}\right) + \gamma^2 \|C - \widehat{C}\| + 2\sqrt{r}\gamma\|T - \widehat{T}\|, \tag{45}$$

where we have used Hölder inequality in to obtain the last two terms in (45). First, we use Bernstein's inequality for bounded random variables [62, Thm 2.8.4] to bound the first term in the r.h.s. of (45), obtaining

$$\mathrm{tr}(D - \widehat{D}) \leq \frac{\ln\frac{2}{\delta}}{3n} + \sqrt{\frac{2\sigma^2 \ln\frac{2}{\delta}}{n}}.$$

Then we use [34, Theorem 7-(i)] to bound the second term in the r.h.s. of (45), and Prop. 2 to bound the last term. The result then follows by a union bound. $\square$

We expand some of the remarks stated after Thm. 3 in the main body of the paper. First, note that when the measure $\pi$ is not assumed to be invariant, we can cover the general CME case. In that case the term $\|C\|$ in the bound should be replaced by $\max\{\|C\|, \|D\|\}$, where, recall, $D$ is the covariance of the output. Second, using [40, Cor. 3.1] in place of Prop. 2 one can derive a related bound which essentially replaces the term $\|C\|$ with $\|\mathbb{E}AA^*\|$ where $A := (\phi(x) \otimes \phi(y) - T)$. This bound is more difficult to turn into a data dependent bound, but it allows for a more direct comparison to (potentially much larger) bounds without the rank constraint, where the quantity $\|\mathbb{E}AA^*\|$ is replaced by the potentially much larger term $\mathrm{tr}\,\mathbb{E}[\phi(x) \otimes \phi(x)\|\phi(y)\|^2 - T^*T]$.

Finally, Thm. 3 can be used to derive an excess risk bound in well specified case $Z_\pi = S_\pi G_{\mathcal{H}}$. The analysis follows the pattern in [29]. We use the decomposition

$$\mathcal{E}(\widehat{G}) \leq 2 \sup_{G \in \mathcal{G}_{r,\gamma}} |\mathcal{R}(G) - \widehat{\mathcal{R}}(G)| + \mathcal{E}(G_{r,\gamma}) \tag{46}$$

where $G_{r,\gamma} = \mathrm{argmin}_{G \in \mathcal{G}_{r,\gamma}} \|S_\pi(G_{\mathcal{H}} - G)\|_{\mathrm{HS}}^2$. We bound the first term in the r.h.s. of (46) by Thm. 3. The second term is the approximation error of $G_{\mathcal{H}}$ in the class $\mathcal{G}_{r,\gamma}$. We next optimize over $\gamma$. A natural choice is $\gamma = \|[\![G_{\mathcal{H}}]\!]_r\|_{\mathrm{HS}}$ so that $G_{r,\gamma} = [\![G_{\mathcal{H}}]\!]_r$, the truncated rank $r$ SVD of $G_{\mathcal{H}}$. For this choice the approximation error $\mathcal{R}(G_{\gamma,r})$ is $\epsilon_r := \|S_\pi(G_{\mathcal{H}} - [\![G_{\mathcal{H}}]\!]_r)\|_{\mathrm{HS}}^2$. If $G_{\mathcal{H}}$ has a fast decaying spectrum this error will be small for moderate sizes of $r$. In general since $G_{\mathcal{H}}$ is Hilbert-Schmidt $\epsilon_r \to 0$ as $r \to \infty$. Replacing $\gamma = \|[\![G_{\mathcal{H}}]\!]_r\|_{\mathrm{HS}}$ in the uniform bound, then yields the excess risk bound (discarding $O(1/n)$ terms and simplifying the constants)

$$\mathcal{E}(\widehat{G}) \leq 3\|[\![G_{\mathcal{H}}]\!]_r\|_{\mathrm{HS}}\left(6\sqrt{r} + \|[\![G_{\mathcal{H}}]\!]_r\|_{\mathrm{HS}}\right)\sqrt{\frac{\|C\| \ln\frac{24n^2}{\delta}}{n}} + \sqrt{\frac{2\sigma^2 \ln\frac{6}{\delta}}{n}} + \|G_{\mathcal{H}} - [\![G_{\mathcal{H}}]\!]_r\|_{\mathrm{HS}}^2.$$

This bound may be further optimized over $r$ if information on the spectrum decay of $G_{\mathcal{H}}$ is available.

## D.2 Uniform Bounds for Data from a Trajectory

To prove Lem. 1 we temporarily introduce extra notation. For a set $I \subseteq \mathbb{N}$ and a strictly stationary process $\mathbf{X} = (X_i)_{i \in \mathbb{N}}$ we let $\Sigma_I$ for the $\sigma$-algebra generated by $\{X_i\}_{i \in I}$ and $\mu_I$ for the joint distribution of $\{X_i\}_{i \in I}$. Notice that $\mu_{I+i} = \mu_I$. In this notation $\pi = \mu_{\{1\}}$ and $\rho_\tau = \mu_{\{1,1+\tau\}}$.

Then the definition of the mixing coefficients reads

$$\beta_{\mathbf{X}}(\tau) = \sup_{B \in \Sigma \otimes \Sigma} \left| \mu_{\{1,1+\tau\}}(B) - \mu_{\{1\}} \times \mu_{\{1\}}(B) \right|$$

which by the Markov property is equivalent to

$$\beta_{\mathbf{X}}(\tau) = \sup_{B \in \Sigma^I \otimes \Sigma^J} \left| \mu_{I \cup J}(B) - \mu_I \times \mu_J(B) \right|,$$

where $I, J \subset \mathbb{N}$ with $j > i + \tau$ for all $i \in I$ and $j \in J$. The latter is the definition of the mixing coefficients for general strictly stationary processes, for which we prove Lemma 1. We first need the following lemma.

**Lemma 3.** *Let* $B \in \Sigma_{[1:m]}$. *Then*

$$\left| \mu_{[1:m]}(B) - \mu_{\{1\}}^m(B) \right| \leq (m-1) \beta_{\mathbf{X}}(1).$$

*Proof.* By stationarity, Fubini's Theorem and the definition of the mixing coefficients, we have for $k \in [m]$, that

$$\left| \mu_{\{1\}}^{k-1} \times \mu_{[k:m]}(B) - \mu_{\{1\}}^{k-1} \times \mu_{\{1\}} \times \mu_{[k+1:m]}(B) \right| \leq \beta_{\mathbf{X}}(1).$$

Then, again with stationarity and a telescopic expansion,

$$\left| \mu_{[1:m]}(B) - \mu_{\{1\}}^m(B) \right| = \left| \sum_{k=1}^{m-1} \left( \mu_{\{1\}}^{k-1} \times \mu_{[k:m]}(B) - \mu_{\{1\}}^{k-1} \times \mu_{\{1\}} \times \mu_{[k+1:m]}(B) \right) \right|$$

$$\leq \sum_{k=1}^{m-1} \left| \mu_{\{1\}}^{k-1} \times \mu_{[k:m]}(B) - \mu_{\{1\}}^{k-1} \times \mu_{\{1\}} \times \mu_{[k+1:m]}(B) \right|$$

$$\leq (m-1) \beta_{\mathbf{X}}(1).$$

$\square$

Now recall the definition of the blocked variables

$$Y_j = \sum_{i=2(j-1)\tau+1}^{(2j-1)\tau} X_i \qquad \text{and} \qquad Y_j' = \sum_{i=(2j-1)\tau+1}^{2j\tau} X_i, \quad \text{for } j \in \mathbb{N}.$$

Since the blocked variables are separated by $\tau$ we have $\beta_{\mathbf{Y}}(1) = \beta_{\mathbf{Y}'}(1) = \beta_{\mathbf{X}}(\tau)$.

**Lemma 1.** *Let* $\mathbf{X}$ *be strictly stationary with values in a normed space* $(\mathcal{X}, \|\cdot\|)$, *and assume* $n = 2m\tau$ *for* $\tau, m \in \mathbb{N}$. *Moreover, let* $Z_1, \ldots, Z_m$ *be* $m$ *independent copies of* $Z_1 = \sum_{i=1}^{\tau} X_i$. *Then for* $s > 0$

$$\mathbb{P}\left\{ \left\| \sum_{i=1}^n X_i \right\| > s \right\} \leq 2 \mathbb{P}\left\{ \left\| \sum_{j=1}^m Z_j \right\| > \frac{s}{2} \right\} + 2(m-1) \beta_{\mathbf{X}}(\tau).$$

*Proof.* We can write

$$\left\| \sum_{i=1}^n X_i \right\| = \left\| \sum_{j=1}^m Y_j + \sum_{j=1}^m Y_j' \right\| \leq \left\| \sum_{j=1}^m Y_j \right\| + \left\| \sum_{j=1}^m Y_j' \right\|.$$

Thus

$$\Pr\left\{ \left\| \sum_{i=1}^n X_i \right\| > s \right\} \leq \Pr\left\{ \left\| \sum_{j=1}^m Y_j \right\| + \left\| \sum_{j=1}^m Y_j' \right\| > s \right\} \leq 2 \Pr\left\{ \left\| \sum_{j=1}^m Y_j \right\| > \frac{s}{2} \right\},$$

where the last inequality follows from identical distribution of $Y_j$ and $Y_{j+1}$. The conclusion then follows from applying Lem. 3 to the event $B = \left\| \sum_{j=1}^m Y_j \right\| > \frac{s}{2}$.

$\square$

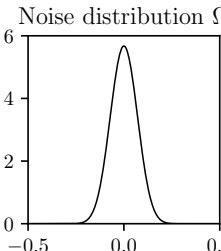 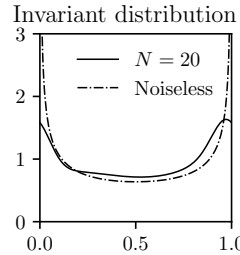 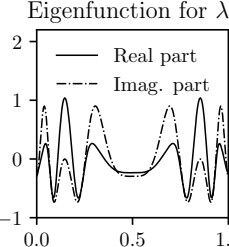

Figure 3: Noise distribution $\Omega$, invariant distribution $\pi$ and Koopman eigenfunction corresponding to the eigenvalue $\lambda_2$ for the case $N = 20$. In the middle panel, the invariant distribution for the noiseless case ($N \to \infty$) is [26] $\pi_{N \to \infty}(dx) := \left(\pi^2 x(1-x)\right)^{-1/2} dx$.

Any available bound on the probability in the right hand side of Lem. 1 can then be substituted to give a bound on the trajectory. To illustrate this we give a proof of Prop. 3 which we restate here for convenience.

**Proposition 3.** *Let $\delta > (m-1)\beta_{\mathbf{X}}(\tau - 1)$. With probability at least $1 - \delta$ in the draw $x_1 \sim \pi, x_i \sim p(x_{i-1}, \cdot)$, $i \in [2{:}n]$,*

$$\|\widehat{T} - T\| \leq \frac{48}{m} \ln \frac{4m\tau}{\delta - (m-1)\beta_{\mathbf{X}}(\tau - 1)} + 12\sqrt{\frac{2\|C\|}{m} \ln \frac{4m\tau}{\delta - (m-1)\beta_{\mathbf{X}}(\tau - 1)}}.$$

*Proof.* We use this Lem. 1 with $X_i = \phi(x_i) \otimes \phi(x_{i+1}) - T$. We have

$$\mathbb{P}\left\{\|\hat{T} - T\| > s\right\} = \mathbb{P}\left\{\left\|\sum_{i=1}^{n} X_i\right\| > ns\right\} \leq 2\mathbb{P}\left\{\left\|\sum_{j=1}^{m} Z_j\right\| > \frac{ns}{2}\right\} + 2(m-1)\beta_{\mathbf{X}}(\tau - 1).$$

To bound the rightmost probability we then use Prop. 7 with $A_i$ i.i.d. operator $\frac{1}{\tau}\sum_{i=1}^{\tau}\phi(x_i)\otimes\phi(x_{i+1})$ and $\mathbb{E}(A_i^* A_i)^{\frac{1}{2}} \leq \|C\|$. We then solve for $\delta > (m-1)\beta(\tau - 1)$. $\qquad\square$

# E Experiments

We developed a Python module implementing different algorithms to perform KOR Both CPUs and GPUs are supported. Code and experiments can be found at https://github.com/CSML-IIT-UCL/kooplearn. The experiments have been conducted on a workstation equipped with an Intel(R) Core™ i9-9900X CPU @ 3.50GHz, 48GB of RAM and a NVIDIA GeForce RTX 2080 Ti GPU.

## E.1 Noisy Logistic Map

We now show how the *trigonometric* noise introduced in [45] allows the evaluation of the *true* invariant distribution, transition kernel and Koopman eigenvalues.

**Trigonometric Noise.** We consider the *noisy logistic map*

$$x_{t+1} = (4x_t(1-x_t) + \xi_t) \mod 1 = (F(x_t) + \xi_t) \mod 1$$

over the state space $\mathcal{X} = [0,1]$. We have defined the logistic map $F(x) := 4x(1-x)$ for later convenience. Here, $\xi_t$ is i.i.d. additive noise with law ($N$ being an *even* integer) given by

$$\Omega(d\xi) := C_N \cos^N(\pi\xi)d\xi \qquad \xi \in [-0.5, 0.5].$$

The normalization constant is given by $C_N := \pi/\mathrm{B}\left(\frac{N+1}{2}, \frac{1}{2}\right)$, where $\mathrm{B}(\cdot, \cdot)$ is Euler's beta function. The noise is additive and as noted in Example 1 of Sec. 2, the transition kernel is

$$p(x, dy) = \Omega(dy - F(x)) = C_N \cos^N(\pi y - \pi F(x)) \, dy. \tag{47}$$

We now show that the transition kernel (47) is *separable*. Indeed, for $i \in [0{:}N]$ let us define the functions

$$\beta_i(x) := \sqrt{C_N \binom{N}{i}} \cos^i(\pi x) \sin^{N-i}(\pi x), \qquad \text{and} \quad \alpha_i(x) := (\beta_i \circ F)(x).$$

By a simple application of the binomial theorem one has that (with a slight abuse of notation)

$$p(x, y) = \sum_{i=0}^{N} \alpha_i(x) \beta_i(y),$$

implying that the transition kernel is separable and of finite rank $N + 1$. Therefore, the Koopman operator $A_\pi$ is compact operator of a finite rank operator at most $N + 1$. Moreover, with the proper choice of the kernel $k$, we have that $\alpha_i \in \mathcal{H}$ for all $i \in [0{:}N]$, implying that the Koopman operator regression problem is well-specified.

Further, the eigenvalue equation for the Koopman operator requires to find $h : [0, 1] \to [0, 1]$ and $\lambda \in \mathbb{C}$ satisfying

$$\lambda h(x) = \int_0^1 h(y) p(x, y) dy \qquad \text{for all } x \in [0, 1]. \tag{48}$$

The solution of this *homogeneous Fredholm integral equation of the second kind* (48) is easily obtained since the transition kernel is separable (see e.g. Section 23.4 of [50]). Indeed, let $P$ be the $(N + 1) \times (N + 1)$ matrix whose elements are $P_{ij} := \int_0^1 \beta_i(x) \alpha_j(x) dx$. For any $\lambda$ eigenvalue of $P$ with corresponding eigenvector $(c_i)_{i=0}^N$, the function $h(x) := \sum_{i=0}^N \alpha_i(x) c_i$ is an eigenfunction of the Koopman operator with eigenvalue $\lambda$. With a similar argument, let $(d_i)_{i=0}^N$ be the eigenvector of $P^T$ corresponding to the eigenvalue $\lambda = 1$. The invariant distribution (up to a normalization constant) $\pi$ is given by

$$\pi(x) dx = \left( \sum_{i=0}^N \beta_i(x) d_i \right) dx.$$

Every result presented in the main text concerned the case $N = 20$. In Fig. 3 we show the noise distribution $\Omega$, invariant distribution $\pi$ and Koopman eigenfunction corresponding to the eigenvalue $\lambda_2$ for the case $N = 20$.

### E.2 Additional experiment: the Lorenz63 Dynamical System

The Lorenz63 system is given by the solution the differential equation

$$\frac{d\boldsymbol{x}}{dt} = \begin{pmatrix} \sigma(x_2 - x_1) \\ x_1(\mu - x_3) \\ x_1 x_2 - \beta x_3 \end{pmatrix}.$$

In our experiments we have used the standard parameters $\sigma = 10$, $\mu = 28$ and $\beta = 8/3$. The solution to the ODE was obtained using the explicit Runge-Kutta method of order 5(4) as implemented by the function `solve_ivp` of the Python library `Scipy` [63]. We discarded any data before $t = 100$ to give time to the solution to converge to the stable attractor [60] and then sampled a data point every $\Delta t = 0.1$ (in natural time units). The Lorenz63 attractor is also known to be mixing [31].

### E.3 Additional details on the numerical verification of the uniform bounds

In this section we discuss how we have obtained the results presented in Fig. 1 of the main main text for the logistic map and in Fig. 4 for the Lorenz63 dynamical system.

As remarked in the main text, the proposed RRR estimator and the classical PCR estimator satisfy the same uniform bound. However, the empirical risk may be (possibly much) smaller for the RRR estimator and hence preferable. To this end, we evaluated, as a function of the number of training points, the empirical risk of PCR and RRR estimators under the same HS-norm constraint, needed to satisfy the assumption of Theorem 3.

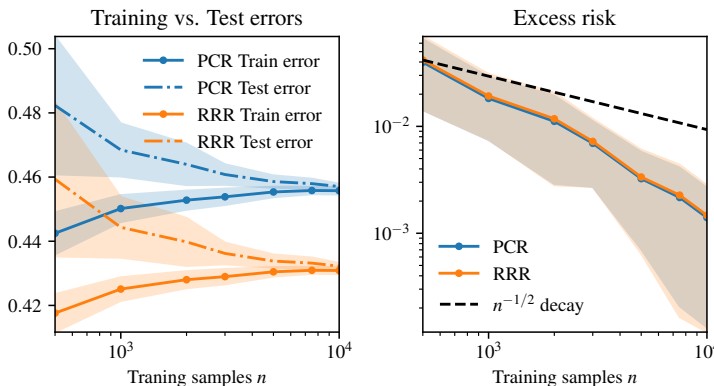

Figure 4: Numerical verification of the uniform bound presented in Theorem 3 for the Lorenz63. Left panel: the training and test risk for RRR are consistently than PCR. Right panel: the deviation between training and test risk decreases faster than $n^{-1/2}$ as a function of the number of training samples.

Table 2: Comparison of the estimators trained for the Beijing air quality experiment.

| Estimator | Training error | Test error |
|---|---|---|
| PCR | 0.5809 | 0.5923 |
| RRR | **0.5780** | **0.5899** |

To achieve the same HS norm for both estimators we first trained the PCR estimator and computed its HS norm. We then adjusted the Tikhonov regularization for the RRR estimator to a value yielding the same HS norm of the PCR estimator. We remark that this procedure in general does not yield the best (w.r.t. the regularization parameter) RRR estimator, but allows us to compare PCR and RRR within the Ivanov setting considered in Theorem 3. Moreover, we have also verified that the upper bound on the scaling $\approx n^{-1/2}$ derived in Theorem 3 empirically holds.

In both Logistic map and Lorenz63 each experiment was independently repeated 100 times, the number of test points is $5 \times 10^4$ and RRR consistently attains smaller empirical risk than PCR.

Table 3: Delay between wind speed peaks and PM2.5 concentration peaks. Positive values correspond to peaks in wind speed occurring *after* peaks in PM2.5 concentration. Coupled modes correspond to complex conjugate pairs. Modes $1, 6, 9$ and $10$ correspond to real eigenvalues and delays can't be evaluated.

| Station | Mode 1 | Modes 2-3 | Modes 4-5 | Mode 6 | Modes 7-8 | Mode 9 | Mode 10 |
|---|---|---|---|---|---|---|---|
| Guanyuan | - | 1.92 hrs. | 2.74 hrs. | - | 1.69 hrs. | - | - |
| Aotizhongxin | - | 1.89 hrs. | 2.61 hrs. | - | 1.64 hrs. | - | - |
| Wanshouxigong | - | 2.01 hrs. | 2.82 hrs. | - | 1.87 hrs. | - | - |
| Tiantan | - | 2.0 hrs. | 2.92 hrs. | - | 1.83 hrs. | - | - |
| Nongzhanguan | - | 2.01 hrs. | 2.96 hrs. | - | 1.84 hrs. | - | - |
| Gucheng | - | 2.06 hrs. | 2.54 hrs. | - | 1.77 hrs. | - | - |
| Wanliu | - | 2.01 hrs. | 3.08 hrs. | - | 1.66 hrs. | - | - |
| Changping | - | 2.04 hrs. | 2.79 hrs. | - | 1.51 hrs. | - | - |
| Dingling | - | 2.0 hrs. | 2.67 hrs. | - | 1.31 hrs. | - | - |
| Huairou | - | 2.02 hrs. | 2.31 hrs. | - | 1.45 hrs. | - | - |
| Shunyi | - | 1.93 hrs. | 2.56 hrs. | - | 1.42 hrs. | - | - |
| Dongsi | - | 1.97 hrs. | 2.76 hrs. | - | 1.8 hrs. | - | - |

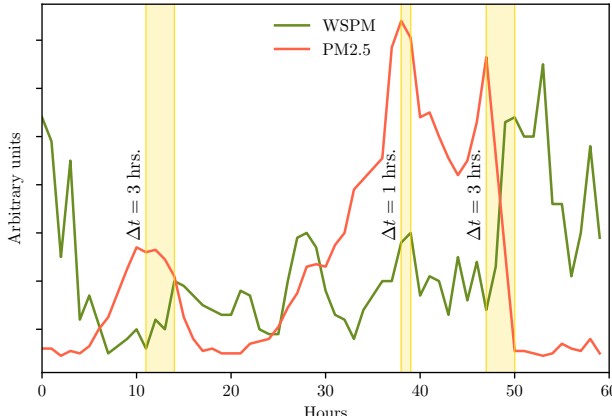

Figure 5: 60 hours of data collected in the Gucheng station. Three peaks of PM2.5 concentration followed by peaks in wind speed. We have annotated the delay in hours between the two peaks.

### E.4 Alanine dipeptide: additional plots

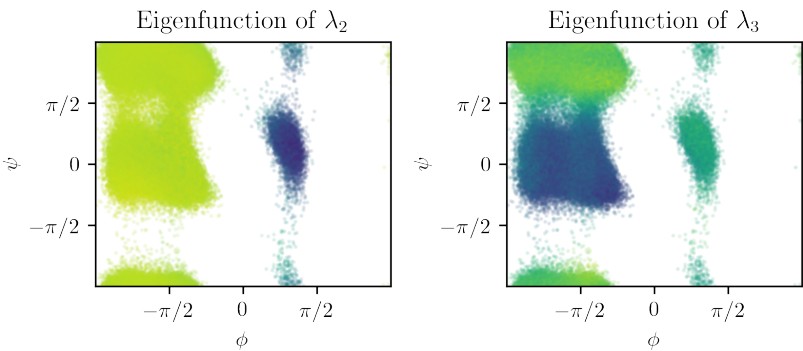

Figure 6: Estimated eigenfunctions in the dihedral angle space of Alanine dipeptide. Each point correspond to a data point in the trajectory. The color encodes the value of the eigenfunction.

### E.5 Additional experiment: Beijing Air Quality Dataset

This dataset [69] consists of hourly measurements of six different air-pollutants along with relevant meteorological variables. Measurements were collected at twelve air-quality monitoring sites in Beijing, from March 1, 2013 to February 28, 2017. The analysis in [69] showed that the presence of Particulate Matter smaller than 2.5 $\mu m$ (PM2.5) is highly correlated to meteorological variables, like humidity and low Wind Speeds (WSPM). In this experiment we show how the modal decomposition of the Koopman operator can enrich the analysis in [69] with dynamical insights. Following this work, we analyse each season of the year separately for better meteorological homogeneity.

We report data for RRR and PCR estimators ($r = 10$) over 7000 out of the 8564 hourly data points collected in winter using an Exponential kernel. Training and test errors are summarised in Tab. 2, RRR achieving slightly smaller test and training errors. The optimal regularization parameter for RRR was chosen by grid search, splitting the data via the `TimeSeriesSplit` as implemented in the `scikit-learn` [47] package. Regularization $\gamma = 10^{-4}$ turned out to be optimal.

As showed in [48], analysing the phase difference of modes corresponding to different observables allows us to infer whether variations of one observable are followed or anticipated by variations of another. Indeed, the modes corresponding to wind speed (WSPM) and PM2.5 concentration reported in Table 3, consistently point out that peaks in PM2.5 are *followed* by peaks in WSPM with a delay

of $\approx 2$ hours. This is reasonable as high wind speeds favour the dispersion of PM2.5, and as wind ramps up toward a peak, PM2.5 concentration is reduced. Reference [64], indeed, argue that pollution concentration is fully readjusted on the basis of wind conditions already after 4 hours.

As a a final illustrative example, in Fig. 5 we show an excerpt spanning 60 hours of data collected in the Gucheng station. We have manually identified three peaks in the PM2.5 concentration followed only $\approx 2/3$ hours later by peaks in the wind speed.

### E.6 Koopman Operator Regression with Deep Learning Embeddings

We have used a Linear kernel $\ell^{-1} \langle x, x' \rangle$ and a Gaussian kernel with length scale $\ell$. Here $\ell = 28 \times 28 = 784$ is the number of pixels in each image. The regularization parameter was chosen by grid search, splitting the data via the `TimeSeriesSplit` as implemented in the `scikit-learn` [47] package. The optimal regularization parameters are, respectively $\gamma_{\text{lin}}=48.33$ and $\gamma_{\text{gauss}}=7.85 \cdot 10^{-3}$.

The CNN kernel is $\langle \phi_{\boldsymbol{\theta}}(x), \phi_{\boldsymbol{\theta}}(x') \rangle$, where the architecture of the network is given by $\phi_{\boldsymbol{\theta}} := \text{Conv2d}(1, 16; 5) \rightarrow \text{ReLU} \rightarrow \text{MaxPool}(2) \rightarrow \text{Conv2d}(16, 32; 5) \rightarrow \text{ReLU} \rightarrow \text{MaxPool}(2) \rightarrow \text{Dense}(1568, 10)$. Here, the arguments of the convolutional layers are Conv2d(`in_channels`, `out_channels`; `kernel_size`). The Tikhonov regularization parameter for the CNN kernel is $\gamma_{\text{CNN}} = 10^{-4}$. The network $\phi_{\boldsymbol{\theta}}$ has been pre-trained as a digit classifier using the cross entropy loss function. Training was performed with the Adam optimizer (learning rate = 0.01) for 20 epochs (batch size = 100). The training dataset corresponds to the *same* 1000 images used to train the Koopman estimators.

In Fig. 7 we compare Linear, Gaussian, and CNN kernels for different initial seeds. As it can be noticed the CNN kernel remains strong across the board, while the forecasting ability of the linear and Gaussian kernels quickly deteriorate as $t$ increases.

Figure 7: Comparison of different kernels in the generation of a series of digits. Starting from a seed image, the next ones are obtained by iteratively using a rank-10 RRR Koopman operator estimator. As in the main text, the first row of each panel corresponds to the Linear kernel, second to Gaussian kernel and last row to CNN kernel.

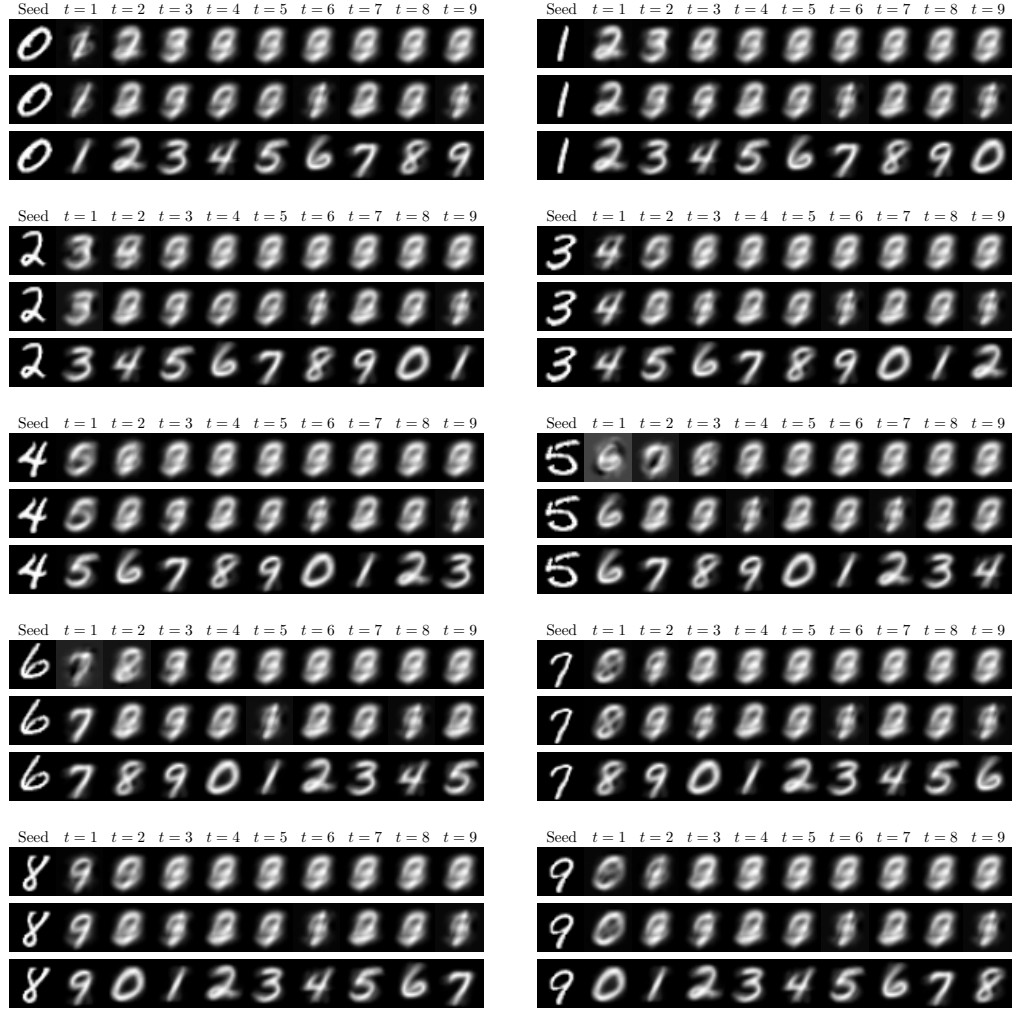