# OpenReview forum: "Learning Dynamical Systems via Koopman Operator Regression in Reproducing Kernel Hilbert Spaces"
_NeurIPS.cc/2022/Conference — NeurIPS 2022 Accept_

### Official Review · Reviewer_Sfxe · 2022-07-01

**Rating:** 7
**Confidence:** 3
**Soundness:** 4 excellent
**Presentation:** 4 excellent
**Contribution:** 4 excellent

**Summary:**

In this paper, entitled "Learning Dynamical Systems via Koopman Operator Regression in Reproducing Kernel Hilbert Spaces", the authors connect the popular and successful theory of Koopman operator theory (and the numerical tool dynamic mode decomposition [DMD]) with statistical learning. This enables a new way to view the estimation of the spectral decomposition of the Koopman operator. In addition, the authors develop new approaches to approximate this spectral decomposition, and find learning bounds for such methods that hold for both i.i.d. and non i.i.d. dynamical systems.

**Questions:**

I have no additional questions other than the ones asked in the Strengths and Weakness section above.

**Limitations:**

The authors did a nice job making it clear where their work required the samples to be i.i.d., and where they did not. I especially appreciated that they had some bounds in the non i.i.d. setting.

As the author's illustrated at the end of Sec. 6, the choice of kernel is very important. The authors mention this, but it would be helpful to have additional discussion around this.

The authors compared their proposed numerical methods to a standard approach PCR. However, there are other existing methods that they could also compare against. While I do not think the author's need to exhaustively test their proposed methods against these, I do think they should mention the existence of these other approaches (e.g. Extended DMD, Hankel-DMD) and be up-front with the fact that how their proposed method compares is unknown.

**Strengths And Weaknesses:**

STRENGTHS:
1. This paper provides a novel connection between Koopman operator theory and statistical learning theory. The authors convincingly demonstrate that this can be leveraged to not only provide a new view on approximating the Koopman mode decomposition, but also inspires new numerical methods for computing such approximation. Given that many of the problems that the community of researchers who use Koopman for forecasting and analysis fit very naturally into this statistical learning framework, I believe this work will be of considerable interest.

2. The bounds developed in the case of the non i.i.d. case represent an exciting direction that I believe the community will be excited about pursuing.

3. The theoretical development of Secs. 3, 4, and 5 are well done. In addition, the appendices provide considerable additional material and background.

4. The emphasis in Sec. 6 on how the proposed numerical methods could improve BOTH forecasting and analysis was appreciated. Not only does it provide support that this framework will be generally useful, but reminds the reader that Koopman operator theory is not just a tool for prediction.

WEAKNESSES:
1. I found the results on the Beijing Air Quality data set somewhat hard to interpret and lacking a real punch. While I understand that wind would disperse PM2.5, why would there be a consistent period over which peaks of PM2.5 would precede peaks in wind speed? It is mentioned that this is in-line with what was argue in Ref. [52], but that paper "argues that pollution concentration is fully readjusted on the basis of wind conditions already after 4 hours". I didn't get how that was related or how the 2 hour estimate found by the Koopman regression agrees with the 4 hour estimate of Ref. [52]. I'm wondering if there might be another result in the data set that would be more interesting and conceptually more straightforward. Are there any modes where the distance between stations is correlated with their corresponding magnitude in the mode? If not, maybe it'd be worth while looking at another data set (like the Washington DC bike share data set) to find something that makes it especially clear what benefits the proposed methods can bring for analysis.

2. In Sec. 6, a nice example of the importance of choosing the kernel is provided. This then makes me want to hear more about the choice of kernels. How should one go about choosing a kernel? For the noisy logistic map and the Beijing Air Quality data, what led the authors to choose the kernel they did? Do other kernels perform differently? This is, of course, a problem in any kernel method, but I think since the author's illustrate its importance, it is necessary to provide a little more discussion on this consideration.

Likewise, how did the author's determine the rank constraint (r)? The value is given, but without any intuition or explanation.

3. The paper was lacking a few references that I thought were relevant and important. In particular, there has been a decent amount of work done recently on Koopman operator theory in reproducing kernel Hilbert spaces. While some of these references were included, the following were not:
   - Gonzalez, et al. 2021 (https://arxiv.org/abs/2106.00106)
   - Mezić, 2020 (https://link.springer.com/article/10.1007/s00332-019-09598-5)
Additionally, inclusion of reference to work on Koopman operator theory for random dynamical systems [Črnjarić-Žic et al., 2020 (https://link.springer.com/article/10.1007/s00332-019-09582-z)] and Koopman operator theory in relation to learning [Mezić, 2022 (https://arxiv.org/abs/2010.05377)] would be helpful for giving the reader a larger picture view of what is already known.

4. The nature of $Z_1, ..., Z_m$ in Lemma 1 is confusing to me.  If they are identical copies, why not represent it as $m Z_1$? I think I'm missing something here.

MINOR POINTS:
1. It would be helpful to define what mixing, with respect to Markov chains is.

2. DMD is one class of methods for approximating the Koopman mode decomposition. There are others (e.g. Generalized Laplace Analysis). Therefore, lines 148-149 should be changed to reflect this.

3. The origin of the square root dependence of R in Eq. 6 is not very clear to me. It would be helpful to provide a sentence or two on why we might expect it to be the case.

4. Do you mean "error increases" as opposed to "error degrades" in line 163?

5. $k(x, \cdot)$ should be defined before it is used (line 183).

6. Should Eq. 13 have an $<$ sign instead of an equality?

7. It should be "using" instead of "use" on line 256.

8. ERM should be defined before it is used in line 273.

9. It should be "shown" instead of "showed" on line 323.

---

> ### Author Response · Authors · 2022-08-02
> **Reply to reviewer Sfxe**
>
> First, we wish to thank you for your detailed assessment of our work, your insightful remarks and suggested references that help us in the revision of the manuscript. In the following we address some of your key comments.
>
> __Remark on the choice of the kernel:__ Thank you for pointing this out, we agree that the reader would benefit from a more detailed discussion on the choice of the kernel. While, in general, we feel that this question deserves a separate study, in the revision we can provide some insights along the following lines. When prior knowledge on the process is available, sometimes it is possible to choose a well-specified RKHS that admits a Koopman operator (see e.g. the logistic map example). This is exactly the idea behind EDMD [Kutz et al. 2016] in which the basis of the observable space is chosen to span (all or at least some) eigenfunctions of the Koopman operator. On the other hand, usually in practice we may have only partial knowledge on the operator. For instance, if we know that the dynamical system satisfies certain physical properties or invariances, then we can choose a kernel that incorporates them (for example, radial kernels are translation invariant). Or, in applications related to (stochastic) differential equations [Mauroy et al. 2020], from the properties of dynamical equations one can deduce that the Koopman eigenfunctions lie in a Sobolev space, and then choose a Matérn kernel. Finally, without any prior knowledge, one can choose a kernel relative to which the empirical cross-covariance has fast decaying singular values.
>
> __Remark on EDMD and HAVOK:__ We would like to highlight that EDMD and projected EDMD are actually a special case of the modal decompositions of KRR and PCR estimators respectively, when one builds a finite-dimensional feature map that embeds the state. With that in mind, we actually provide novel theoretical results for EDMD, introduce optimal low-rank EDMD via RRR estimator and establish their efficient computation (when the dimension of the feature map is bigger than the number of samples). Moreover, Henkel-DMD relates very much our approach, and we are happy to provide more details if the reviewer wishes so. As suggested, we plan to briefly discuss this in the revised version so that readers can better grasp our contributions in comparison to the existing literature, as well as the potential implications of our work.
>
> __Remark on the Beijing Air Quality dataset:__ Thank you for pointing out this weakness. In the revised manuscript we’ll try to make the following observations more evident.  For the air quality dataset we have performed quite a bit of experimentation with different observables which are highly correlated with the presence of pollutants (such as dew point humidity). We ended up reporting the wind speed because the delay times are remarkably consistent across all modes and all measuring stations, making our results non-ambiguous. Comparing the mode delays, we capture temporal _correlations_ between different observables. Our results indicate that whenever there are close peaks in PM2.5 and wind speed in the data, wind speed peaks typically _follow_ pollution peaks, as expected. Conversely, _not_ every peak in the PM2.5 is followed by a peak in wind speed, as meteorological conditions are not affected by the presence of pollution (over such a short time scale). Ref. [52] was added to qualitatively substantiate our claims. The 4 hours estimate of Ref. [52]  depends on the theoretical details of the model, which might be tuned to exactly capture Beijing’s climate, and is an indication of a _full reasjustement_ of the pollution conditions. It should be therefore interpreted as an upper bound.
>
> More importantly, as noted in the general response, we have added an experiment showcasing the use of the spectral decomposition in computational chemistry.
>
> __Remark on the References:__ Thank you very much for pointing out these references. Indeed they are very relevant and they will help readers to better position our work on this fastly evolving topic. We will cite and comment on these references in the revised version.
>
> __Remark on weakness number 4:__ The phrase “independent identical copies of a ” random variable $Z_1$ means that for each $i,j\in[m]$, $i\neq j$, random variables $Z_i$ and $Z_j$ are independent and share the same law as $Z_1$.
>
> __Remarks on Minor points:__ Thank you for pointing out these minor issues, which we will address in the revised version. Additionally, we would like to comment on the following two:
>
> 3) Notice that the risk is HS-norm squared. At line 655 we can see that the $L^2$-norm error of the forecast is bounded by the norm, and not norm squared.
>
> 6) It seems that $\leq$ relation in eq. 13 is correct. We fail to see the reason to have a strict inequality since bounding rhs of equality in line 255 is based on Hölder’s inequality and HS-operator norm inequality for finite rank operators, which may not be strict.

---

> > ### Comment · Reviewer_Sfxe · 2022-08-05
> > **Reply to authors**
> >
> > Thank you very much for your response - it was well written and provided clarity, especially around the choice of kernel, the Beijing Air Quality results, and the relation to EDMD/Hankel DMD. I am glad to hear you are also extending your experimental results. Computational chemistry/(bio)physics is definitely a very exciting realm to pursue this in. However, when I click the link you provided, I get the error "unable to open because a secure connection to the server "annonymous.4.open.science" ". Can you confirm that you indeed sent the right link and that the settings are made so I can access them.
> >
> > Thank you for taking the suggested references into consideration. I think this will strengthen the paper and how other readers understand how it fits into the context of the broader literature.
> >
> > Thank you for explaining weakness number 4 and minor point 3. I was indeed confused.
> >
> > For minor point 4, it appears that the originally submitted version of the manuscript had a "=" in Eq. 13. However, if I look at the version with the supplement, this has been changed to a $\leq$ sign. Maybe it was corrected later?

---

> > > ### Author Response · Authors · 2022-08-05
> > > **reply to the reviewer Sfxe**
> > >
> > > Thank you for your reply.
> > > Indeed, eq. 13 was a typo that was fixed in the supplementary version, sorry for the confusion.
> > >
> > > Concerning the link issue, we have tried to verify the access on different devices and two browsers, Safari and Google Chrome. On our side, each time it worked in both ways, by clicking on the link, and by copying the link in the address bar. One possible issue we can think of is that you might be using a public wi-fi network whose settings are not supporting secure protocol (https) needed to ensure anonymity. In this case, accessing via private network should fix it. However, if the problem persists, please let us know and we’ll look for another way to share the results.
> > >
> > > Thank you!

---

> > > > ### Comment · Reviewer_Sfxe · 2022-08-05
> > > > **Reply to authors**
> > > >
> > > > Thank you for confirming the link. You were indeed right - once I moved to a different wi-fi network I was able to access it.
> > > >
> > > > The new results for the noisy logistic map and Lorenz 63 systems are nice. I agree that they provide new insight and support to the theory developed. It would be worth remarking on why the noisy logistic map appears to follow the $n^{-1/2}$ decay of risk, and Lorenz decays faster. I see that the train error increases for the Lorenz system, with increased number of samples. Given that your metric is $|R_{test} - R_{train}|$, might this train increase in error actually be leading to a much smaller excess risk value? How should one interpret such a scenario?
> > > >
> > > > The molecule Alanine dipeptide results are very nice. I think you explained it well, in simple language. To me, this provides a bigger punch than your Beijing Air Quality results and you may consider changing focus.

---

> > > > > ### Author Response · Authors · 2022-08-05
> > > > > **Reply to reviewer Sfxe**
> > > > >
> > > > > Thank you for your reply, we are glad it worked and that you find additional experiments useful.
> > > > >
> > > > > We don’t have a definite answer for this question, but we might give some insights. Notice that Lorenz-63 dataset comes from a deterministic continuous dynamical system. For any time-step this gives a Markov chain where the transition kernel is deterministic, implying that $Z_\pi^*Z_\pi = S_\pi^* \S_pi$, impling that  the irreducible risk is zero. As you have argued, this probably has an impact, since the chosen metric for the Lorenz-63, contrary to the case of the noisy logistic map, coincides with the statistical bound on the excess risk. We will give it more thought, and provide a discussion in the revised version.
> > > > >
> > > > > Thank you!

---

> > > > > > ### Comment · Reviewer_Sfxe · 2022-08-07
> > > > > > **Reply to authors**
> > > > > >
> > > > > > Great thank you!

---

### Official Review · Reviewer_VeMo · 2022-07-02

**Rating:** 8
**Confidence:** 3
**Soundness:** 4 excellent
**Presentation:** 4 excellent
**Contribution:** 3 good

**Summary:**

This paper introduces a class of algorithms to learn dynamical systems from observation data. One of the common approaches for this problem is to rely on Koopman operator theory, giving algorithms related to the well-known Dynamic Mode Decomposition (DMD). Here, the authors work in the setting of stationary (discrete time) Markov chains with a continuous state. In this framework, they analyze the problem of approximating the (stochastic) Koopman operator when the hypothesis space is that of Hilbert-Schmidt operators over a RKHS, from a statistical learning perspective. They study how to define a notion of risk for this problem, show connections to the usual one-step-ahead prediction error which is often used to learn such dynamic representations. They introduce several algorithms generalizing existing approaches, and study a number of its theoretical properties, such as the approximation capabilities of the risk minization, what happens for the empirical risk estimation and derive learning bounds in that case. They also address the (usual case) of when the data are not only pairs (x_t, x_t+1) sampled at random points, but a full trajectory with regular sampling, in which case the effective sample size is reduced. An experimental part shows how the proposed method perform to approximate the actual Koopman operator for a version of the logistic map, and prediction capabilities for a real time series dataset and a dynamic version of the MNIST data.

**Questions:**

- Maybe the authors could provide practical examples of systems satisfying some of the key hypotheses or not (e.g. discrete/continuous time, existence of a stationary distribution, stochasticity or not, time homogeneity of the markov chain, time reversibility that makes the spectral analysis easier), or state which ones are satifsied in the experiments.

- What is Ivanov regularization compared to Tikhonov (mentioned in Sec. 5)

- How hard would it be to test the sharpness of the learning bounds on the logistic map case ? I believe it would make for an interesting experimental analysis, as a function of the number of trajectories, the time, the rank of the operator, the kernel, etc

- Similarly, indicating the complexity as a function of the number of samples and other factors (rank…) would be interesting.

- In table 1, it looks like KRR is overfitting the training data, while obtaining comparable performance on test data to RRR (except on the eigenvalues estimation). How do the authors explain this behavior and the fact that low rank is preferable here?

**Limitations:**

The authors did not discuss specifically limitations, but the working hypotheses are clearly stated. As such, the analysis is limited to discrete time systems, which does not seem that big a problem, though it would have been nice to test the approaches on data arising from the integration of ODEs.    The complexity of the algorithms are not discussed directly either, it would help having an idea of how well the method would scale to larger data. These limitations are relatively minor and the depth of the results presented here is extensive already.

**Strengths And Weaknesses:**

Strengths :

- The authors introduce a very nice theoretical framework to study Koopman operators as Hilbert Schmidt operators, allowing to endow a norm to the space of bounded operators (of finite rank) on a Hilbert space, corresponding to the space of observables. This in turns allows to precisely define a notion of risk in this case, that is consistent with intuition. Overall the theoretical analysis is extensive and well presented.

- They show the approximation capabilities of such a framework (in terms of approximation of the operator, and the corresponding spectrum) and derive learning bounds, both in expectation and in the empirical case, including in the (susual in practice) non i.i.d. case.

- There are clear connections with existing approaches aimed at learning dynamical systems from data, among them the classical (kernel) DMD. The framework encompasses and unifies these different approaches nicely.

- The kernel point of view is not restrictive since in practice it can correspond to features learned by a neural network, which is a common strategy when learning dynamical systems from data.

Weaknesses :

- The framework that is introduced is limited to discrete time dynamical systems, which is sufficient in most cases. I wonder how hard it would be to extend all the theory to continuous time systems. I am not that familiar with continuous Koopman operator theory but maybe the authors could discuss this and the additional difficulties that would arise.

- The experimental part, though interesting, only considers one non-standard real dataset. While I appreciate the application to real data, it feels like the paper would benefit having experiments of standard benchmarks in the field, e.g. the Lorenz-63 model (possibly noisy, as e.g. in [1]), or other systems arising from differential equations.

- On a related note, the three competing methods (PCR, RRR,KRR) can all be interpreted in the proposed theoretical framework, which is nice, but other classical approaches could have been compared, e.g. more empirical deep-learning based estimations of a Koopman operator, e.g. ref [27] in the paper. It would have been nice to position such works, if possible, within the framework that is proposed here.

[1] Li, X., Wong, T. K. L., Chen, R. T., & Duvenaud, D. (2020, June). Scalable gradients for stochastic differential equations. In International Conference on Artificial Intelligence and Statistics (pp. 3870-3882). PMLR.

---

> ### Author Response · Authors · 2022-08-02
> **Reply to reviewer VeMo**
>
> First, we wish to thank you for your detailed assessment of our work and your suggestions that help us in the revision of the manuscript. In the following we hope to clarify your questions, and address your concerns.
>
> __Remark on continuous time dynamical systems:__ In general for a continuous time dynamical system one doesn’t have one Koppman operator, but rather a semigroup of Koopman operators whose generator is a Lie operator. As you pointed out, we address the Koopman operator for discrete Markov chains, which is essentially enough for studying continuous time dynamical systems from equally spaced observations in time. Namely, the time step $\Delta_t$ then defines a time-scale of a single Koopman operator from the semigroup, and a continuous model can be built as $\exp(\frac{t}{\Delta t} A_\pi)$. However, beyond this case, when one is limited to observations that are irregular in time, one should approach the learning through the Koopman semigroup generator, or its adjoint Liouville operator. An interesting way to use RKHS in the continuous setting was very recently proposed by _Rosenfeld et al. (2022). Dynamic Mode Decomposition for Continuous Time Systems with the Liouville Operator. Journal of Nonlinear Science_, where the kernel is built between trajectories. However, at present, this line of work comes only with consistency results and lacks statistical bounds and a statistical learning framework. Among other things, addressing this question is one of our future research directions.
>
> __Remark on the experiments:__ Thank you for your suggestions. We have expanded the experiments and tried to incorporate your suggestions. In particular, the noisy Lorenz-63 dynamical system will be included in the revision, while some preliminary results are reported in the general answer to all reviewers.
>
> __Addressing the questions:__
>
> - In Ex. 1 we give a practical example of a discrete-time dynamical system with additive noise and conditions under which there exists an invariant measure. If the reviewer finds it useful, in the revised version we can expand on this and discuss when equally time spaced observations of a stochastic ODE form a time-homogeneous Markov chain, and give an example arising in computational chemistry of time-reversible Markov process. Additionally, we will clarify that noisy logistic map is the case of non-self-adjoint finite rank Koopman operators with universal RKHS.
>
> - Ivanov regularization is formulated as minimizing the risk over a ball of HS-operators of radius $\gamma$, and it can be shown that it is equivalent to performing a Tikhonov regularization with another regularization parameter $\tilde{\gamma}$. Often one can use Ivanov regularization for theoretical analysis, while in practice performing Tikhonov one.
>
> - We are currently investigating the sharpenes of the bounds in this experiment.
>
> - In the appendix, we will add the complexity for computing empirical estimators KRR, PCR and RRR from data, as well as the complexity for computing their modal decompositions.  We’d be glad to address the complexity issue more in detail if the reviewer wishes so.
>
> - We do not have a definitive answer to this question but we give some insights. First, notice that the true rank of the Koopman operator is 21 in this case, with eigenvalues quickly approaching zero. Next, the rank of KRR equals the sample size $10^4$, while RRR and PCR are limited to rank $r=3$. This results in the smaller training error for KRR. On the other hand, it means that the KRR estimator has a dominant cluster of nonzero eigenvalues around zero whose eigenfunctions try to approximate the kernel of the Koopman operator. This introduces additional difficulty for KRR compared to RRR which fits only three dominant eigenvalues.

---

> > ### Comment · Reviewer_VeMo · 2022-08-05
> > **Reply to authors**
> >
> > Thank you for your reply and addressing my comments .
> >
> > It is nice to see the additional experiments on the Lorenz-63 data (actually noiseless, judging from the equation in the code repository, contrary to what the authors say in the reply) and the computational chemistry.
> >
> > I would indeed appreciate the mention of which hypotheses the studied systems satisfy, I believe they help showing the theoretical framework is set in reasonable and broadly applicable settings.
> >
> > I believe that the continuous Markov Chain setting is a natural extension that fits a number of cases of practical interest and is definitely a nice perspective to want to generalize the results in this context. Thanks for providing the reference as well. I could be worth citing in this paper for the reader to know what has been done so far in the continuous case.
> >
> > I have no further questions as of now.

---

> > > ### Author Response · Authors · 2022-08-05
> > > **Reply to reviewer VeMo**
> > >
> > > Thank you for your reply.
> > >
> > > If the paper is accepted, we will use the additional page allowed for the camera ready to implement all of the above suggestions.  In particular, continuous time dynamical systems will be discussed, and the salient properties of the logistic map and Lorenz-63 will be stated.
> > >
> > > Concerning the Lorenz-63, thank you for pointing it out. Indeed, we implemented noiseless Lorenz63 as it was considered for example in Giannakis et al. 2021.
> > >
> > > Thank you!

---

### Official Review · Reviewer_bVru · 2022-07-12

**Rating:** 7
**Confidence:** 2
**Soundness:** 3 good
**Presentation:** 1 poor
**Contribution:** 3 good

**Summary:**

The paper provides the formalism and statistical learning theory of learning a dynamical systems in RKHS where the problem of estimating the Koopman Operator in the Hilbert space turns into a nonparametric kernel regression. The mathematical framework encompasses DMD and kernel DMD as special cases and enjoys the connections to classical conditional mean embeddings where some developed theory already exists. Based on the formalism several estimators for the Koopman Operators are provided and the advantages of Reduced Rank Regression (RRR) estimator over others are discussed. Finally some experiments on toy data and sequential generation of MNIST digits are provided showing that deep learning kernels can be plugged into the algorithms for the kernel regression.

**Questions:**

**Numbers are line numbers in PDF**

77 Is the transition kernel simply the conditional probability distribution $p(X_{t+1}|X_t)$?

79 In Eq. 1, I hadn't seen the notation $p(x,dy)$; is it the same as $p(x,y)dy$? Can you clarify this or change the notation?

101 I don't quite understand the decomposition described in Eq. 2. I thought after lifting nonlinear dynamical system using the observables the Koopman Operator evolves the dynamics in the lifted space linearly. If this is the case then why is there a superscript $t$ on the left side and what do the $\lambda_i^t$'s correspond to?

108 interested into -> interested in

121 Why do we approximate $Z_{\pi}$ via $S_{\pi}G$ and not with $G$ itself if we want $Z_{\pi}$ to be a  Hilbert Schmidt operator?

129 Multiple measures are introduced throughout the paper making it confusing; $\rho,p,\pi,dx,dy$ can you clarify in words what each of these means?

147 later -> latter

164 the error degrades as t increases ... forecasting will gets [get] increasingly harder for larger t; aren't these two in contradiction?

160 How should we interpret the RHS of Eq. 7 and the bound in line 160? Aren't we expecting these to vanish with $r \rightarrow \infty$?

175 Remark 2 is providing other bounds for normal $A_{\pi}$ and $G$ but still it's not clear that the error vanishes for large $r$.

194 In Eq. 10, the function $\phi$ is imposed by the choice of the kernel; while this is common in kernel regression problems, I thought that the nonlinear dynamical system dictates the what the Koopman Operator observables or lifting function are; meaning that the choice of kernel is not arbitrary; can you clarify this?

248 In Thm. 3, the error again doesn't seem to go zero with n going to infinity, isn't expected for the error to vanish with large data size? Same for Prop. 2 and 3.



**Limitations:**

* Empirical validation of the theoretical results and bounds.
* Tighter bounds going to zero with increased approximation precision or data size.
* Complex notation with minimal intuitive arguments helping the reader going through the logical flow of the mathematical arguments.
* Limited experiments on toy examples.

**After rebuttal**

* New experiments address my first and last points.
* My second point was a technical oversight on my side, no action in needed from authors' side.
* I still think that the notation is too heavy and makes it hard to follow the arguments on an intuitive level for the broader community.

**Strengths And Weaknesses:**

**Strengths**

* The proposed contributions are significant and natural next steps of the transport operator theory. The empirical risk minimization helps providing a statistical foundation to learning dynamical systems using Koopman Operators. It further allows for the extension of existing kernel regression methods for learning dynamical systems as proposed by the authors.

* The paper is mathematically rigorous, theorems are accompanied with proofs and statistical bounds are given providing statistical guarantees.


**Weaknesses**

* Overall the notation is too complex and heavy, I would suggest communicating the main ideas through a lighter notation and moving the full notation to the supplementary. More comments about the notation are in the questions section.

* The experimental part of the paper is quite limited. Although I understand that most of the contributions of the paper are theoretical but I think some of the mathematical notation can be moved to the supplementary and more experiments can be added to 1) confirm the theoretical results empirically by plotting the errors for known cases as a function of r, data size, and other variables 2) show applications in time series which is the main motivation of the theoretical developments in the paper 3) compare different Koopman Operator estimation methods with existing ones such as extended DMD and the neural net-based ones. At least a discussion on under what circumstances kernel based methods should be preferred to existing ones would be helpful.

* Some of the bounds provided are quite loose and they don't vanish with infinite data or infinite rank precision. Am I missing something or are the bounds indeed weak?

* There are a few mentions of statistical guarantees missing from dynamical systems literature. I don't know to what extent this is true given that many statistical time series methods are developed (such as variants of HMM, LDS, SLDS, etc.) with their corresponding theoretical bounds and inference algorithms. Can the authors be more clear about what they exactly mean by this and also include a short summary of the above-mentioned methods and how they compare to Koopman Operator based methods?

* In table 1 the $\lambda_{2,3}$ error seems pretty large, shouldn't we expect to see excellent performance on this toy example? Also can the authors report the relative errors instead of the absolute ones?

* In Fig. 1, why are the images so blurry even for the CNN kernel?

---

> ### Author Response · Authors · 2022-08-02
> **Reply to reviewer bVru**
>
> First, we wish to thank you for your suggestions that help us in the revision of the manuscript. In the following we hope to clarify all your questions, and address your concerns about the  presentation.
>
> __On the notation:__ In this paper we have followed usual notation in the theory of RKHS and CME, see e.g. [Caponnetto & De Vito, 2007; Ciliberto et al. 2016] and Markov transfer operators, see e.g. [Meyn & Tweedie 1993], and reference therein. Moreover, we tried our best to simplify the notation while being mathematically rigorous.
>
> __On the experiments:__ Thank you for the remark. As reported in the general answer to all reviewers, we have expanded the experiments and incorporated your suggestions. Concerning the questions on eigenvalues (Tab. 1), we’ll report relative errors. Moreover, note that from spectral theory we know that eigenvalue perturbation bounds for non-self-adjoint operators often are much larger than the norm of perturbation of the operator (in our case the true risk). Concerning MNIST images (Fig. 1), please note that KOR estimators predict the _expectation_ of the next image. So, the blurry appearance is expected since the output is a sort of barycentric image for a class of digits.
>
> __On EDMD and NN:__ EDMD is a special case of KOR when one builds a finite-dimensional feature map (see also our reply to rev Sfxe). Unlike the usual EDMD empirical setup where one matches a feature map to the Koopman operator, in our experiments we assume no such prior knowledge is available. Concerning NN-based estimators, we presented an example of CNN kernel, showing the possibility to combine KOR with NN.
>
> __On the bounds:__ We respectfully disagree, all statistical bounds (Prop. 2 and 3, Thm. 3) converge to zero w.r.t. the increasing number of samples. Similarly, the KMD approximation bounds (Thm. 1, Rem. 6) for the consistent estimators and well-specified RKHS, converge to zero, see reply to rev. bsCx.
>
> __On statistical guarantees in the literature:__ Thank you for pointing out the ambiguity in lines 20-27. We will change it accordingly in the revision. As you correctly state, previous works on the statistical guarantees for learning (stochastic) linear systems from data exist, as well as extensions to specific forms of non-linearity. All of these can be seen as a parametric approach. In fact, what we intended to say is that, until now, estimating the Koopman operator, which is a very powerful tool to investigate general non-linear stochastic dynamical systems, comes without statistical guarantees. Ours is a nonparametric estimation of DS from data, and so it seems more applicable. In particular, specifying a linear kernel reduces the KOR framework to learning stochastic linear systems from data.
>
> __Answer to questions:__
>
> 77 yes.
>
> 79 $p(x,\cdot)$ is a measure, $dy$ is a notation for integration and not the Lebesgue measure.
>
> 101 Superscript $t$ means that the linear operator is applied $t$-times
>
> 108 ok
>
> 121 $Z_\pi$ is an operator from RKHS to L2, while $G$ is an operator from RKHS to RKHS. Moreover, the objective is to approximate spectral decomposition of Koopman. Hence, we need an operator $G$ that maps a space to itself.
>
> 129 $rho$ is joint probability measure of two consecutive states when the first is drawn from the invariant measure $\pi$, $p$ is Markov transition kernel (a family of probability measures), while $dx$ and $dy$ are just integration symbols.
>
> 147 ok
>
> 164 It is not a contradiction. Thm. 1 implies that, relative to the risk of the estimator, the forecast time-horizon needs to be shorter to control the error on the forecast. This is also intuitively expected.
>
> 160 As mentioned above and in reply to reviewer bsCx, since those are worst-case bounds for general estimator $G$, we do not expect them to vanish in general. However, with more assumptions on $A_\pi$, they will vanish. More refined spectral bounds can be obtained for specific estimators such as KRR, PCR and RRR, which is the subject of our future work.
>
> 175 For normal $A_\pi$ and normal estimator, choosing a higher $r$ of a consistent estimator leads to a vanishing bound. If only the $A_\pi$ is normal, then Rem. 2 just translates eq. (7) to a distance of eigenvalues using the theory of pseudospectra. As said above, more assumptions on $A_\pi$ are needed to make it a vanishing bound for any consistent estimator.
>
> 194 This is a usual misconception from the EDMD narrative that we wished to address. Namely, yes, sometimes you are able to choose the RKHS such that the Koopman operator can be defined on it (well-specified case) in which case learning is easy. However, without prior knowledge, there is no way to practically do it. We obtained results that, even if you cannot choose appropriate RKHS as the space of observables, if the kernel is universal, you can still obtain some theoretical guarantees.
>
> 248 This is probably an oversight, bounds are of the order $\widetilde{\mathcal{O}}{(1/\sqrt{n})}$.

---

> > ### Comment · Reviewer_bVru · 2022-08-05
> > **Reply to authors**
> >
> > I would like to thank the authors for writing the detailed response and addressing my comments and questions. I went through the replies and reviews from other reviewers as well and I'll update my score and review accordingly.
> >
> > I agree with the authors that the statistical bounds indeed go to zero as the number of samples increases. For some reason I missed the logarithmic term in the bounds. I apologize for the oversight.
> >
> > I still find the notation too complex, but I realized that other reviewers don't share this concern with me. This might be because I'm more used to the statistical learning theory notation as opposed to transport theory. If the authors find the current notation necessary for understanding the context while keeping the arguments rigorous I'm fine with keeping it as is, however my personal preference would be to include a simpler and lighter notation in the main text to make it easier for the readers to follow the theoretical arguments and contributions. If the authors decide to keep the notation, it'd be helpful if the authors clarified in notation section of the paper the questions that I wrote in my review.
> >
> > Summarizing the discussion with other reviewers here's a list of what needs to be added to the paper before publication, please make sure these are added in the final version.
> >
> >
> > **Major**
> >
> > - New experiments mentioned by the authors.
> > - Technical oversight on risk vs. excess risk and how it changes the current theoretical results.
> > - Clarification of what statistical guarantees exist (parametric) and what the contributions are (nonparametric).
> > - Citations mentioned by other reviewers on deep learning based estimators.
> >
> > **Minor**
> >
> > - A note on computational complexity.
> > - A note on how to choose the kernel for specific applications.
> > - A note on what other assumptions can lead to better bounds and intuitive arguments space permitting
> > - Clarify $Z_{\pi}: \text{RKHS} \rightarrow L_2$ and $G: \text{RKHS} \rightarrow \text{RKHS}$, also in the notation section clarify $\rho, p, \pi, dx, dy$.

---

> > > ### Author Response · Authors · 2022-08-05
> > > **Reply to reviewer bVru**
> > >
> > > We wish to thank you for your reply, for reconsidering your rating of our work, and for the summary of the planed changes.
> > >
> > > If our work is accepted, we will use the additional allowed page for the camera ready to implement all of the above points. In particular, we will include the additional explanations of the used notation to help the broader community of readers.
> > >
> > > Thank you!

---

### Official Review · Reviewer_bsCx · 2022-07-12

**Rating:** 3
**Confidence:** 4
**Soundness:** 3 good
**Presentation:** 4 excellent
**Contribution:** 1 poor

**Summary:**

In this paper the question of learning the Koopman operator of a Markov chain from samples is studied.

* It is proposed to introduce an appropriate RKHS $\mathcal{H}$ and to restrict the domain of the operator to be $\mathcal{H}$,
while the image would still be in the full $L_2(\pi)$ where  $\pi$ is the stationary measure of the chain.
* This has the advantage that such a restriction is compact and hence can be approximated by finite dimensional operators.

* A risk measure is introduced which measures the quality of the approximation. This measure is equivalent to a risk used in the field of Conditional Mean Embeddings.

* The main novelty is that in this framework it is proposed to approximate the Koopman operator by a finite dimension operator that minimizes the risk, and is regularized by the Hilbert Schmidt norm as an operator $\mathcal{H} \mapsto \mathcal{H}$. This is different from kernel PCA approximation, for instance.

* Results are proved on matrix stability (i.e. small risk implies closeness of eigendecompositions in a certain sense), and generalization properties for the above finite rank regularized operator empirical risk minimization.



**Questions:**

Please see the questions above.


**EDIT Post rebuttal and discussion**


**1 -  Novelty**


It is important to note that studying dynamical systems in RKHS is not new.
Completely equivalent setting, with equivalent risk function, is studied in the field of Conditional Mean Embeddings (CME), which by now is a well developed field.  This is discussed in the paper in Remark 3, and in lines 189-191 in particular.

**1a)** This has consequences, since experiments need to present comparison to other methods. In particular the KRR and PCR. Experiments on Beijing Air Quality data,  and Alanine Dipeptide do not do this (no comparison to KRR) and thus contribute very little to establishing novelty. Other experiments are discussed in what follows.

**1b)**  What is new, is the proposal to minimize the risk among low rank operators $G$, i.e. eq. (12).
 The authors study this proposal in three parts:
   * **1b1)** How properties of $G$ translate to properties of true Koopman operator (Theorem 1).
   * **1b2)** How empirical risk relates to true risk for this approximation (Theorem 3).
   * **1b3)** Some experiments with this approximation.

We now discuss the issues.

**2 - Issues**

The paper has a critical problem in points (1b1) and (1b2) above.
Specifically, in Theorem 1 it is shown that properties related to spectral decomposition of the true  operator can be reconstructed from $G$ if all the quantities $\|G\|$, $\|Gf\|$ and risk $R(G)$ are small.


**2a)** However, as pointed out in my review, *and confirmed by the authors*, there is no apriori reason
       why these quantities should be small. They can be arbitrarily large, no matter how large the rank $r$ is.
       As a consequence, it may happen that $G$ is useless for reconstructing the true operator.

Similarly, the quantity $|G|_{HS}$ is key in Theorem 3 (via $\gamma$). It is not clear that both risk and $|G|_H$ can be
small together. This also is not sufficiently investigated empirically.

**2b)** In their response to the review, the authors state a new result, that the above issue does not happen
       when the true Koopman is self-adjoint. However, Koopman operators are almost never self-adjoint.
       This, for instance, is *explicitly* stated in reference [A] below, see third paragraph of Section 2.2.

This is a standard fact, and the authors should have been aware of this.

Note also that even if $\|G\|$ were small, as the authors claim in the self-adjoint case, this does not solve the issue of Theorem 3 with $\|G\|_{HS}$.

**2c)**  Note that when the true operator is not known, Theorem 1 is the only basis upon which one may conclude that  $G$ represents the true operator. Moreover, while there is no aprori way to bound $\|G\|$, $\|Gf\|$ and  $R(G)$, these bounds may be computed empirically.

 Thus, all that is needed, is to present experiments where we see that empirically, the quantities $\|G\|$,$\|G\|_{HS}$, $\|Gf\|$ and  $R(G)$ take values that imply practical  bounds in Theorems 1,3.

However, there is not nearly enough experimental evidence that Theorems 1,3 are useful, and and consequently that the paper is useful.

I am not completely convinced that indeed the quantities can be small in realistic situations. To put it simply, working in RKHS instead of the full function space $L_2$, has computational advantages. However, it has also a price, that the $G$ norm can explode. The authors failed to verify that there exist useful situations where the price is not too high.


Therefore in my opinion this paper should not be accepted.



[A] Mauroy, Alexandre and Susuki, Yoshihiko and Mezi{\'{c}}, Igor,
    Introduction to the Koopman Operator in Dynamical Systems and Control Theory,
    The Koopman Operator in Systems and Control: Concepts, Methodologies, and Applications,2020,
    Springer International Publishing.

open access at: https://link.springer.com/chapter/10.1007/978-3-030-35713-9_1






**Limitations:**

No discussion of negative societal impact was included.

**Strengths And Weaknesses:**

This is a well written paper, and the idea using the kernel in this way is elegant.

At the same time, I believe the paper would benefit significantly from additional empirical evaluation.

To better understand the significance of the results, and the theoretical novelty, I have the following questions.
Based on this, I will be glad to reevaluate the review following the author feedback period.

1) In Proposition 1, it is shown that there is always a finite rank operator $G$ making the risk arbitrary small.
However, the norm of $G$ as an operator $\mathcal{H} \mapsto \mathcal{H}$ can become arbitrarily large. Is this correct?

2) In Theorem 1, are the norms that are used for $\|Gf\|$ and $\|G\|$ norms in $\mathcal{H}$?
3) In that case, when rank $r$ grows, is it even clear that the bounds go to $0$? This is not obvious, since when the rank grows,
 the risk would go to $0$, but the norm of $G$ can grow to infinity. Correct?
  Then, is the only way to determine whether the bounds are non-trivial empirical? I.e computing the norm and using generalization bounds?

4) Can the authors describe the novelty/difference in methods used to show generalization results, compared to existing literature on CME?

5) It is claimed (line 260) that the proposed RRR should have better risk than the PCR estimator for the same HS norm. While this is reasonable, I would expect this do be verified experimentally, since this is the proposed novelty of the paper.
Similarly, in view of question 3 above, it is not clear there exist $G$ with small rank, risk, and HS norm. This should be shown empirically.

---

> ### Author Response · Authors · 2022-08-02
> **Reply to reviewer bsCx**
>
> First, we wish to thank you for your suggestions to elaborate on our bounds. Some detailed answers that will be included in the revision are given below.
>
>
>
> __Answers to questions:__
>
> 1) As you correctly point out, minimizing excess risk $\Vert Z_\pi - S_\pi G \Vert_{HS}^2$ may lead to the growth of the Hilbert-Schmidt norm of $G$ as an operator ${\cal H}\to {\cal H}$. However, this happens only in the misspecified case, i.e. when the RKHS is not an invariant subspace of $A_\pi$. On the other hand, the *operator norm* $\Vert G \Vert$ that appears in Thm. 1 may remain bounded. To clarify this aspect we provide a stronger version of Prop. 1 where for time-reversal-invariant Markov chains, i.e. self-adjoint Koopman, one additionally has $\Vert G \Vert \leq 1$. The idea is the following:
>
>
>
> Observe that $\Vert Z_\pi - S_\pi G \Vert^{2}\_{HS} = \Vert(I-\Pi_{\cal H}) Z_\pi \Vert^{2}\_{HS} + \Vert \Pi_{\cal H} Z_\pi - S_\pi G \Vert^{2}\_{HS}$, where $\Pi\_{\cal H} = S\_\pi (S\_\pi^* S\_\pi)^\dagger S\_\pi^*$ is the orthogonal projector onto $Im(S_\pi)$. Now, if either RKHS is well-specified or dense in $L^2_\pi({\cal X})$, the first term is zero.
>
> For the second term, take $G = [[ C\_{\gamma}^{-1} S\_{\pi}^* Z\_{\pi} ]]\_{r}$, and observe that $\Vert A_\pi\Vert=1$. Hence, $\Vert\Pi\_{\cal H} Z\_\pi - S\_\pi G \Vert\_{HS} \leq \Vert\Pi\_{\cal H} - S\_\pi C_\gamma^{-1} S\_\pi^* \Vert \Vert S\_\pi \Vert\_{HS} + \Vert C\_{\gamma}^{-1} S\_\pi^* Z\_\pi - [[ C\_\gamma^{-1} S\_\pi^* Z\_\pi ]]\_r \Vert \Vert S\_\pi \Vert\_{HS}\leq \Vert S\_\pi \Vert\_{HS} ( \Vert\Pi\_{\cal H} - S\_\pi C\_\gamma^{-1} S\_\pi^* \Vert + \sigma\_{r+1}(C\_\gamma^{-1} S\_\pi^* Z\_\pi ) ).$ So, using $Z_\pi^* S_\pi C_\gamma^{-2} S_\pi^* Z_\pi\preceq \frac{1}{\gamma} S_\pi^* A_\pi^* A_\pi S_\pi \preceq \frac{1}{\gamma} S_\pi^* S_\pi$, we obtain $\sigma_{r+1}(C_\gamma^{-1} S_\pi^* Z_\pi )\leq \sqrt{ \frac{\lambda_{r+1}(C )}{\gamma}}$. Next, since $S_\pi C_\gamma^{-1} S_\pi^* $ converges to $\Pi_{\cal H}$ as $\gamma\to0$, given $\delta>0$, set $\gamma=\frac{4\Vert S_\pi\Vert^2_{HS}}{\delta} \lambda_{r+1}(C)$ and choose $r$ large enough s.t. $\Vert\Pi_{\cal H} - S_\pi C_\gamma^{-1} S_\pi^* \Vert\leq \frac{\sqrt{\delta}}{2 \Vert S_\pi \Vert_{HS}}$. This implies $\Vert\Pi_{\cal H} Z_\pi - S_\pi G \Vert_{HS}^2 \leq \delta$.
>
> Finally, if $A_\pi$ is self-adjoint, then $S_\pi^* Z_\pi = S_\pi^* A_\pi S_\pi\preceq S_\pi^* S_\pi$. So, $|| G || = \sqrt{\lambda_1( C_\gamma^{-1} (S_\pi^* Z_\pi)^2 C_\gamma^{-1}) }\leq \frac{\lambda_1(C )}{\lambda_1(C)+\gamma} \leq 1$.
>
> 2) Yes. Note, however, that even if $\Vert G\Vert$ is large, $\Vert G f \Vert$ may be small for some $\Vert f\Vert=1$.
>
> 3) As proved above, for the well-specified RKHS, there are no issues with the bounds. Moreover, in the more difficult misspecified case, Thm. 1 / Rem. 2 still provide useful bounds for important time-reversal-invariant Markov processes.
>
>
> 4) Our results readily apply to the CME setting with two different RKHSs and to the best of our knowledge, reduced rank regression is novel in CME literature as well. Moreover, the statistical analysis in Sec. 5 has novel components, sharp (iid and non iid) estimation bounds for the cross-covariance operator and a general risk analysis, a starting point for further study in the CME context. All in all, although our results are relevant to CME, our work is primarily focused on linking the risk to the spectral decomposition of the operator, something which is not considered in the CME literature.
>
>
> 5) This ultimately depends on the problem at hand. For instance, if $A_\pi$ is compact with few dominant eigenvalues and the problem is well-specified, then there is a low rank $G$ with bounded norm and small excess risk. As suggested, we show empirically that for the logistic map and Lorenz 63 dynamical system, RRR has a consistent advantage over PCR.
>
> __Remarks on our contributions:__ Apart from the novelties mentioned in our reply on question 4), we wish to clarify that, to the best of our knowledge, the existing guarantees stating that DMD techniques recover Koopman eigenvalues and eigenfunctions are based on the assumptions that cannot be verified when the dynamical system is unknown (see Mauroy et al. 2020, and replies to bVru and Sfxe). Our intention was to provide useful worst-case guarantees when one has none, or little, prior knowledge. In this respect, Thm. 1 gives bounds that reveal an interplay between the excess risk, the number of eigenfunctions to reconstruct (rank), the observable of interest, and the forecast time-horizon. Moreover, we believe that our results are an important initial step in developing a statistical learning framework for spectra, eigenspaces and modes of the Koopman operator that is currently missing in the literature, despite the overwhelming number of new methodological papers on estimating Koopman operators from data.

---

> > ### Comment · Reviewer_bsCx · 2022-08-08
> > **response to authors**
> >
> > Thank you for your reply.
> > Unfortunately, I still have considerable concerns.
> >
> >
> > Point 5:
> > >we show empirically that for the logistic map and Lorenz 63 dynamical system, RRR has a consistent advantage over PCR.
> >
> > Please explain this. For the logistic map experiment in the paper, there is no considerable advantage of RRR over KRR, and KRR is known. This leaves us with at **most one experiment**,  where RRR has an advantage, the Lorentz 63. This one was added post review, and I cann't access it at the moment (similar problem to what other reviewers had). Please add the experiment results to the paper revision, this is the preferred method of adding results at NeuRIPS.
> >
> > Even if there were no accessibility issues, one dataset showing improvement hardly justifies a new method. Am I missing something?
> >
> >
> > Point 4: This will be taken into account, thank you.
> >
> > Points 1-3:
> >
> > **(a)** Can you state formally the new results? To me, this is far too central a matter to be mentioned in passing in a few lines.
> >
> > **(b)** In any case, I do not see how this addresses the main issue:  Why would the RKHS not be miss-specified?
> > Why do you expect that
> > >either RKHS is well-specified or dense in $L_{\pi}^2$
> >
> > ?
> >
> > To my understanding, this assumption is absolutely crucial. Yet no justification is made for it either theoretically or experimentally. This significantly affects the impact of the paper.

---

> > > ### Author Response · Authors · 2022-08-09
> > > **Reply to reviewer bsCx's additional comments**
> > >
> > > Thank you for your response and your time. As you have requested __a revised version of the paper and supplementary material is uploaded__ (we would like to point out that this is the first year that this has become possible at NeurIPS). Changes are colored in red. Due to current space constraints, all experiments are in the Appendix E, while the revised Proposition 1 is in the main body with the proof in the Appendix B.1. Please note that, if the paper is accepted, we will use the additional page allowed for the camera ready version to __provide discussions and implement all the remarks of the reviewers__. Below we address your concerns.
> > >
> > >
> > > __Point 5:__
> > >
> > > > “Please explain this.”
> > >
> > > Please see our earlier general reply to all reviewers for a discussion of the new experiments, which is now also included in Appendix E of the updated manuscript.
> > >
> > > > “For the logistic map experiment in the paper, there is no considerable advantage of RRR over KRR, and KRR is known.”
> > >
> > > We respectfully disagree. As we have discussed in the paper _“the RRR estimator always outperforms PCR, and in the estimation of the non-trivial eigenvalues [...] attains the best results”_, see also the last column of Table 1 and Appendix E.1.
> > >
> > >
> > >
> > > >”one dataset showing improvement hardly justifies a new method. Am I missing something?”
> > >
> > > We believe that you have not considered our results on the Beijing Air quality dataset (see Table 2 in Appendix E). More importantly, __we feel you are focusing only on one aspect of the paper, while its goal – as stated in the abstract is to _“formalize a framework to learn the Koopman operator from finite data trajectories of a dynamical system”_.__  To this end, the paper contains a number of novel theoretical results; proposing a new empirical estimator is just one of them.
> > >
> > >
> > >
> > > Finally, __the goal of the experimental section is to highlight the practical implications of the proposed framework__. In the original paper we have illustrated the following:
> > > 1. Superiority of RRR over other methods (Logistic map)
> > >
> > > 2. Modal decomposition (Beijing air quality)
> > >
> > > 3. Ease of interaction with deep learning methods (MNIST)
> > >
> > > While, in the new experiments we have also shown:
> > >
> > > 4. Empirical validation of the learning bounds (Logistic map + Lorenz63)
> > >
> > > 5. Spectral decomposition (Alanine Dipeptide example)
> > >
> > > Once again, the superiority of the RRR estimator is only one aspect that we illustrate empirically. Having said this, we recall that the RRR estimator outperforms PCR on three datasets (Logistic map, Lorenz63 and Beijing air quality).
> > >
> > >
> > >
> > >
> > > __Points 1-3: a)__
> > > As stated above, the revision will incorporate our detailed reply to you by making a formal statement of Proposition 1 in the main body and the proof in Appendix B.1.
> > >
> > >
> > >
> > > __Points 1-3: b)__
> > >
> > > > “In any case, I do not see how this addresses the main issue”
> > >
> > >
> > > As we understood, the main issue is that the norm of the estimator might grow indefinitely making our bounds vacuous. Here is the formal statement of Proposition 1 included in the revision.
> > >
> > >
> > >
> > > >> __Proposition 1.__ If $Im(A_\pi)\subseteq Im(S_\pi)$ or $\mathcal{H}$ is dense in $L^2_\pi(\mathcal{X})$, then for every $\delta{>}0$ there exists a finite rank non-defective operator $G\in HS(\mathcal{H})$ such that $\mathcal{R}(G)<\delta$. Moreover, if $A_\pi$ is self-adjoint, then, additionally, $\Vert G \Vert \leq1$.
> > >
> > >
> > >
> > > We remark that $Im(A_\pi)\subseteq Im(S_\pi)$ corresponds to a well-specified case, while the assumption that $\mathcal{H}$ is dense in $L^2_\pi(\mathcal{X})$ is common in the kernel methods literature, and it holds for many popular kernels such as the Gaussian kerne.
> > >
> > >
> > >
> > > > “ Why would the RKHS not be miss-specified?”
> > >
> > > In this respect, we refer to our comment in Appendix E.1., and to our first response to Sfxe concerning the remark on choice of the kernel.
> > >
> > >
> > > > “Why do you expect that either RKHS is well-specified or dense in $L^2_\pi$? To my understanding, this assumption is absolutely crucial. Yet no justification is made for it either theoretically or experimentally. This significantly affects the impact of the paper.”
> > >
> > >
> > >
> > > The assumption on a well-specified problem, while important, is standard in both CME and classical KRR theory. See e.g. Thm 4.1 of the review by [Muandet et al. 2017] cited in the paper, and references therein. On the other hand, RKHS being dense in $L^2_\pi$ is _not_ truly an assumption on the problem, since it solely depends on the choice of the kernel, see our reference [48] and [Micchelli, Xu and Zhang, 2006]. Further, we point out that the assumption about the well-specified problem is not crucial for our theoretical results. For instance, the uniform bounds presented in Sect. 5 do not make this assumption.
> > >
> > >
> > >
> > >
> > > Finally, concerning Points 1-3, we wish to stress that __our results go beyond classical ones, since we also address the aspect of spectral decomposition which is key in the dynamical systems setting.__

---

> ### Comment · Area_Chair_WJuW · 2022-08-08
> **Response needed**
>
> Dear Reviewer bsCx,
>
> Please kindly respond to the rebuttal provided by the authors and/or engage in the discussion with them. If it addresses your concerns, please react accordingly. Otherwise, please elaborate in your review on why you think the rebuttal/discussion is inadequate. Thank you.
>
> Best, AC

---

### Author Response · Authors · 2022-08-02
**General reply to all reviewers**

We would like to thank all reviewers for their time and effort in assessing our work. Their valuable and insightful remarks helped us to improve the quality of our paper in several ways. In this general comment we address some issues which are of interest to all reviewers:



__Additional empirical studies:__ We have followed different suggestions of all reviewers to expand the empirical evaluation in Sec. 6. In particular we performed an additional set of experiments on the Noisy logistic map and considered two more dynamical systems, the Lorenz 63 attractor and the molecular dynamics of Alanine dipeptide.



- On the logistic map and Lorenz 63 we tested empirically an important theoretical result of our paper, the uniform bound for rank-constrained operators in Theorem 3. As remarked in the manuscript, the proposed RRR estimator and the classical PCR estimator satisfy the same uniform bound. However, the empirical risk may be (possibly much) smaller for the RRR estimator and hence preferable. To this end, we evaluated, as a function of the number of training points, the empirical risk of PCR and RRR estimators under the same HS-norm constraint (needed to satisfy the assumption of Thm. 3). On both datasets RRR consistently attained *smaller empirical risk* than PCR (every experiment was repeated 100 times independently to have a solid statistical basis). Moreover, we have verified that the bound $\approx n^{-1 / 2}$ derived in Thm. 3 empirically holds. These additional results corroborate those already presented in the manuscript concerning the good performance of our RRR estimator.



- In the Alanine dipeptide experiment we showcased the use of the spectral decomposition in computational chemistry. We conducted this experiment in the early stages of the paper development, but we decided to exclude it from the final manuscript, as it requires quite a bit of physics technicalities to be fully appreciated. However, we remark that computational (bio)physics might be one of the most spectacular applications of the ML+Dynamical Systems theory, with far-reaching applications related to drug design and chemical process optimization.



Finally, we mention that in the past weeks we have polished, simplified and improved our code. Most importantly, to foster its adoption, we have implemented sklearn-compliant estimators.



The new experiments and new code can be previewed at the same repository link provided in the manuscript: https://anonymous.4open.science/r/DynamicalSystems/README.md



__One technical issue is spotted and corrected:__ After the submission of our work, we have spotted a technical oversight. Namely, what is referred to as risk $\Vert Z_\pi-S_\pi G\Vert^2_{HS}$ is in fact the excess risk w.r.t. $\mathcal{R}(G)$ of eq. (3). As it is proven in Prop. 4 of the supplementary material, $\mathcal{R}(G)=\mathcal{R}\_{ex}(G) + \mathcal{R}\_{ir}$, where the excess risk is $\mathcal{R}\_{ex}(G)=\Vert Z_\pi-S_\pi G\Vert^2_{HS}$ and the irreducible risk (i.e. the variance term in the classical bias-variance decomposition) is $\mathcal{R}\_{ir}=\Vert Z_\pi\Vert^2_{HS}-\Vert S_\pi \Vert^2_{HS}\geq 0$ which in general will be positive and not always zero as we wrote in line 606. As a consequence, instead of Eq. 9 we now have $\mathcal{R}(G)=\mathbb{E}\_{(x,y)\sim\rho} \Vert \phi(y) - G^* \phi(x) \Vert^2$, $\mathcal{R}\_{ex}(G)=\mathbb{E}\_{x\sim\pi}\Vert g_p(x) - G^* \phi(x) \Vert^2$ and $\mathcal{R}\_{ir}=\mathbb{E}\_{(x,y)\sim\rho}\Vert g_p(x) - \phi(y) \Vert^2$. We’ll correct this in the revised version, but we stress that this change has no impact on our results. Indeed, Prop. 1 and the approximation bounds of Thm. 1 and Remark 2, relate to the excess risk, while the statistical bounds in Sec. 5 relate to the full risk.

Once again we thank the reviewers and AC for their time and would be happy to provide any further clarification on our rebuttal or answer any further questions.

---

### Meta-Review · Area_Chair_WJuW · 2022-08-28

**Recommendation:** Accept
**Confidence:** Certain

**Metareview:**

This paper provides a connection between the Koopman operator theory and statistical learning theory, enabling one to approximate the Koopman operator from empirical data using the Hilbert-Schmidt operator on a reproducing kernel Hilbert space (RKHS). The expert reviewers agree that this paper contains substantial contributions that are deemed adequate for publication at NeurIPS2022.

Nevertheless, the major concerns raised by one review include the claimed novelty and the practical value of the main theoretical result (Theorem 1). In my opinion, the authors did a superb job at responding to these concerns by providing detailed responses as well as additional empirical results. The reviewer is of course free to disagree with the authors and to maintain a low score. After a reviewer discussion phase, the remaining reviewers came to a conclusion that the criticisms raised by this reviewer are valid (and hence should be addressed in the camera-ready version), but do not outweigh the merits of this paper. Specifically, the reviewers recommended that the authors ought to weaken the claim of novelty and make clearer the existing literature, especially [1] and related works in the conditional mean embedding (CME) literature.

[1] Stefan Klus et. al. 2019; Eigendecompositions of Transfer Operators in Reproducing Kernel Hilbert Spaces.

Last but not least, I do hope that the authors will respect the time the reviewers spent providing constructive criticisms by implementing the suggested changes summarized by Reviewer `bVru`.

**Award:**

No

---

### Decision · Program_Chairs · 2022-09-14

Accept